



# Weathering rates in Swedish forest soils

Cecilia Akselsson[1], Salim Belyazid[2], Johan Stendahl[3], Roger Finlay[4], Bengt Olsson[5], Martin Erlandsson Lampa[6], Håkan Wallander[7], Jon Petter Gustafsson[3], Kevin Bishop[8]

[1]Department of Physical Geography and Ecosystem Science, Lund University, Lund, SE-223 62, Sweden
5 [2]Department of Physical Geography, Stockholm University, Stockholm, SE-223 62, Sweden
[3]Department of Soil and Environment Swedish University of Agricultural Sciences, Uppsala, SE-750 07, Sweden
[4]Department of Forest Mycology and Plant, Swedish University of Agricultural Sciences, Uppsala, SE-750 07, Sweden
[5]Department of Ecology, Swedish University of Agricultural Sciences, Uppsala, SE-750 07, Sweden
[6]Water authorities, Västerås, SE-721 86, Sweden
10 [7]Department Biology, Lund University, Lund, SE-223 62, Sweden
[8]Department of Aquatic Sciences and Assessment, Swedish University of Agricultural Sciences, Uppsala, SE-750 07, Sweden

*Correspondence to*: Cecilia Akselsson (cecilia.akselsson@nateko.lu.se)





**Abstract.** Soil and water acidification was first recognised as a severe environmental problem in the 1970s. The interest in establishing 'critical loads' led to a peak in weathering research in the 1980s, since weathering is the long-term counterbalance to acidification pressure. Assessments of weathering rates and associated uncertainties have recently become an area of renewed research interest, this time due to demand for more harvest to provide renewable bioenergy. Increased demand for forest fuels increases the risk of depleting the soils of base cations produced in situ by weathering. This is the background to the research programme 'Quantifying Weathering Rates for Sustainable Forestry' (QWARTS), which ran from 2012 to 2019. The programme involved research groups working at different scales, from lab experiments to extensive modelling. The aims of this paper are to summarise the state of knowledge about weathering rates in Swedish forest soils at different scales, with an emphasis on the knowledge added by the QWARTS programme, to discuss the uncertainties in relation to sustainable forestry, and to highlight knowledge gaps where further research is needed. The variation at single-site level was large, but most sites could be placed reliably in broader classes of weathering rates. At regional to national level, the results from the different approaches were in general agreement. Comparisons of base cation losses after stem-only and whole-tree harvesting showed sites with clear imbalances between weathering supply and harvest losses, and other sites where variation in weathering rates from different approaches obscured the overall balance. Clear imbalances appeared mainly after whole-tree harvesting in spruce forests in southern and central Sweden. Research findings in the QWARTS programme support the continued use of the PROFILE/ForSAFE family of models, but it is important to continue comparisons between these and other approaches. Uncertainties in the model approaches can be further reduced, mainly by finding ways to reduce uncertainties in input data on soil texture and associated hydrological parameters. Another way to reduce uncertainties is by developing the models to better represent the delivery of weathering products to runoff waters and biological feedbacks under the influence of climate change.



## 1 Introduction

Acidification of soils and water, caused by long-range transport of acid compounds, was recognised as an environmental problem in Europe in the late 1960s (Odén, 1968). In subsequent decades, extensive research examined acidifying processes and processes counteracting acidification (Reuss and Johnson, 1986). Two key research programmes were the National Acid
Precipitation Assessment Program (1980-90, Irving, 1991) funded by the US government (USD 17 million) and the Surface Water Acidification Programme (1985-1990, Mason, 1990) funded by the UK (GBP 5 million, Mason, 1990).

The critical load concept, in which long-term weathering of base cations (Ca, Mg, Na and K) is a key parameter, was developed (Nilsson and Grennfelt, 1988), and served as a link between science and policy within the framework of the UNECE Convention on Long-Range Transport of Air Pollutants (Lidskog and Sundqvist, 2002). The critical load is defined
as "a quantitative estimate of an exposure to one or more pollutants below which significant harmful effects on specified elements do not occur according to present knowledge" (UNECE, 1994). To calculate critical loads and their exceedance, mass balance calculations of acidity are used together with a critical limit for a chemical criterion, defining the maximum acidity of soil/runoff water that can be allowed without a risk of negative effect on a chosen biological indicator (Sverdrup and de Vries, 1994).

Weathering estimates play a key role in mass balance calculations, as weathering is an important long-term natural source of base cations and sink of acidity. Weathering was therefore studied extensively, to enable accurate weathering quantifications. Various methods were used, e.g. process-based modelling (Sverdrup and Warfvinge, 1993), soil measurements where the depletion of weathering products in different soil layers is determined in order to assess average weathering rates since soil formation, i.e. the last ice age (Olsson et al., 1993), and budget calculations where all other parameters except weathering are
measured (Lundström, 1990; Jacks and Åberg, 1987; Wickman and Jacks, 1991; Sverdrup et al., 1998).

The political and scientific agreement on the critical load concept as a basis for managing acid deposition was a major factor in subsequent policy success on limiting acidifying emissions (Lidskog and Sundqvist, 2002). However, one side-effect of the success in agreeing on specific weathering model estimates as the basis for long-term policy agreements to reduce atmospheric emissions was that interest waned in further weathering research that might revise these weathering estimates.
Since critical loads were aimed at establishing long-term balances, those agreed reductions did not address specifically the recovery of soils and waters already acidified by acid deposition. Nonetheless, major acid deposition reductions occurred, and some recovery of soils and surface waters has been noted by the monitoring operations put in place by the UNECE CLTRAP (Graf Pannatier et al., 2011; Pihl Karlsson et al., 2011; Akselsson et al., 2013).

As the severity of climate change became more fully recognised, the demand for renewable fuels increased rapidly, thereby
increasing the pressure on forests. Whole-tree harvesting, here defined as harvesting of branches, was seen as an important source of renewable fuel. Since 2000 in Sweden, the proportion of clearcuts involving whole-tree harvesting has increased from around 15% to 25-35%, according to statistics from the Swedish Forest Agency, except for 2014-2016 when a drop in





energy prices reduced demand and the proportion declined to 15-25%. Demand is likely to increase in the future (Börjesson et al., 2017).

Harvesting of branches and the nutrient-rich needles also means a substantially increased removal of base cations compared to conventional harvesting. Several studies have shown that this may counteract the recovery from acidification. Olsson et al. (1993) compared Ca, Mg and K weathering rates when applying the depletion method to estimate losses of those base cations at different harvesting intensities. They found that harvest losses of Ca generally exceeded the Ca weathering rates already at stem-only harvesting, and this pattern was even more noticeable at whole-tree harvesting. For Mg and K, the weathering rates were generally higher than the losses at stem-only harvesting, but for whole-tree harvesting the Mg and K losses exceeded the weathering rates over large areas, especially in southern Sweden. They concluded that Ca in particular, but also Mg and K in certain regions, will be depleted in soils if whole-tree harvesting is applied.

Akselsson et al. (2007) used the PROFILE model, along with mass balances, including deposition and leaching, and drew similar conclusions. Both stem-only and whole-tree harvesting gave negative balances in most areas in Sweden for all three elements in spruce forests, and for Ca and Mg in pine forests. Whole-tree harvesting increased the net losses substantially, especially for Ca and K. Iwald et al. (2013) compared the acidification effect of biomass removal with the effect of acid deposition for 1996-2009, and concluded that the acidifying effect of harvesting of stems, branches and stumps in spruce forests was 110-260% of that of acid deposition. For pine, the corresponding interval was estimated to be 60-110%. The importance of whole-tree harvesting was also demonstrated in a study from Finland, where it was shown that whole-tree harvesting doubled the removal of base cations (Aherne et al., 2012).

The prospect of increasing demand for forest biomass and particularly concerns about the effects of whole-tree harvesting on nutrient sustainability renewed the interest in weathering for new forest policy issues. However, the accuracy of the weathering calculations, and the conclusions about the long-term sustainability of forests, were questioned by Klaminder et al. (2011), who compared the weathering rates for Ca and K at a site in northern Sweden, using different approaches. These estimates differed widely, and the study concluded that nutrient budgets, based on calculations including weathering rates, are too uncertain to be useful in shaping forest policies regarding harvest practices.

Futter et al. (2012) suggested that some of this variation is due to differences in boundary conditions, for example the depth to which weathering has been calculated. They examined weathering estimates from 82 sites, with up to eight different weathering estimates per site, and found considerable variability in weathering rates estimated with different methods, often with results differing on the same site by several hundred percent. They identified uncertainties in input data as the largest contributor to the variability, but differences in the soil depth for which weathering was calculated also contributed, in the same way as in Klaminder et al. (2011). Futter et al. (2012) concluded that the uncertainties are large, and that at least three independent methods should be used when making management decisions.

In 2012, an SEK 25-million research programme, 'Quantifying Weathering Rates for Sustainable Forestry' (QWARTS) was started in Sweden. The programme, which focused on weathering rates for the base cations Ca, Mg, Na and K, included approaches covering the whole spectra from lab-scale experiments, through plot and catchment scale experiments in the



field, to extensive weathering modelling. Different approaches, including modelling, the depletion method, mass balance calculations and total analysis regression were tested at different scales and in some cases refined (Stendahl et al., 2013; Casetou-Gustafson et al., in review a (this issue); Casetou-Gustafson et al., in review b (this issue); Akselsson et al., 2016; Belyazid et al., in review (this issue); Kronnäs et al., in press (this issue); Erlandsson et al., 2016).

The input data were also examined for uncertainties relating to the generalisations made when estimating normative mineralogy based on total chemical analyses (Casetou-Gustafson et al., 2018; Casetou-Gustafson et al., in review (this issue)). Two other potential sources of uncertainties that have been explored, but that are still widely discussed, were revisited. One source was the role of biological weathering that might generate weathering not included in the current generation of biogeochemical models (Banfield et al., 1999; Finlay et al., 2009; Finlay et al., in review (this issue)). The
other source concerned simplifications relating to base cation exchange and aluminium complexation (Tipping 2002; Gustafsson et al., 2018 (this issue); van der Heijden et al., 2018). Furthermore, weathering rates representing not only the rooting zone, but the full catchment scale, were studied, to assess the export of weathering products to surface waters (Ameli et al., 2017; Erlandsson et al., in review (this issue)).

     While weathering is important for understanding the acidification of both soils and surface waters, this paper focuses on
summarising the work of QWARTS on quantifying base cation weathering rates in the rooting zone (approximately 50 cm) of Swedish forest soils. Findings from single, well-investigated sites up to the regional and national scales are synthesized and compared. Weathering rates from new weathering studies, a few based on new datasets, have been complemented with a revisiting of older weathering studies, in the cases where methods and assumptions have been thoroughly described. Several different approaches are included, and weathering rates from the ForSAFE model are included in weathering rate
comparisons for the first time (Kronnäs et al., in press (this issue); Belyazid et al., in review (this issue)). The results are compared with the base cation flows from forest harvesting, and are discussed from a forest sustainability perspective. Possible implications of new insights regarding effects of biological weathering (Finlay et al., in review (this issue)) and a more accurate description of base cation exchange and aluminium complexation (Gustafsson et al., 2018 (this issue)) are discussed. Finally, future research directions, aiming to reduce uncertainties, are outlined, based on the overall progress
achieved in the QWARTS programme.

## 2 Methods for estimating weathering rates

The release of the base cations Ca, Mg, Na and K through weathering is difficult to quantify through direct measurements in the field due to the complexity of base cation dynamics in soil. There are several different pools of base cations in soil, and also several different flows, e.g. decomposition, uptake, ion exchange and weathering, and it is difficult to distinguish
between these different flows (Rosenstock et al., in review (this issue)), or even to define the pools accurately (van der Heijden et al., 2018). Therefore, a number of indirect methods to quantify weathering rates have been developed.



## 2.1 Modelling

Mechanistic modelling, based on laboratory experiments, describing the kinetics is one way to get around the difficulties of measuring weathering in the field. The PROFILE model (Sverdrup and Warfvinge, 1993) is a steady state soil chemistry model with process-oriented descriptions of chemical weathering and solution equilibrium reactions, and has been used

widely for estimating weathering in Europe (Akselsson et al., 2004; 2016; Stendahl et al., 2013; Holmqvist et al., 2003; Koptsik et al., 1999; Langan et al., 1995), the US (Phelan et al., 2014) and Asia (Fumoto et al., 2001).

Weathering rates are calculated for different layers, with different soil properties, using transition state theory and the geochemical properties of the soil system, such as soil wetness, temperature, mineral surface area and mineral composition, and organic acid concentrations. Deposition of sulphur, nitrogen and base cations, as well as net losses of base cations and

nitrogen through harvesting, are used as input for modelling of pH and base cation concentrations in soil water, which is required for the weathering modelling. Weathering rates are calculated for each mineral separately, using rate coefficients from laboratory studies for four reactions: with hydrogen ions, water, carbon dioxide and organic ligands (Sverdrup and Warfvinge, 1993).

The weathering submodel in PROFILE was later built into the dynamic version of PROFILE, SAFE (Alveteg et al., 1995)

and in the forest ecosystem model ForSAFE (Wallman et al., 2005; Belyazid et al., 2006), where the SAFE model has been coupled with the tree growth model PnET (Aber and Federer, 1992), the decomposition model DECOMP (Walse et al., 1998; Wallman et al., 2004) and the hydrology model PULSE (Lindström and Gardelin, 1992). SAFE is used mainly for acidification assessments, and ForSAFE is used for studying effects of climate change, atmospheric deposition and forest management on tree growth, soil chemistry and runoff water quality.

The models have also been used to explicitly study weathering rates (Warfvinge et al., 1995; Kronnäs et al., in press (this issue); Belyazid et al., in review (this issue)). The ForSAFE model simulates the integrated biogeochemical processes of a forest ecosystem. It covers the processes of photosynthesis, allocation and growth, water and nutrient uptake, litterfall, organic matter decomposition and mineralisation, ion exchange, chemical speciation of and reactions between different elements, as well as hydrological transport (Wallman et al., 2005). All process rates are internally regulated by

microenvironmental conditions such as acidity, water availability, temperature and element concentrations. The model requires inputs of external drivers in the form of climate, atmospheric deposition and forest management, and inputs on the properties of the forest ecosystem, such as soil texture, mineralogy and tree species.

Internationally, some other weathering models such as WITCH (Godderis et al., 2006) or CrunchFlow (Maher et al., 2009) employ transition-state theory rate laws for the kinetic descriptions of mineral dissolution (Aagaard and Helgeson, 1982),

just like in PROFILE, SAFE and ForSAFE. However, there is a difference from the kinetic equations of PROFILE, SAFE and ForSAFE in terms of response to pH and aluminium concentrations (Erlandsson et al., 2016). Nonetheless, PROFILE has previously been demonstrated to replicate field weathering rates as determined by independent methods for a wide range of soils without calibration (Sverdrup and Warfvinge, 1993).



With a few exceptions, all process-based weathering models have been restricted to the 1D plot scale and unsaturated flow. Model estimates of the weathering flux from larger volumes of soil, such as hillslopes or catchments, are rare. Exceptions include the use of PROFILE for surface waters (Rapp and Bishop, 2003), the WITCH model to calculate the weathering rates in the Vosges catchment in Luxemburg (Godderis et al., 2006), and Erlandsson et al. (in review, this issue)), who used a mixing model, where the water transit time distribution and weathering rates were modelled separately to estimate the weathering flux from a hillslope in the Vindeln research park in Northern Sweden.

The MAGIC model (Cosby et al., 2001) was developed to predict effects of acidic deposition on surface water acidification. Weathering is not mechanistically modelled as in the models described above; instead, weathering rates are calculated internally using mass balances (Maxe, 1995; Köhler et al., 2011). MAGIC uses input fluxes from atmospheric deposition and weathering and output fluxes through net uptake and loss in biomass and to runoff. These fluxes govern processes in the soil, e.g. cation exchange, with the pool of exchangeable base cations in the soils at the centre. When the fluxes change over time, it affects the chemical equilibria between soil and soil solution, which has an impact on surface water chemistry. Observed values of surface water and soil chemistry are used to calibrate the model.

### 2.2 The depletion method

Another approach for estimating weathering rates is the depletion method. The method estimates historical weathering, i.e. the average weathering rate since the last deglaciation, of mobile (weatherable) elements, based on element concentrations in weathered soil horizons as compared to unweathered parent material. The method accounts for the general losses of soil material in a horizon by including an immobile (inert) element in the estimation. Concentrations of mobile elements will decrease as a result of weathering, while the immobile element will be enriched towards the soil surface. The concept has a long history (Marshall and Haseman 1942), while the theoretical framework was later formalised by Brimhall and Dietrich (1987) and Brimhall et al. (1991). The most commonly used immobile element is zirconium, which is found in the resistant mineral zircon ($ZrSiO_4$) with negligible weathering (Hodson, 2002). The assumptions for the depletion method are: (1) there is no weathering of the immobile element, (2) the soil pedon consists of homogeneous parent material, and (3) no weathering occurs beyond a certain depth. The average annual weathering rate is calculated from the soil age, i.e. the time since deglaciation or since the land rose from the sea due to glacio-isostatic uplift. The average rate may deviate from current levels depending on the variation in weathering rates over time (Taylor and Blum, 1995).

### 2.3 Budget calculations

Weathering rates can also be estimated through mass balance calculations (Paces, 1986; Lundström, 1990; Sverdrup and Warfvinge, 1991; Sverdrup et al., 1998). In mass balance calculations, sources and sinks of base cations are considered, and weathering is calculated as the difference between sinks and sources. However, one difficulty is to distinguish between weathering, changes in the exchangeable pool and net mineralisation, so steady state is often assumed, and the weathering rates are calculated as leaching + net uptake – deposition. This approach was applied in Svartberget in northern Sweden



(Lundström, 1990). In Gårdsjön in southwestern Sweden, net mineralisation was estimated and considered in the weathering estimations, but changes in the exchangeable pool were disregarded (Sverdrup et al., 1998). Simonsson et al. (2015) estimated weathering rates over a 14-year period in Skogaby in southern Sweden, based on measurements of atmospheric deposition, leaching, accumulation in biomass and changes in the exchangeable pool. The same approach was used by

Casetou et al. (in review b, this issue)) in the mass balance estimates for Asa and Flakaliden in southern and northern Sweden, respectively. In the latter two studies, leaching, soil change and accumulation in biomass were measured in control plots of long-term fertilisation experiments in young Norway spruce forests. When using weathering estimates from budget calculations, the assumptions used must be carefully evaluated (Sverdrup and Warfvinge, 1991; Rosenstock et al., in review (this issue)).

**2.4 Budget calculations using the strontium isotope ratio**

In weathering estimations based on strontium (Sr) isotope ratios, the difference in the ratio $^{87}$Sr/$^{86}$Sr in bedrock and in atmospheric deposition is used (Wickman and Jacks, 1991). Since soil water is a mixture of what comes from deposition and what comes from weathering, the weathering rate of Sr can be estimated. Ca and Sr follow each other closely in forests (Wickman and Jacks, 1991), so the weathering of Ca is assumed to be linearly correlated to the weathering of Sr in the

calculations. To estimate base cation weathering, as was done for Gårdsjön (Sverdrup and Warfvinge, 1993), and in Svartberget and Risfallet (Sverdrup and Warfvinge, 1991), a constant base cation/Ca fraction is assumed.

**2.5 Total analysis regressions**

The total analysis regression method is based on the fact that weathering rates of different elements have been found to correlate with total content of the elements in soil and temperature (Olsson and Melkerud, 1990). Based on weathering

estimations from the depletion method, linear regressions containing total chemical contents for base cations in the C horizon (either separately or lumped together), and temperature or temperature sums, have been produced for a number of sites, and the regressions have then been applied to other sites (Sverdrup et al., 1998; Maxe, 1995; Olsson et al., 1993).

**3 Weathering rate comparisons at site level**

Weathering rates have been estimated on a number of well-investigated sites in Sweden, using the methods described above.

Here the weathering rates are presented for sites where at least two methods have been used for estimations to the same soil depth, and for which method descriptions have been found. This enabled us to quantify the span of weathering rates produced by the different approaches, as a way of "framing" the weathering rates on the sites (Fig. 1-2, Table 1).

At Gårdsjön, the different approaches resulted in weathering rates in the interval 36-62 meq m$^{-2}$ yr$^{-1}$ (Table 1, Fig. 2). Some of the differences can be explained by the somewhat different depths; the highest rates in the interval are associated with a

profile 17-20 cm deeper than the other ones. The similarities in weathering rates indicate that the framing of the weathering



rates at Gårdsjön has been successful, i.e. that a limited range of weathering rates at Gårdsjön has been successfully defined. Moreover, it indicates that the soil conditions in the Gårdsjön area are rather homogeneous.

For Svartberget B (0.8 m), three methods gave weathering rates within the interval 31-42 meq m$^{-2}$ yr$^{-1}$ whereas the fourth one, the budget study, produced a substantially higher weathering rate, 85 meq m$^{-2}$ yr$^{-1}$. The aim of the budget study was to

evaluate the importance of organic substances on weathering rates, which involved comparing two soil compartments, 0-20 cm and 30-80 cm, estimated as the difference between the sum of base cation deposition and base cation losses and the sum of net uptake and leaching (Lundström, 1990). The weathering estimates from the budget calculations can be expected to be overestimated for several reasons. The method does not distinguish between weathering, base cation exchange and base cation release through decomposition. The measurements were carried out in the 1980s when the acidification process was

taking place, leading to base-cation release from the exchangeable pool, although this effect was much more pronounced in southern Sweden. Moreover, dry deposition was not included in the calculations due to lack of data.

The nearby sites, Svartberget A (0.5 m) and Vindeln (0.5 m), were expected to give lower weathering rates than Svartberget B, due to the shallower depth, and this proved to be the case, 17-38 meq m$^{-2}$ yr$^{-1}$ for Svartberget A and 13-30 meq m$^{-2}$ yr$^{-1}$ for Vindeln. The lower values in the intervals are based on the depletion method, whereas the higher are from PROFILE.

Although the results from Svartberget were more scattered than for Gårdsjön, it could be concluded that the weathering rate is probably lower at Svartberget than at Gårdsjön (Table 1, Fig. 2).

At Risfallet B (1 m), two methods were applied to the same depth, PROFILE and the Sr method. They gave similar results, 25-29 meq m$^{-2}$ yr$^{-1}$. It is notable that the weathering rate for Risfallet A (0.5 m) was higher, in spite of the shallower depth, with an interval of 29-68 meq m$^{-2}$ yr$^{-1}$, where the higher value is from PROFILE. The weathering rate modelled by

PROFILE in Risfallet A (0.5 m) is one of the highest of all sites reported in Stendahl et al. (2013), which can be explained by a relatively high clay content (7%) and high density. In contrast, Risfallet B (1 m) has a very low specific surface area (Sverdrup and Warfvinge, 1993), much lower than in Risfallet A (0.5 m), which can explain the low weathering rate produced by the model. Based on the approaches used so far at Risfallet, the weathering rate is hard to to define with such a high level of accuracy as for Gårdsjön and Svartberget (Table 1, Fig. 2), which can be at least partly explained by varying

soil texture within short distances.

At Stubbetorp, total analysis regression and MAGIC gave weathering rates in the interval 30-51 meq m$^{-2}$ yr$^{-1}$, whereas PROFILE gave a higher rate, 67 meq m$^{-2}$ yr$^{-1}$. Maxe (1995) noted that the PROFILE weathering rate was higher than expected, given the soil properties in Stubbetorp, and argued that it may be due to unreasonably high specific surface area of the soil as input to PROFILE on the site. Specific surface area has been determined by BET analysis, and could, according to

Maxe (1995), be overestimated due to a large occurrence of Al and Fe precipitates.

For Flakaliden and Asa, three different weathering estimates were available, and the span was very large, particularly for Asa in southern Sweden. The mass balance method produced the highest weathering rates at both sites. However, weathering estimates for Flakaliden by the mass balance method and PROFILE were similar to the estimates produced by the same methods for the nearby (≈ 40 km) Svartberget B site. For Flakaliden, PROFILE produced a weathering rate twice as high as





that produced by the depletion method, which is the same order of magnitude as many of the other sites. However, at Asa the estimated weathering rate with the depletion method was very low, 11 meq m$^{-2}$ yr$^{-1}$, which is among the lowest estimated weathering rates found on all sites, with all methods, whereas PROFILE gave a weathering rate of the same size as many of the other sites. At both Asa and Flakaliden, base carion accumulation in biomass was the major sink in the mass balances.

The high weathering rates produced by the mass balances were largely explained by the measured depletions of base cations in the soil being much lower than the accumulation in the young Norway spruce stands. The very high mass balance weathering rate at Asa was therefore mainly attributed to the higher growth rate at that site. The very low weathering rates produced by the depletion method at Asa originated from a fairly flat Zr depth gradient in the soil, which indicates that the soil had probably been disturbed, so the necessary assumptions for the Zr depletion method were not satisfied (Casetou

Gustafson et al., in review b (this issue)).

At Västra Torup and Hissmossa, weathering rates produced by PROFILE and ForSAFE were compared in detail for the first time. Although the process descriptions of weathering are the same in the two models, the dynamics of ForSAFE were expected to produce differences in estimated weathering rates. The results at Västra Torup showed that ForSAFE produced weathering rates 8% higher than the rates produced by PROFILE. At Hissmossa, PROFILE gave 19% higher weathering

rates than ForSAFE. Much of the difference could be attributed to the difference between soil moisture input data in PROFILE and modelled soil moisture in ForSAFE. The sandy soil in Hissmossa gave substantially lower modelled soil moisture than the moisture estimates based on field observations used as inputs to PROFILE, resulting in substantially lower weathering rates.

Three of the 16 sites reported in Stendahl et al. (2013) were disqualified. One site in the north gave unrealistically high

weathering values with PROFILE, which is probably at least partly due to a very high bulk density; the simplified hydrology routine in PROFILE, with only vertical flow, is unsatisfactory for these conditions. For the other two sites the profiles did not fulfil the assumptions for the depletion method. The weathering rates calculated with the depletion method for the remaining 13 sites were generally lower, on average 50% lower, than the PROFILE-modelled weathering rates. This is contrary to what was expected, since the weatherability of the soil is believed to decrease over time due to the depletion of

more easily weathered minerals and formation of resistant coatings on the mineral surfaces (Taylor and Blum, 1995). For Mg, the depletion method did indeed give higher values than PROFILE, in accordance with the theories, but for Ca and Na the reverse applied. The reasons are not fully known, but probably the declining weatherability over time is less pronounced in these young glacial till profiles, where the easily weathered minerals remain in the profile. Furthermore, some drivers of weathering, such as forest growth, are more prominent today, which may overshadow long-term decline in soil

weatherability. Another plausible factor is that the original till probably partly comprises already weathered till from previous glaciations (Stendahl et al., 2013). For the total sum of base cations there was a tendency towards higher modelled rates when the rates from the depletion method were higher, but the relationship was weak (r$^2$=0.20) on the 13 sites. However, for Ca and Mg there was a much stronger relationship, r$^2$=0.46 for Ca and r$^2$=0.64 for Mg. The relationship was weaker for K and Na, possibly due to conceptual limitations in the models. Based on current knowledge and models, it



seems possible to identify a narrow weathering rate interval Ca and Mg weathering rates, but it seems difficult for K and Na, as discussed in Stendahl et al. (2013).

In Fig. 2 the intervals of all sites are compared with four reference weathering intervals, based on weathering rate approximations frequently used in the critical load work (de Vries, 1994). The four intervals are based on different

combinations of parent material (acidic, intermediate and basic) and texture (coarse-, medium- and fine-grained). The coarse fraction, defined as soil with a clay content of less than 18 % (Umweltsbundesamt, 1996) is predominant in Sweden, and acidic parent material, defined as material composed of e.g. sandstone, granite and gneiss, is the most common parent material in Sweden.

The intervals for many of the sites were within or close to the interval outlined for acidic/intermediate parent material with

coarse texture, corresponding to Swedish conditions (Fig. 2). Gårdsjön had small enough intervals to be able to define the intervals in the weathering rates with relatively high accuracy. This was also the case for Svartberget, if the outlier from the budget calculations was disregarded. Risfallet showed more contradictory results, which are hard to interpret, since the soil sampling was performed several decades ago. The weak correlation between the weathering rates from PROFILE and the depletion method in Stendahl et al. (2013) makes it difficult to, in detail, rank all the sites based on the weathering rates.

However, some patterns are clear, e.g. that Gårdsjön seems to have higher weathering rates than most other sites, whereas Vindeln and Hjärtasjö lie at the lower end.

The weathering rates for the sites with 0.5 m soil depth (including Gårdsjön G1 where the soil depth is 0.47 m) can roughly be divided into four different groups depending on the intervals, except for four sites with contradictory results (Table 2).

The assessment of whether the weathering rate intervals are accurate enough depends on their intended use. The weathering

rates in relation to forest sustainability assessments are analysed in Sect. 7.

## 4 Weathering rate comparisons on a regional scale

Weathering rates have been up-scaled to the regional level in Sweden using three different methods: the total analysis regression method (Olsson et al., 1993), the PROFILE model (Akselsson et al., 2004; 2016) and the ForSAFE model (Belyazid et al., in review (this issue)). The three studies were performed independently from each other. PROFILE and

ForSAFE are built on the same basic concepts for weathering calculations, with the difference that ForSAFE is dynamic. This means that climate, deposition and forest changes over time are taken into account, that soil moisture is modelled dynamically based on precipitation data and soil properties, and that there are feedbacks between tree growth, decomposition, weathering and ion exchange. The ForSAFE calculations have been made on a subset of the sites that were used for PROFILE, so the input data are at least partly the same. However, even when the sites are the same, there are some

differences in the input data, e.g. depending on different methodologies for estimating mineralogy from total chemistry. Those estimations require assumptions about qualitative mineralogy on the sites (Casetou-Gustafson et al., in review a (this



issue)), and those assumptions might differ between different modellers. The three approaches are described in detail below, and then compared for different climate regions in Sweden.

### 4.1 Total analysis regressions

Weathering rates for Ca, Mg and K has been calculated on a regional level in Sweden, based on the total analysis regression method, as a basis for assessments of nutrient sustainability after whole-tree harvesting (Olsson et al., 1993). In the study from 1993, regressions between weathering rates calculated with the depletion method and different site factors were analysed on 11 sites. The strongest relationships were found between weathering rates of an element and the product of the concentration of the element in the C horizon and the temperature sum. In the present study, the regression functions in Olsson et al. (1993) were used to calculate weathering rates of Ca, Mg and K on 640 sites in the SAFE database (Alveteg, 2004), which contains the data required for the regression calculations. The weathering rates estimated with the functions needed to be corrected for the fraction of stones and boulders, which was available on 512 of the sites (Stendahl et al., 2009). The temperature sum was calculated based on latitude and altitude according to Morén and Perttu (1994). Some of the calculations gave negative results for one of the base cations. This can be explained by the fact that the regressions are used for a new dataset, covering a broader range of temperature sums than the original dataset, which illustrates a limitation of the method. The sites where rates were negative were removed, which reduced the database to 445 sites. The estimations apply down to the weathering depth, which means different soil depths on different sites, but normally between 40 and 70 cm in the mineral soil (Mats Olsson, pers. comm).

### 4.2 PROFILE

The latest PROFILE weathering map for Sweden is presented in Akselsson et al. (2016), where weathering rates for the upper 50 cm of the soil (including the organic layer) were modelled based on data from 17,333 Swedish National Forest Inventory (NFI) sites (Fridman et al., 2014), of which the 640 SAFE sites are a subset. PROFILE requires soil input data, as well as climate and deposition data and information about net uptake of nutrients in trees. Mineralogy was derived from an earlier regional study (Akselsson et al., 2007), where total chemistry data from the Geochemical Atlas of Sweden (Andersson et al., 2014) has been used to calculate normative mineralogy for each site. Soil texture and moisture classifications are available on all NFI sites, and were transferred to specific surface area and volumetric water content using translation tables from Warfvinge and Sverdrup (1995). The same dataset as above was used to correct for stones and boulders. Net uptake of base cations and nitrogen was calculated based on tree growth data on the NFI sites. Deposition data from the MATCH model (an average for 2006-2008) was used (Langner et al., 1996). Temperature data from SMHI, representing 1981-2010, were taken from Akselsson et al. (2016). The methodology is described more in detail in Akselsson et al. (2007; 2008; 2016).



### 4.3 ForSAFE

Weathering rates to 50 cm depth (including the organic layer) were modelled with the ForSAFE model on the 640 sites in the SAFE database (Alveteg, 2004). Information on soil texture and mineralogy were derived from the database of the Research Infrastructure National Forest Inventory of Sweden (RINFI, Hägglund, 1985). Mineralogy was derived from total

elemental analysis translated through the UPPSALA model (Sverdrup et al., 2002). Stones and boulders were taken into account using the same data on contents of stones and boulders as for PROFILE and the depletion method.

The climatic input data were derived from simulations of historical and future trends, based on the IPCC's A2 story line of emissions (David Rayner, pers. comm.) and downscaled to historical records from the Swedish Meteorological Institute (SMHI). The climate data include monthly information on air temperature, precipitation, photosynthetically active radiation

and atmospheric $CO_2$ concentrations.

Atmospheric deposition data for sulphate, nitrate and ammonium were derived from simulations by the EMEP model (Simpson et al., 2012) according to the emissions history in Schöpp et al. (2003) and the projected emissions following the current legislation of the UN Convention on Long Range Transboundary Air Pollution (LRTAP). Atmospheric deposition data for chloride, calcium, sodium, potassium and magnesium were derived from the MATCH model (Persson et al., 1996).

### 4.4 Comparison between regional estimates produced by the three methods

In order to compare the results from the different methods regionally, a subset of 346 sites where all three methods were applied was used. These were the sites for which all methods had been applied successfully, and where data on stones and boulders were available. For a regional comparison, the sites were divided into seven climate regions, simplified from 19 weather forecast regions used by the Swedish Meteorological and Hydrological Institute (SMHI) (Fig. 3-4). One of the

regions, northwestern Sweden, only contained one site, and was therefore excluded from the analysis.

The weathering rates varied widely within the regions for all methods, but there were no large systematic differences between the medians or ranges for the different methods. However, ForSAFE gave somewhat higher medians than PROFILE for all regions, especially in the northern regions. One explanation could be differences in the method for estimating mineralogy from total chemistry, where 'possible' minerals have to be set by the modeller. Since qualitative data on mineral

contents in soils in most cases are not available at the modelling sites, data must be based on a number of assumptions. In the ForSAFE database, limestone seems to have been set as a possible mineral more often than in the PROFILE database, due to differences in the assumptions made. Since even a small amount of limestone has a large effect on weathering rates, the Ca weathering rates are substantially higher at some sites in the ForSAFE database.

The total analysis regression method gave somewhat higher weathering rates than PROFILE for several of the regions. This

was contradictory to the site-level comparisons, where PROFILE generally gave substantially higher weathering rates than the depletion method. This can partly be explained by the site-level comparisons being made for the same depths (50 cm), whereas the regional calculations with the total analysis regression method gives the weathering to the maximum weathering




depth, which is often more than the 50 cm (including a 10 cm organic layer) used for PROFILE. Methodological differences between the old and the new calculations for the depletion method can also be part of the explanation, e.g. concerning how the weathering depth has been defined based on curves of the elemental variation with depth, which is partly a subjective operation. Finally, the fact that the regional calculations combine two methods (depletion method and total analysis

regression), whereas the site-level estimates are based only on the depletion method, may contribute to the differences.

In most cases, the comparison between regions showed no major differences. An exception is region 4, the eastern part of central Sweden, where the span was wider, including much higher weathering rates than the other regions. In region 4, lime rich soils are common, which may explain this pattern. Smaller differences could be distinguished between the other regions. The ranges in the southern regions (5 and 6) were generally on a somewhat higher level than the others, whereas the western

part of central Sweden was towards the lower end.

Although PROFILE and ForSAFE were based on the same weathering module and the same input database, it is not self-evident that they give similar results. The dynamics of ForSAFE, which e.g. involves dynamic modelling of moisture content in soil instead of a fixed value based on rough field assessments, could lead to differences, as discussed in Kronnäs et al. (in press, this issue). Moreover, different methods for processing input data, e.g. estimating mineralogy from total chemistry,

may cause differences (Casetou-Gustafson et al., in review a (this issue)). Despite those differences, the methods gave comparable results, with similar weathering rate levels and geographical patterns. The depletion method, based on a completely different concept, also gave similar results.

## 5 Potential for biological weathering

Plants play a fundamental role in soil formation, since root activity and decomposing plant material increase weathering rates

by producing acidifying substances (H+, organic acids) and ligands that form complexes with metals in the minerals. In addition, uptake of ions released from weathering reduces the likelihood of saturated conditions that retard weathering rates. Many of these effects are mediated by mycorrhizal fungi. Biological weathering often takes place in conjunction with physical and chemical processes but there is still disagreement over the extent of its quantitative contribution to overall weathering (Finlay and Clemmensen, 2017; Leake and Read, 2017; Smits and Wallander, 2017). Below, a description of

how biological weathering is presently represented in the PROFILE/ForSAFE models is given, followed by a discussion about biological weathering, the role of mycorrhizal fungi and potential shortcomings in the PROFILE/ForSAFE models. A more thorough description of the state of knowledge and a more comprehensive discussion can be found in the article by Finlay et al. (in review, this issue).

Chemical elements are released from minerals to a dissolved form following four pathways in the PROFILE/ForSAFE

models: the first dependent on soil solution H+ concentrations, the second on water availability, the third on the partial $CO_2$ pressure, and the fourth on the concentrations of dissolved organic carbon (DOC) (Sverdrup and Warfvinge, 1993). The four weathering pathways are chemical (the dismantling of mineral matrices by charged or dissolving particles to produce free





elements), but their drivers are strongly dependent on biological activity in the soil. Soil solution $H^+$ is determined by the charge balance resulting from uptake, ion exchange, mineralisation of organic matter (solid and dissolved), and hydrological transport, all of which are affected by biological activity. Water availability is directly controlled by water uptake. The partial pressure of dissolved $CO_2$ stems from decomposition and hydrological transport. Finally, DOC is directly and indirectly produced by plants.

The transition state theory governing the weathering kinetics in PROFILE/ForSAFE dictates that the net weathering rates should decline towards zero near equilibrium. This is represented in the model by retardation factors that increase in strength with the concentrations of the weathering products (Erlandsson et al., 2016). These concentrations are in turn dependent on biological activity, such as uptake reducing nutrient base cation concentrations or the mobilisation of aluminium through biological acidification.

Although the weathering process is strongly affected by biological processes in the current generation of the PROFILE/ForSAFE family of models, the models still fail to capture the biological feedback mechanisms in their entirety. Carbon allocation for example is still rudimentary in the model, and more elasticity in carbon allocation is needed to capture the empirical observations. Exudation, another example, seems to be a more active process in response to nutrient status than is currently assumed in the models.

The possible roles of fungi in biological weathering in boreal forests were summarised by Finlay et al. in 2009, and have been the subject of many subsequent studies. Prior to the QWARTS project the widespread occurrence of tubular pores, 3-10 µm in diameter, was demonstrated in weatherable minerals in podzol surface soils and shallow granitic rock under European coniferous forests (van Breemen et al., 2000; Jongmans et al., 1997; Landeweert et al., 2001). Fungal hyphae were found occupying some of these pores and it was speculated that they could be formed by the weathering action of hyphae (and possibly associated bacteria), releasing organic acids and siderophores. The aetiology of pore formation has been questioned, however, with some authors claiming that the observed pores are of abiotic origin (Sverdrup, 2009), and their quantitative contribution to total weathering rates has been calculated to be negligible (Smits et al., 2005). This means either that fungal weathering is negligible, or that tunnel formation reflects only a small proportion of the total weathering effect of the fungi.

The endolithic biosignatures of rock-inhabiting microorganisms can be distinguished from purely abiotic microtunnels (McLoughlin et al., 2010), and the biomechanical mechanisms used by fungi to penetrate rock have received increasing attention. Fungal-mineral attachment, biomechanical forcing, and altered interlayer spacing associated with depletion of potassium from biotite by a mycorrhizal fungus have been demonstrated (Bonneville et al., 2009). Extensive mineral surfaces are accessible for microbial colonisation, and atomic force microscopy has been used to demonstrate nanoscale alteration of surface topography and attachment and deposition of organic biolayers by fungal hyphae (McMaster, 2012; Gazze et al., 2013; Saccone et al., 2012). These nanoscale mineral-fungal interactions undoubtedly occur, but their quantitative significance has yet to be revealed.

There is an extensive literature on the role of fungi as biotic agents of geochemical change (Gadd, 2013a, b; 2017). We know that ectomycorrhizal fungi can allocate carbon selectively to different minerals (Rosling et al., 2004; Smits et al., 2012). The



latter (laboratory) study demonstrated that, when P was limiting, 17 times more plant-derived C was allocated to ectomycorrhizal fungal mycelium of Paxillus involutus colonising apatite than to mycelium colonising quartz, as well as that fungal colonisation of the substrate increased the release of P by a factor of almost three. Grain-scale 'biosensing' (differential colonisation) of different minerals by the same fungus has also been demonstrated by Leake et al. (2008) and

oxalate secretion by this fungus also appears to be mineral-specific (Schmalenberger et al., 2015). These, and other, similar laboratory experiments suggest that plant-fungal-mineral interactions are tightly coupled, and that distinct, local weathering environments exist.

However, there is disagreement over the extent to which the observed laboratory-scale processes contribute to soil-scale mineral dissolution rates and field processes. One view is that the coevolution of fungi and plants has enabled them to exert

increasing influence as biogeochemical engineers. The ubiquity and significance of lichens as pioneer organisms in the early stages of mineral soil formation, and as a model for understanding weathering in a wider context, are well understood (Banfield et al., 1999). It is argued that, during evolution, successive increases in the size of plant hosts and the extent of substrate colonisation by their fungal symbionts (Taylor et al., 2009; Leake and Read, 2017) have enabled them to have larger effects as biogeochemical engineers, affecting the cycling of nutrients and C at an ecosystem and global level. These

ideas are based on observations of alteration of silicate surfaces in the proximity of roots and associated mycorrhizal fungi of different trees exposed to atmospheres with different levels of $CO_2$ (Quirk et al., 2012; 2014).

It is accepted that ectomycorrhizal fungi access and degrade organic nitrogen sources (Lindahl and Tunlid, 2015), and soil carbon storage has been shown to be greater in ecosystems dominated by ectomycorrhizal plants than in systems dominated by other types of mycorrhiza (Averill et al., 2014). Transfer of increasing amounts of photosynthetically derived carbon to

ectomycorrhizal fungi and improved colonisation of mineral substrates during evolution of plants (Quirk et al., 2012; 2014) is consistent with the idea that weathering of silicate minerals and sequestration of C into ocean carbonates has led to drawdown of global $CO_2$ levels during the past 100 M years (Taylor et al., 2011). Enhanced weathering of minerals applied to different ecosystems has now been suggested as a global $CO_2$ reduction technology (Taylor et al., 2016; Beerling et al., 2018).

However, Smits and Wallander (2017) pointed out that there is no clear evidence that processes observed at the laboratory scale play a significant role in soil-scale mineral dissolution rates. Furthermore, many of the theories about evolutionary development of weathering have been elaborated in the absence of detailed molecular identification of the microorganisms involved. Detailed studies of the liquid chemistry of local weathering sites at the micrometre scale, together with up-scaling to soil-scale dissolution rates, are advocated, and the authors suggest that future research should focus on whole-ecosystem

dynamics, including the behaviour of soil organic matter, and that early-stage primary succession ecosystems on low reactive surfaces, such as fresh granites, should be included. Smits and Wallander (2017) also recommend the use of stable isotopes by choosing minerals and soils with distinct isotope ratios.

Most studies of ectomycorrhizal influence on weathering rates have been done over short periods in laboratory settings, and there is no clear evidence that processes observed at laboratory-scale play a significant role in soil-scale mineral dissolution



rates. In an attempt to span a longer time-scale for biological weathering studies, Smits and Wallander (2017) used a vegetation gradient from bare soil, via sparse grass to Norway spruce forest in a natural, lead-contaminated area in Norway. This gradient had probably been present since the last glaciation, and made it possible to study long-term effects of vegetation on apatite weathering in moraine material deposited at the end of the last glaciation. The presence of vegetation

had a strong stimulatory effect on apatite weathering, mainly because of the acidifying effect of plant growth, which was probably mediated through activity of the associated ectomycorrhizal fungi.

This effect of plant growth on weathering of apatite, through changes in soil pH, is probably captured sufficiently in weathering models like PROFILE/ForSAFE under situations when P is not in short supply and nitrogen is limiting tree growth, which is the general case in boreal forests. However, under conditions of low P availability, intensive colonisation of

apatite particles by EMF has been seen both in laboratory experiments (Rosling et al., 2004; Smits et al., 2012) and in the field (Bahr et al., 2015, Rosenstock et al., 2016, Almeida et al., in press). Weathering of apatite may be enhanced under these conditions through biomechanical mechanisms and accumulation of weathering agents in localised microenvironments that are colonised by EMF but separated from the soil solution as explained below.

The nutrient status of the forest is probably of great importance when estimating the role of biota in mineral weathering.

Belowground carbon allocation is usually increased under N and P limitation, but reduced under K and Mg limitation (Ericsson 1995). For this reason, weathering of K- and Mg-containing minerals may not be enhanced under K and Mg limitation, since root and mycorrhizal activity is expected to decline under these conditions. Support for this view was found by Rosenstock et al., (2016), who studied EMF colonisation of biotite and hornblende under varying K and Mg conditions in Norway spruce forests in the Czech Republic.

Laboratory studies of the capacity of different fungi to mobilise P and base cations from granite particles (conducted within QWARTS) (Fahad et al., 2016) suggest that some ectomycorrhizal fungi can mobilise and accumulate significantly higher concentrations of Mg, K and P than non-mycorrhizal fungi. The mycorrhizal fungi fractionate Mg, discriminating against heavier isotopes, and we found a highly significant inverse relationship between $\delta^{26}$Mg tissue signatures and mycelial concentration of Mg. This provides a theoretical framework for testing hypotheses about fungal weathering of minerals in

future experiments.

If active mobilisation and uptake of lighter $^{24}$Mg isotopes results in relative enrichment of heavy Mg isotopes left in soil solution and soil, this should be evident in areas of active weathering. Mesocosm experiments conducted within the QWARTS project (Mahmood et al., unpublished), employing a gradient of increasing organic matter depletion to simulate progressively more intense biomass harvesting, revealed significant and successive enrichment of $^{26}$Mg signatures in the soil

solution in the B horizon, associated with increased availability of organic matter and resultant increases in plant and fungal biomass. No such enrichment was found in other horizons or in systems without plants (and therefore without mycorrhizal fungi). This suggests that significant biological weathering of Mg takes place in the B horizon, driven by higher plant biomass that enables improved carbon allocation to the fungal mycelium and also constitutes a larger sink for uptake of mobilised base cations.





Although the experiments provide strong support for the idea of biologically driven mobilisation of Mg from B horizon mineral soil, the process was not sufficient to maintain tree growth in systems severely depleted of organic matter. Mycorrhizal fungi play a central role in mobilising N and P from organic substrates (Lindahl and Tunlid, 2015) and when these are depleted, N and P limit tree growth resulting in reduced C supply to the mycorrhizal mycelium and reduced capacity for mobilisation of base cations from the mineral horizons. Although mobilisation of Mg from the B horizon was sufficient to support increased biomass production in systems supplied with extra organic material, it was not sufficient to sustain plant growth when organic material was most depleted and insufficient N was available. The results of these experiments are therefore consistent with the predictions of modelling that, under intensive forestry with removal of organic residues, base cation supply will not be sustainable in the long term.

## 6 Implication of higher resolution of chemical reactions in weathering modelling

Aluminium (Al) and base cation concentrations are the primary weathering brakes in unsaturated soil (Warfvinge and Sverdrup, 1992). Higher concentrations of these elements have a negative effect on the dissolution rates of the minerals containing the elements (Sverdrup et al., in review, this issue)). It is therefore imperative to correctly simulate the concentrations of Al and base cations in the soil solution.

Different soil chemical models simulate the dynamics of inorganic Al and base cations in different ways. These can be classified into two categories: 1. Simpler ion-exchange equations (e.g. Gaines-Thomas or Gapon) that conceptualise sorption and desorption of $Al^{3+}$, $H^+$ and base cations as a series of ion-exchange reactions, and 2. More advanced 'state-of-the-art' organic complexation models such as WHAM, NICA-Donnan or SHM (Tipping, 2002; Kinniburgh et al., 1999; Gustafsson, 2001) that treat organic matter as the main cation sorbent, where proton dissociation over a wide pH range drives complexation and exchange of Al and base cations.

In general, the use of organic complexation models to simulate base cation and Al dynamics is strongly supported by empirical evidence (e.g. Tipping, 2002). For a long time, the former type of model approach has been more widely used in popular biogeochemical models such as MAGIC, PROFILE and ForSAFE. However, already in 1996, the CHUM model was introduced, which incorporates a version of WHAM (Tipping, 1996), and today SMARTml and HD-MINTEQ provide additional examples of (bio)geochemical codes that employ organic complexation models (Bonten et al., 2011; Löfgren et al., 2017).

Gustafsson et al. (2018, this issue) investigated the implications of using the two model approaches on the dynamics of Al, base cations and acidity. Overall, the two model approaches provided the same type of response to changes in input chemistry, implying that in many cases, there may be a rather limited benefit from using organic complexation models when calculating weathering rates. However, the exchange models tested stress the importance of including $H^+$ in the exchange dynamics to be able to account for the effects of rapid changes in proton concentrations, such as in the case of sea spray events (Gustafsson et al., 2018 (this issue)).



Although, for the most part, these results suggest that the current model setup in e.g. ForSAFE may be sufficient, certain differences remain between the two categories of models. The Gaines-Thomas and Gapon exchange equations produce a relatively stronger buffering of soil solution pH over a relatively narrow pH range. Together with the general oversimplification of the cation binding process this also causes the ion-exchange equations to overestimate the historical

levels of exchangeable base cations (Gustafsson et al., 2018 (this issue)). Consequently, it may be necessary to include organic complexation under such conditions as prolonged or substantial changes in acid input. Not explicitly simulating organic complexation may require additional coefficients that account for temporal changes in cation selectivity to correctly predict pH, base cations and Al, thereby entailing more uncertainty. Excluding organic complexation can bring into question the ability of biogeochemical models to predict the effect of large changes in acid input on weathering rates.

**7 Weathering in a sustainability perspective**

The effect on policy applications of the variability in weathering rates among different methods depends on the context in which they are to be used. A few decades ago, weathering rates were discussed in the context of acid rain and critical loads of acid deposition. Then the weathering rates were mainly compared with deposition, which at that time (1970-1980) was much greater than estimated weathering. Now, when the focus has shifted to sustainable forest management, the interesting

comparison is between base cation weathering rates and base cation losses through harvesting. The effect of uncertainties in this context is dependent on site properties and the size of other base cation flows.

Budget calculations, with weathering and deposition as inputs and harvest losses and losses through leaching as outputs, are often used in sustainability assessments (Akselsson et al., 2007; Hultberg et al., 2004). Simplified calculations, based only on weathering and harvest losses, are also often used (Olsson et al., 1993; Klaminder et al., 2011; Stendahl et al., 2013).

Weathering rates that are substantially higher than the harvest losses indicate less risk for depletion of base cations in soils than if weathering rates are lower than harvest losses.

Harvest losses were estimated and compared with weathering rates for the sites in Table 1 where weathering has been calculated for the root zone (defined here as <0.7 m) and for which data on site index could be found, in spruce forests (Fig. 5) and pine forest (Fig. 6). The calculations of harvest losses were based on the stand index of the forest and generalised

densities and nutrient base cation concentrations in different tree parts. Two types of harvesting were considered, conventional stem-only harvesting and whole-tree harvesting, where, in addition to the bole (stem), tops and branches are removed for biofuel. It was assumed that, in whole-tree harvesting, all stems and 60% of the branches were harvested, and that 75% of the needles were removed with the harvested branches. The methodology is more thoroughly described in Akselsson et al. (2007).

Although the difference in weathering rates between methods is large on several sites, a number of general conclusions can be drawn. For the stem-only harvesting scenario, the harvest losses were generally at the same level or lower than PROFILE




weathering rates. The depletion method gave generally higher weathering rates than harvest losses at stem-only harvesting in the northern sites, but lower in the southern sites.

In most of the spruce forests in southern and central Sweden (the sites to the right in Fig. 5), the harvest losses in the whole-tree harvesting scenario were substantially higher than the weathering rates, regardless of the method used to calculate

weathering rates. Exceptions were Asa, where the weathering rates from the mass balance calculations gave a very high weathering rate, higher than the harvest losses, and Gårdsjön, where the weathering rate for most of the methods was similar to the harvest losses. On some of the sites in these areas the variation in weathering rates between methods was large, for example in Bodafors and Skånes Värsjö, but both methods still led to the same conclusions regarding the sustainability of harvests.

On the four spruce sites in northern Sweden (to the left in Fig. 5), PROFILE gave higher weathering rates than harvest losses after whole-tree harvesting, whereas it was the other way around for the depletion method. The mass balance method in Flakaliden gave very high weathering rates, similar to those at Asa. However, according to the discussion above, these mass balance calculations do not give reliable estimates of weathering rates, since they most likely also include release from other base cation pools (Casetou-Gustafson et al., in review b (this issue)).

All pine sites where comparisons could be made in this study are situated in northern or central Sweden. For all three sites, the PROFILE weathering rates were substantially higher than the harvest losses, both for the stem-only and whole-tree harvesting scenario. Weathering rates calculated with the depletion method were similar to the harvest losses at whole-tree harvesting, except for Kloten where the weathering rates were lower (Fig. 6). The differences between the methods made it more difficult to draw conclusions, as for the spruce sites in northern Sweden. However, the results indicate that the effects

of whole-tree harvesting in pine forests in central and northern Sweden are substantially smaller than for spruce forests in southern and central Sweden.

In the above assessment the extent to which whole-tree harvesting itself affected the weathering rates was not explicitly considered. As an increased forest harvest intensity leads to slightly more acidic conditions, it could be hypothesised that increased intensity leads to an increased proton-promoted dissolution of minerals, thereby providing a feedback mechanism

in which increased weathering could partially alleviate the effect on soil acidity and base cation status. However, according to recent HD-MINTEQ modelling in which PROFILE was used to simulate weathering, the weathering rate was largely unaffected by soil solution pH and by the harvesting method used (McGivney et al., in review (this issue)). This was explained as being the net result of the opposing effects of pH and dissolved Al on the weathering rate. While a decreased pH itself leads to an increased weathering rate, it also leads to increased levels of dissolved Al, which is a potent weathering

'brake', offsetting the pH effect.

Despite the uncertainties, the results clearly indicate that whole-tree harvesting is not sustainable in the long term in spruce forests in southern and central Sweden, since the weathering rates are much lower than the base cation losses at harvest. In spruce forests in northern Sweden and in pine forests, long-term losses are less of a concern, although the uncertainties in



weathering rates make it difficult to say whether the weathering rates are higher or lower than the harvest losses in those forests.

When discussing base cation sustainability, it is important to not only focus on uncertainties in the actual soil weathering rates. Other important topics are how much of the weathered material the tree roots can reach, the size of the base cation deposition, and how the base cation losses are distributed between soil, biomass and runoff water. The base cation deposition in Sweden is assessed to be of similar size as the base cation weathering (Akselsson et al., 2007), but a national survey of total base cation deposition, including dry deposition, is not available, so uncertainties of base cation deposition are large. The rate of tree growth and leaching can decline after whole-tree harvesting, mitigating some of the impacts of harvest on soil base cation status (Zetterberg et al., 2013; Egnell, 2016). Currently there is even less research on weathering rates in relation to root depth, base cation deposition and alterations in base cation flows as pools change, than on weathering.

## 8 Future research

By far the most widely used, and most evaluated, method for estimating weathering rates for soils in Sweden is the PROFILE model. In this paper, as well as in Kronnäs et al. (in press, this issue), weathering rates from the dynamic model ForSAFE, which contains the same kinetic equations as PROFILE, have been compared with PROFILE results, leading to the conclusion that the two models produce similar weathering rates on average, as long as the hydrology input to PROFILE is similar to the modelled hydrology in ForSAFE (Kronnäs et al., in press, this issue). However, ForSAFE offers the opportunity to dynamically model weathering rates, and their variations within and between years, which is essential for sustainability assessment in a future with a changing climate and management intensity. A number of different research areas where further research is needed have been identified in the QWARTS programme, regarding development of those two models and reducing uncertainties in input data, as well as further evaluations and comparisons with other weathering estimates.

### 8.1 Model development: Biological weathering

Many fungal hyphae produce extracellular polysaccharides (EPS) at their hyphal tips, providing an interface that ensures intimate contact between the hyphae and mineral substrates. The contact area between hyphae and mineral surfaces is increased by EPS haloes (Gazze et al., 2013), and many fungal exudation products such as organic acids and siderophores may be released into polysaccharide matrices (Flemming et al., 2016) in close proximity to mineral surfaces. Here, they are effectively isolated from the bulk soil solution and may be protected from microbial decomposition by antibiotic compounds also produced by the fungi. This may increase the effective concentrations of organic weathering agents at sites of active weathering, and structure the bacterial communities associated with particular mycorrhizal fungi (Marupakula et al., 2016).



Bacteria associated with ectomycorrhizal fungi may have a significant influence on mobilisation of different nutrients (Calvaruso et al., 2013). Although weathering at the mineral surface is determined by chemical reactions (chemical weathering), the conditions and agents involved in these reactions are often derived from biological activity and may be defined as biological weathering.

Existing models that explicitly simulate mineral weathering use soil solution chemistry to determine the weathering rates (Erlandsson et al., 2016). It remains challenging to consider the EPS microenvironments described above in models, as there is a lack of knowledge of the processes determining the former. This difficulty starts in the empirical description of the difference between bulk soil solution and micropore EPS chemistry. Extraction methods using high speed centrifugation may remove EPS from micropores and hyphal interfaces but the resulting bulk concentrations of weathering agents will not

reflect those at active sites of weathering. The implication of excluding the said difference between EPS micropore chemistry and soil solution chemistry on weathering rates remains unclear.

Active uptake of weathering products by fungal hyphae, followed by translocation towards the plant root, will prevent their accumulation at sites of weathering. Mineral elements mobilised by fungal hyphae may remain within the fungal mycelium for different lengths of time before becoming available for plant uptake, and this may represent an important pool of base

cations to be included in models. Currently, the active uptake process in the PROFILE/ForSAFE models does not distinguish between roots and mycorrhizae, treating both as a lumped uptake organ. Since both minerals and organic residues contain ectomycorrhizal fungal cycle nutrients, it is imperative that better methods are developed to distinguish between these two sources.

The stable isotope fractionation patterns of ectomycorrhizal fungi, shown by Fahad et al. (2016) to involve discrimination

against heavier isotopes of Mg, provide a useful tool for use in future studies. They can be applied in field situations but further information about isotope fractionation patterns in organic and inorganic substrates is needed, since it is important to distinguish between the de novo supply of elements supplied via weathering and re-circulation of elements via decomposition of organic residues by both mycorrhizal and saprotrophic fungi

### 8.2 Model development: Higher resolution chemical reactions

The comparisons performed in QWARTS indicated that adding a higher resolution description of aluminium complexation and cation exchange reactions to ForSAFE generally led to small effects on long-term chemical dynamics of Al and base cations, which induces small effects on modelled weathering rates. The strongest effect was seen when replacing the ion-exchange equations to describe base cation dynamics with an organic complexation model such as SHM, whereas the replacement of the gibbsite model with more sophisticated model descriptions for Al mattered less (Gustafsson et al., 2018

(this issue)). The effects were rather small, except when large pH fluctuations occur in the data, caused by large changes in acid input. It was concluded that other factors such as uncertainties in deposition and uptake values, as well as the calibration procedure, are likely to be of larger importance for the model performance (Gustafsson et al., 2018 (this issue)).




Due to the significant reduction in execution speed caused by the introduction of organic complexation models, this modification will currently be less prioritised for inclusion in PROFILE/ForSAFE. However, we note that such a modification would be desirable concerning (i) long-term simulations over hundreds of years when large changes occur in the chemical drivers, and (ii) sites experiencing frequent or strong sea salt episodes causing large changes in the chemical composition of the influent water. HD-MINTEQ will be developed further as a scenario tool with a relatively long time steps (weekly).

### 8.3 Model development: Implementing weathering brakes

According to the transition state theory, mineral dissolution rates in PROFILE/ForSAFE are retarded by elevated soil solution concentrations of weathering products, as the equilibrium between the solid and aqueous phases is approached. In the unsaturated zone, weathering retardation is mainly caused by elevated concentrations of base cations and aluminium, called weathering brakes. Based on this assumption, PROFILE/ForSAFE produce reasonable weathering rates in the unsaturated rooting zone (Sverdrup and Warfvinge, 1993; Erlandsson et al., 2016). However, moving into the saturated zone, the strength of the usual weathering brakes fails to slow down the mineral dissolution, which leads to grossly overestimated estimates of weathering rates (Stendahl et al., 2013; Erlandsson et al., in review (this issue)). In this environment, soil solution silicate concentrations play a central role in hindering mineral dissolution (Sverdrup et al., in review (this issue)). For this reason, the kinetics of silicate release from mineral dissolution has been added to the traditional elements, as well as the dynamics of silicate concentrations in the soil solution. Erlandsson et al. (in review, this issue) tested a prototype of this addition, and the results proved promising in keeping weathering rates within observation levels in the saturated zone, but this is yet to be implanted and tested in PROFILE/ForSAFE.

### 8.4 Model development: Weathering below the root zone – for surface water quality assessments

Water residence times in the hillslope, and the proportion of old water generating stream flow, need to be more accurately characterised, since this fraction influences delivery of weathering products from within the catchment to the stream (Bishop et al., 2004). This older water has higher concentrations of weathering products. It is not sufficient to predict the rate of weathering within a catchment; the spatial distribution of that weathering in relation to catchment flow pathways and water residence times must also be quantified (Erlandsson et al., in review (this issue)). The possibility that the older water will never even reach the headwater streams most sensitive to acidification, but will appear further downstream in a larger catchment as groundwater subsidy (Ameli et al., 2018), needs to be examined.

Lateral flow has recently been included in ForSAFE, and a new version, ForSAFE-2D, has been developed (Zanchi et al., 2016). The model has been evaluated on the basis of hydrological flows and chloride concentrations and transport, with good results. Evaluating the modelled base cation concentrations in surface water highlighted the need for adjusting the weathering brakes (see discussion above about silicate brakes), and also a need to revisit the decomposition process descriptions, thereby validating them for the saturated zone. Further development of ForSAFE-2D has the potential to





provide a mechanistic tool for assessing weathering rates also for surface water applications. The importance of correctly defining the flow pathways and residence times for the delivery of weathering products to the surface waters, and the potential value of concentration-dischare relationships for calibrating biogeochemical models was explored by Ameli et al. (2017).

## 8.5 Reducing uncertainties in model input data

Mineralogy, specific surface area and soil moisture are of key importance in weathering modelling, but are often burdened with high uncertainties. To reduce input data uncertainties, a focus should be placed on those three parameters.

In the A2M model, mineralogy is often estimated based on total chemistry, since direct mineralogy measurements are not available on most sites. To accurately estimate a probable mineralogy, not only good soil chemistry measurements are required, but also information about which minerals can be expected in the soil. In Sweden, three different geographical mineralogy regions have been used since the 1990s to assign qualitative mineralogy to a site (Warfvinge and Sverdrup, 1995).

Casetou-Gustafson et al. (in review a, this issue) compared weathering rates calculated based on three sets of mineralogies: one based on direct measurements of quantitative mineralogy, one based on normative modelling with A2M using direct measurements of qualitative mineralogy, and one based on normative modelling with A2M using data from the regions mentioned above. The results gave weathering rates for both normative methods that were close to the weathering rates based on directly measured mineralogy. It could not be concluded that the normative mineralogy based on the regions gave worse results. Although these results strengthen the credibility for the normative mineralogy regions, Casetou-Gustafson et al. (in review a, this issue) recommend continued work to reduce uncertainties related to mineralogy, mainly by revisiting and, if appropriate, updating mineral rate coefficients. More comparisons of weathering rates from normative mineralogies based on generalised and site-specific quantitative mineralogy are needed, to adequately assess whether the regional divisions need to be revised and refined in order to further reduce the uncertainties in the mineralogy estimates.

A2M gives as output a multidimensional space of solutions, all of which have the same probability. Often, the centre point of the space is used for weathering calculations. However, the span can be quite broad, which leads to uncertainties in the calculated weathering rates (Casetou-Gustafson et al., in review a (this issue)). Future research focusing on constraints that could help to narrow the space of possible solutions that A2M creates, e.g. based on the grain size distribution, could reduce those uncertainties.

Minerals are assumed to be evenly distributed among grain sizes in PROFILE and ForSAFE. The effect of this assumption has not been fully analysed. The most obvious example showing that minerals are not evenly distributed among grain sizes is clay minerals, which are found in the clay fraction. The extremely high surface area of clays leads to very high base cation weathering rates when the clay fraction is high, although the content of base cations is low. Due to this, Phelan et al. (2014) introduced a correction factor. A thorough analysis of all grain size fractions can help to further refine these methods.



The surface area of soils is often calculated with regressions based on old BET measurements (Warfvinge and Sverdrup, 1995). The uncertainties could most likely be reduced through revisions of the regressions, using modern technology.

The soil moisture is one of the most important factors in weathering modelling that introduces large uncertainties in the results, both in PROFILE where it is an input (Rapp and Bishop, 2003), in ForSAFE where it is modelled based on hydrological parameters. Kronnäs et al. (in press, this issue) applied both PROFILE and ForSAFE on two sites, and used a rough assessment of soil moisture as input data for PROFILE, whereas soil moisture was modelled by ForSAFE based on soil properties and precipitation. In both those cases, the modelled soil moisture was relatively close to the rough assessment of soil moisture, but the difference was bigger for one of the sites, which could also be seen in the difference in modelled weathering rates. Improved input data quality for soil moisture would substantially reduce uncertainties in PROFILE and, even more importantly, soil moisture modelled by ForSAFE needs to be evaluated, and the sensitivity to soil input data needs to be examined.

## 8.6 Comparison between modelled weathering and other estimates of weathering

Next to the PROFILE model, the depletion method is the most used method in Sweden, often in combination with the total analysis regression method. This method is relatively easy to perform on new sites, although detailed data of the soil profile is needed. As for PROFILE, the method requires soil sampling and total elemental analysis of the soil. In an undisturbed soil profile, if it can be assumed that most of the soil was developed after the last glaciation as well as that zirconium does not weather, the depletion method should give an accurate measure of the average weathering since the last glaciation.

To further evaluate the accuracy of results from the depletion method, as a proxy for the weathering rates of today, the reliability of the assumptions needs to be further evaluated, and the relationship between the average weathering rate since the last glaciation and today's weathering rate needs to be assessed. The latter can be done by performing ForSAFE modelling on a site where the depletion method has been applied. A similar exercise has been done with the SAFE model (Warfvinge et al., 1995), but the inclusion of tree growth and decomposition in ForSAFE can be expected to improve the results. Furthermore, a manual for the depletion method needs to be developed, including requirements that must be fulfilled for soil profiles to be regarded as undisturbed. Standardised methods for setting the weathering depth based on the elemental content curve in the soil profile would enable objective and comparable estimates.

The budget method requires more measurements than the depletion method. Different applications of the budget method handle the distinction between sources of base cations in the soil in different ways, as explained above. Also, the base cation deposition contains large uncertainties. In the compilation of weathering rates in this paper, the most extreme outliers came from the budget method, which can be explained by the fact that other sources than weathering are included (Rosenstock et al., in review (this issue)). For a fair comparison between weathering rates from the budget method and from other methods, ways to distinguish between different sources need to be further developed.

An advantage of the budget method using the Sr isotope ratio is that it can distinguish between weathering and release from the exchangeable pool. It has been applied on three sites in the research performed in the 1990s. The results were on the


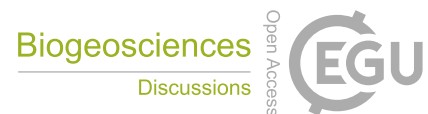

same level as those from PROFILE, and also from the depletion method and the total analysis regressions for the sites where those methods were used. It requires deposition and leaching measurements (as in all budget calculations), which may be the reason why it has not been used more. The few comparisons made in this study show promising results, and we therefore encourage estimates on more sites to enable evaluation of the budget method based on the Sr isotope ratio.

In the MAGIC model, the release from the exchangeable pool is thoroughly modelled, but for other sources of base cations, the same problems apply as for the budget method, i.e. uncertainties in base cation deposition. These uncertainties exacerbate the uncertainties in weathering rates that derive from the mass balances in MAGIC. Nevertheless, MAGIC theoretically provides a good basis for conducting independent weathering rate assessments. On sites with relatively small input data uncertainties, our recommendation is to carry out such comparisons.

Futter et al. (2012) recommended that at least three independent methods be used to quantify weathering rates on a site for sustainability assessments. They also emphasised the importance of similar assumptions in such comparisons, most importantly calculating weathering rates to the same soil depth. In the review of weathering studies in this paper, we found only five locations where at least three methods had been implemented and where the criterion of the same depth is fulfilled. In four of those sites, budget calculation was one of the methods, and on three of those sites, the budget calculations gave

unreliably high weathering rates. The explanation is that weathering cannot be distinguished from base cation exchange and release from other sources. Consequently, the recommendation to always use three independent methods seems unrealistic in practice, and the reliability of the different methods needs to be considered in the comparison.

## 9 Conclusions

Uncertainties in weathering rates have often been presented as an obstacle in the assessment of sustainable forestry. The

comparison between approaches in this paper, on a regional level as well as on a site level, suggests that both weathering rate gradients and approximate weathering rate levels can be captured with available methods. Although the variation in weathering rates between methods was large on single sites, most of the sites could be grouped into broader classes representing very low, low and intermediate weathering rates, which can be used for general, but not specific, weathering rate assessments at site level. The more and better input data that is available, and the more methods that are applied and

compared for a single site, the more robust overall assessments can be done at site level, providing the conceptional differences, boundary conditions and assumptions between methods are kept in mind.

Based on the results from this study, we argue that modelled weathering rates can be used for sustainability assessments, as long as the uncertainties, i.e. the intervals on single sites presented in this paper, are recognised. The ability to draw conclusions about sustainable forestry at site level depends not only on uncertainties in weathering rates, but also on other

site properties, relating to forest properties and other base cation flows, such as base cation deposition, and the associated uncertainties. Irrespective of the uncertainties related to the sustainability assessments, a robust conclusion was that weathering rates in spruce forests in southern and central Sweden generally were substantially lower than the harvest losses

at whole-tree harvesting, indicating that whole-tree harvesting without nutrient compensation is not sustainable in these areas. However, there is less risk of negative effect for spruce forests in northern Sweden, as well as pine forests in central and northern Sweden.

The research performed in the five years of the QWARTS programme supports the continued use of the PROFILE/ForSAFE models. ForSAFE is the only method that gives time-resolved results, so is the only method that can be used to study dynamic effects of changing climate and changing management methods. Although there is still scope for improving process understanding and incorporation of that understanding into PROFILE and ForSAFE, e.g. regarding weathering brakes and biological weathering, the most important way to reduce uncertainties in modelled weathering rates is to reduce input data uncertainties, mainly regarding soil texture and associated hydrological parameters. However, it is also important to continue evaluating and comparing with other approaches.

**Data availability**

Weathering data presented in this synthesis paper is compiled from other studies, which are published in other papers and refered to in the paper.

**Author contributions**

C. Akselsson planned and led the work, performed most of the calculations and wrote most parts of the paper. K. Bishop and S. Belyazid were highly involved in the planning and writing of the paper from start. S. Belyazid particularly contributed to the parts about modelling, including the chapters about biological weathering and the implications of higher resolution chemical reactions. J. Stendahl mainly contributed to parts about the depletion method and the total analysis regression, including the recalculation of weathering rates on a national scale using those methods. R. Finlay was the main author of the chapters about biological weathering, which also H. Wallander and S. Belyazid substantially contributed to. B. Olsson contributed to the methods descriptions, results and discussions concerning the budget method. J-P. Gustafsson wrote about the implications of higher resolution chemical reactions together with S. Belyazid and contributed to other parts in the paper where the chemistry in the weathering models was discussed. M. Erlandssons main contributions concerned the modelling parts and the parts about weathering brakes.

**Competing interests**

The authors declare that they have no conflict of interest.





## Acknowledgements

This study was funded by the Swedish Research Council Formas (reg. no. 2011-1691) within the strong research environment "Quantifying weathering rates for sustainable forestry (QWARTS)".

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





**Table 1: Weathering rates (meq m$^{-2}$ y$^{-1}$) at sites where at least two approaches for estimating weathering rate have been applied to the same depth.**

| Site | Depth | References | PROFILE | Depletion | Budget | Sr | Tot. anal. regr. | MAGIC | ForSAFE |
|---|---|---|---|---|---|---|---|---|---|
| Gårdsjön A | 0.5 m[b] | Stendahl et al., 2013 | 52 | 41 | - | - | - | - | - |
| Gårdsjön B | 0.67 m[a,c] | Sverdrup et al., 1998; Köhler et al., 2011 | 57 | 53 | 54 | - | 44-53 | 62 | - |
| Gårdsjön C | 0.47 m[a] | Sverdrup et al., 1998 | 37 | - | 36 | 39 | 38-42 | - | - |
| Svartberget A | 0.5 m[b] | Stendahl et al., 2013 | 38 | 17 | - | - | - | - | - |
| Svartberget B | 0.8 m[a] | Sverdrup et al., 1991; Sverdrup and Warfvinge,1993; Lundström, 1990 | 42 | 31 | 85 | 35 | - | - | - |
| Vindeln | 0.5 m[b] | Stendahl et al., 2013 | 30 | 13 | - | - | - | - | - |
| Risfallet A | 0.5 m[b] | Stendahl et al., 2013 | 68 | 29 | - | - | - | - | - |
| Risfallet B | 1.0 m[b] | Sverdrup et al., 1991; Sverdrup and Warfvinge, 1993; Jönsson et al., 1995; Maxe, 1995 | 29 | - | - | 25 | - | - | - |
| Fårahall | 1.0 m[b] | Sverdrup et al., 1991; Sverdrup and Warfvinge, 1993; Jönsson et al., 1995; Maxe, 1995 | 60 | 60 | - | - | - | - | - |
| Stubbetorp | 1.0 m[a] | Maxe, 1995 | 67 | - | - | - | 35-51 | 30-40 | - |
| Flakaliden | 0.5 m[b] | Casetou-Gustafson et al., 2019b | 43 | 22 | 69 | - | - | - | - |
| Asa | 0.5 m[b] | Casetou-Gustafson et al., 2019b | 37 | 11 | 137 | - | - | - | - |
| Bodafors | 0.5 m[b] | Stendahl et al., 2013 | 41 | 22 | - | - | - | - | - |
| Hjärtasjö | 0.5 m[b] | Stendahl et al., 2013 | 29 | 20 | - | - | - | - | - |
| Hässlen | 0.5 m[b] | Stendahl et al., 2013 | 52 | 18 | - | - | - | - | - |
| Kloten | 0.5 m[b] | Stendahl et al., 2013 | 42 | 11 | - | - | - | - | - |
| Kullarna | 0.5 m[b] | Stendahl et al., 2013 | 36 | 16 | - | - | - | - | - |
| Lammhult | 0.5 m[b] | Stendahl et al., 2013 | 35 | 33 | - | - | - | - | - |
| Skånes Värsjö | 0.5 m[b] | Stendahl et al., 2013 | 37 | 8 | - | - | - | - | - |
| Stöde | 0.5 m[b] | Stendahl et al., 2013 | 41 | 18 | - | - | - | - | - |
| Söderåsen | 0.5 m[b] | Stendahl et al., 2013 | 36 | 19 | - | - | - | - | - |
| Västra Torup | 0.5 m[b] | Kronnäs et al., 2019 | 58 | - | - | - | - | - | 63 |
| Hissmossa | 0.5 m[b] | Kronnäs et al., 2019 | 25 | - | - | - | - | - | 21 |

[a] Including O-layer

[b] Not including O-layer

[c] For MAGIC the weathering rate was calculated to 0.6 m, including O layer



**Table 2: Classification of the sites with soil depth of approximately 0.5 m, in four classes: Very low, low and intermediate weathering rates and a group with non-conclusive results, i.e. that did not fit into any of the other groups. Sites were placed in one of the three weathering groups if the median fell within the main interval given, and if the maximum and minimum values fell within the extended interval (±5). A and C refer to different profiles, with different soil depths.**

| | | Very low weathering rates | Low weathering rates | Intermediate weathering rates | Non-conclusive results |
|---|---|---|---|---|---|
| **Interval** | | 10-37.5 [a] (±5)[a] meq m$^{-2}$ yr$^{-1}$ | 37.5-60[b] (±5) meq m$^{-2}$ yr$^{-1}$ | ≥60[c] meq m$^{-2}$ yr$^{-1}$ | |
| Sites | | Vindeln | Gårdsjön A (0.5 m) | Fårahall | Hässlen |
| | | Hjärtasjö | Gårdsjön C (0.47 m) | | Risfallet A |
| | | Söderåsen | Västra Torup | | Asa |
| | | Stöde | | | Flakaliden |
| | | Kullarna | | | |
| | | Svartberget A | | | |
| | | Bodafors | | | |
| | | Lammhult | | | |
| | | Skånes Värsjö | | | |
| | | Kloten | | | |
| | | Hissmossa | | | |

[a] Corresponding to the lowest span in Fig. 2, 'acidic/intermediate parent material, coarse-textured'.

[b] Corresponding to the lower part of the two spans 'acidic/intermediate parent material, medium-textured' and "basic parent material, all grain sizes" in Fig. 2.

[c] Corresponding to the upper part of the span 'acidic/intermediate parent material, medium-textured' and the lower-

10    intermediate part of the span 'basic parent material, all grain sizes' in Fig. 2.





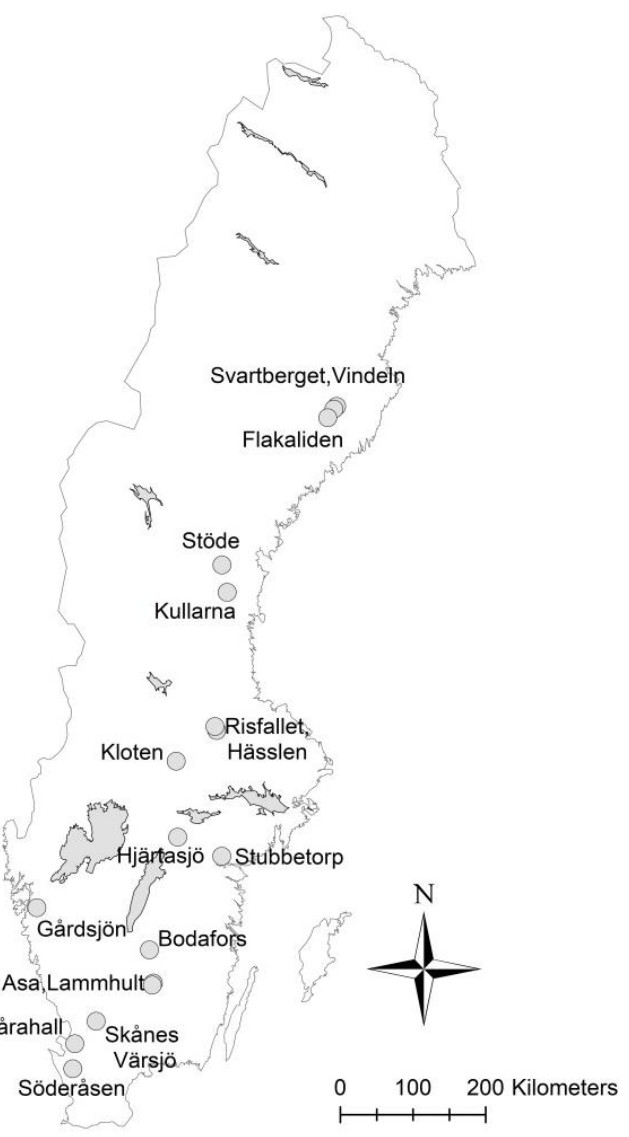

**Figure 1: Sites where weathering rates have been calculated for the same soil depth with at least two different approaches. See also Table 1 and Fig. 2.**





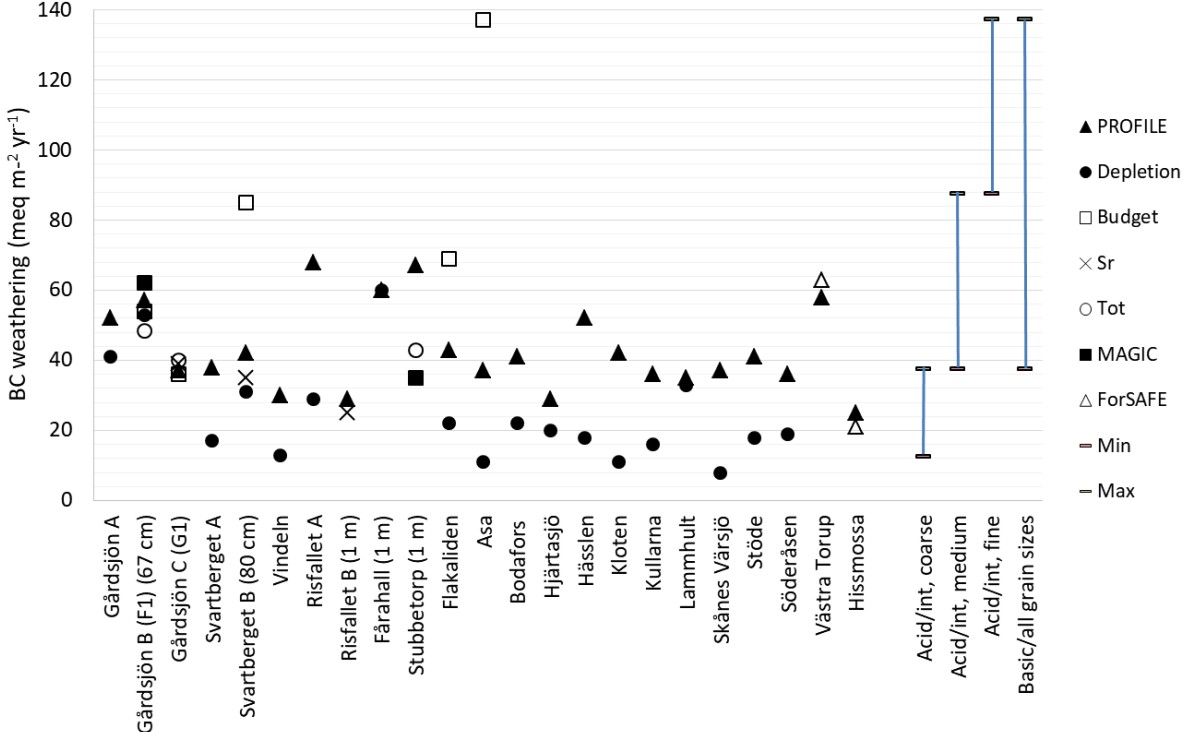

**Figure 2: Base cation weathering rates (sum of Ca, Mg, Na and K) for sites where different methods have been applied for the same depth on the same site (Fig. 1). The soil depth is around 0.50 m (with or without organic layer, see Table 1), except for a few cases where greater depths are given. The four spans to the right are intervals that were commonly used in the critical load work of CCE (Coordination Centre of Effects) within the UNECE Convention on Long-range Transboundary Air Pollution (LRTAP Convention) (de Vries, 1994; Umweltsbundesamt, 1996). The intervals correspond to weathering rates for different parent material classes: acidic, intermediate and mafic, and different texture classes: coarse, medium (including the mix between medium and course material) and fine (including the mix between fine and medium material).**



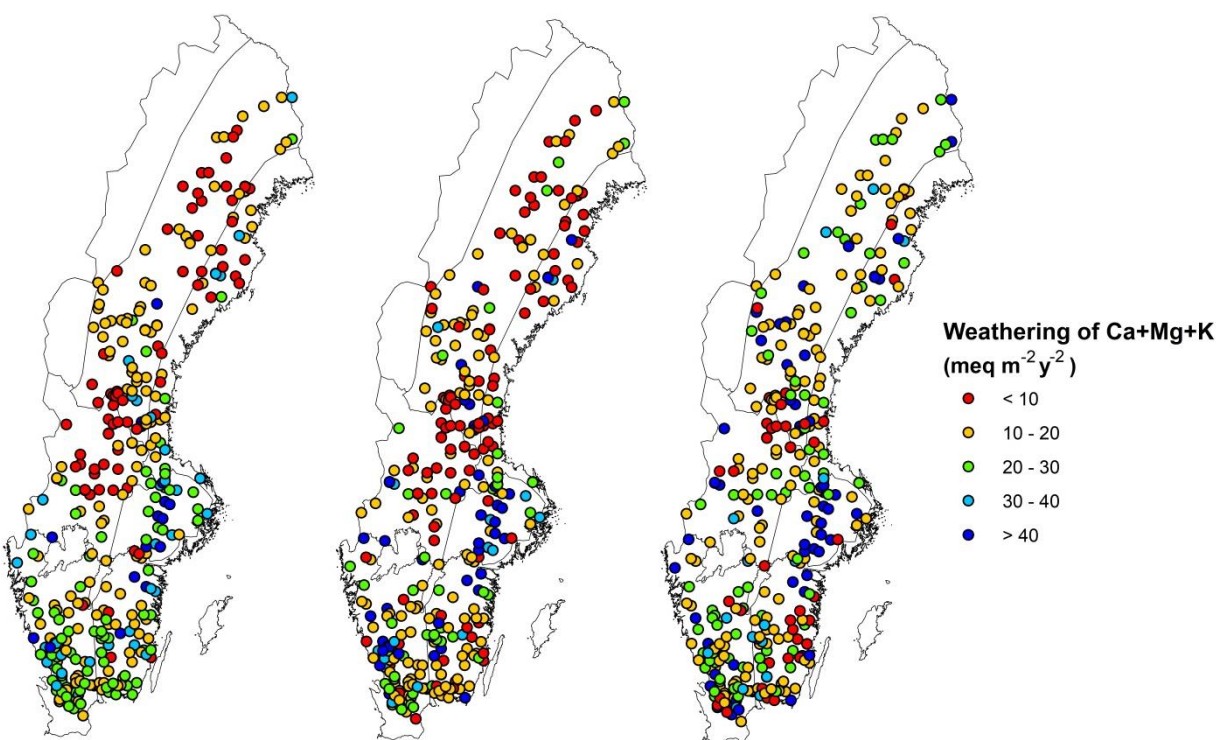

**Figure 3: Weathering calculated with the depletion method (left) modelled with PROFILE (centre) and with ForSAFE (right), in seven climate regions in Sweden, delimited by black lines.**



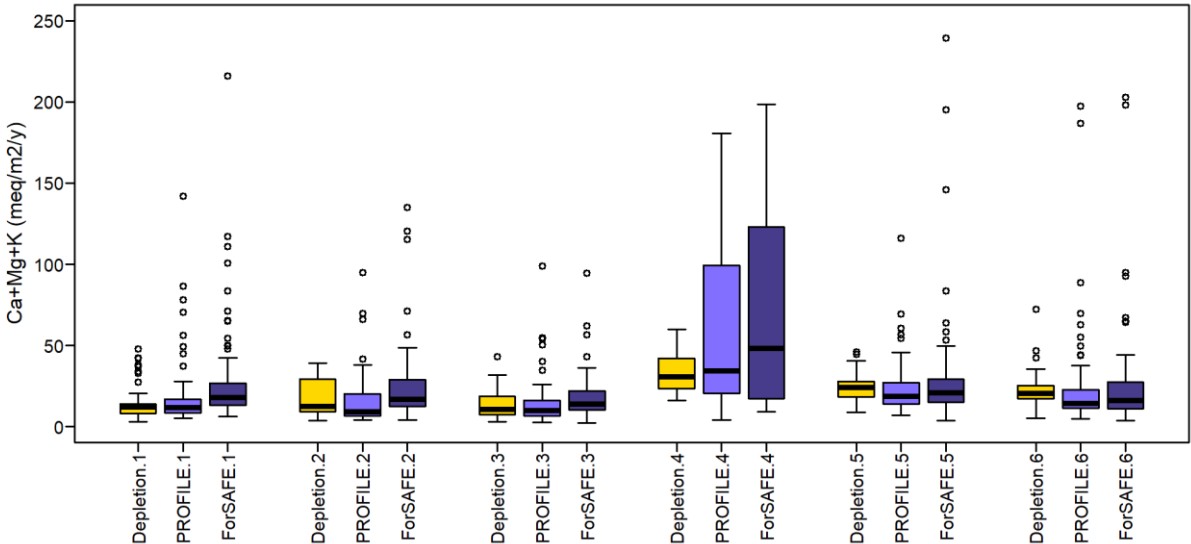

**Figure 4: Box plots for the three methods and for the climate regions: (1) Inner part of northern Sweden, (2) Coastal part of northern Sweden, (3) Western part of central Sweden, (4) Eastern part of central Sweden, (5) Southwestern Sweden, and (6) Southeastern Sweden. The Northwestern mountain region was excluded since it only contained one site.**



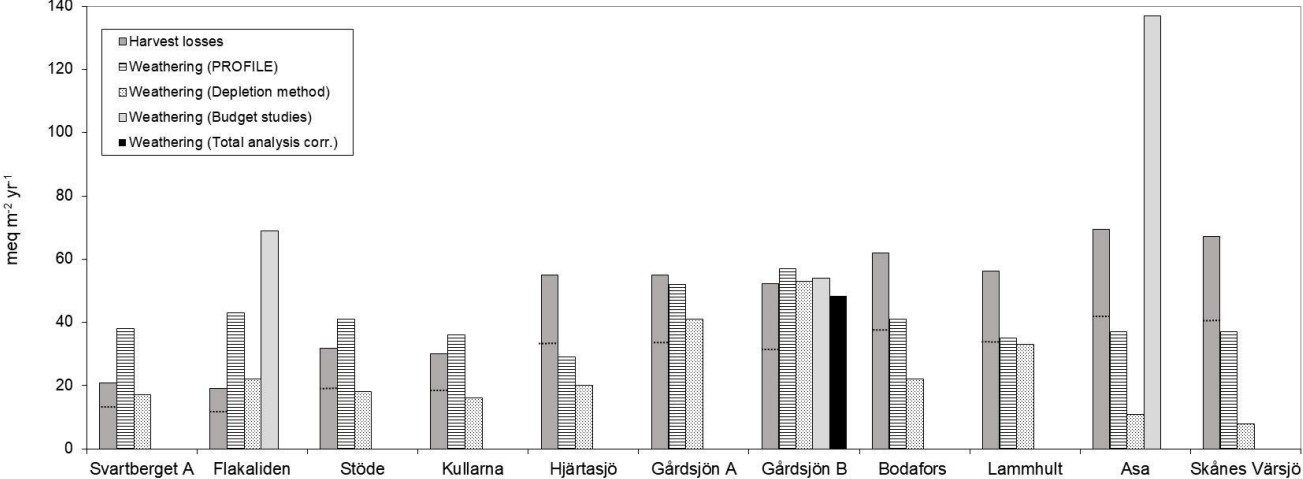

**Figure Figure 5: Weathering rates of base cations calculated with different methods on spruce sites, compared with harvest losses of base cations at whole-tree harvesting (60% of the branches harvested, 75% of the needles on the branches removed). The horizontal dashed lines in the harvesting bars show the levels for stem-only harvesting. The sites are ordered from north to south.**

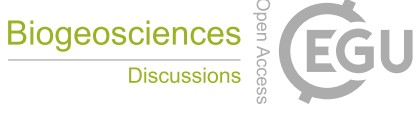

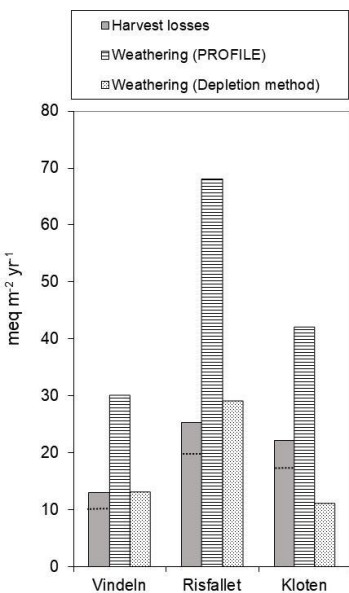

**Figure 6: Weathering rates of base cations calculated with different methods on pine sites, compared with harvest losses of base cations at whole-tree harvesting (60% of the branches harvested, 75% of the needles on the branches removed). The horizontal dashed lines in the harvesting bars show the levels for stem-only harvesting. The sites are ordered from north to south.**