# Peer review of "Weathering rates in Swedish forest soils"

_Biogeosciences, 2019_

## Referee Comment (RC1) · Anonymous Referee #1 · 19 Mar 2019

The authors present an overview or synthesis of base cation weathering studies carried out under the Swedish QWARTS project. The paper is clearly part of a special issue, as much of the text refers to other papers in 'this issue'. Given the dependence on other papers, it is not a standalone paper but a very 'Swedish' view of soil weathering. Nonetheless, the paper has an important objective, to demonstrate that despite the variation in estimated soil base cation weathering rates at the site level, there is general agreement and these data can be used to support the assessment of sustainable forestry. However, the paper falls down in several areas: (1) the text is overly long, at times there is extensive repetition within and between sections, (2) the text had a tendency to loose focus, the manuscript jumps between project summary, scientific review, and comparison of specific results, and while the authors forewarn of the contents in the abstract, the conclusion more succinctly speaks to the true contribution of the paper, if there are other papers in this special issue, do the authors need to

be so broad in their coverage?, (3) section names and section contents are confusing, the section on future research seems to focus on limitations, while repeating text from previous sections, and generally has the feeling that much of the text could have been integrated into previous sections, and (4) unfortunately, much of the comparison between weathering estimates is too qualitative, there is no quantitative assessment, statements such as 'they agree', 'do not agree' or 'estimates are similar' need quantitative support. I suggest the authors (a) step back from their manuscript and try to pinpoint their exact (unique) contribution, (b) they should remove repetitious text, and remove text that is described (reviewed) elsewhere in the special issue, and (c) add a stronger quantitative element to their comparison / assessment of weathering / sustainable forestry. I have no reservation in supporting the publication of the manuscript, however, I believe it currently needs major revisions before being accepted. However, much of these revisions only require reorganising, restructuring and rethinking. I have provided a number of specific comments below. Please excuse typographical errors.

Specific comments by Page and Line (L) number. Page 2 L1. It was internationally recognised during the 1970s but regionally recognised long before that... 1 to 2 decades! L2. one could argue that the peak was a little later... 1980s to 1990s? L4. Reword / clarify 'more harvest', more correctly you are referring to the use of forest residues for renewables! L7. lab –> laboratory L7. There was no intensive modelling? Perhaps 'extensive' is superfluous? L8. Simplify (here and throughout): 'This paper presents the state...' L9. You jump too quickly into the specific of the results, give the reader a more guided introduction, 'Under the project, we found that... ' L10. Variation from what? Data? Methods? Remember the international audience knows nothing of the project! L12. Important but the manuscript would greatly benefit from the 'word smiting' of the native English-speaking co-authors, Finlay and Bishop? L13. I think this is an important result but the term 'clear imbalances' obscures the implications of the findings. The activities are unstainable. L16. Step back and provide greater support... approaches based on the weathering of (observed) mineralogy, such as PROFILE..., provide the most important fundamental understanding of the contribution of weathering to long-term availability of base cations to support forest growth, nonetheless, these approaches should be continually assessed against...' L19. this point needs further development / clarity Page 3. L1. change acid to acidic throughout L3. remove one 'processes' L5. refer to SWAP first, it started before NAPAP (and is more important in a European context) L7. You need to provide more context for critical loads; it is an effects-based approach for emissions reductions, essentially a direct response to the recognition that emissions of sulphur dioxide were causing significant impacts. Notably it has nothing to do with SWAP or NAPAP! L9. 'A critical load...' L11. '... critical loads of acidity...' L15. Yes, very true but those of us interested in water barely consider weathering directly...? L16. '... and as such a sink of acidity'. L19. You need to differentiate between plot and catchment scale estimates, models such as MAGIC are process-based, and can be used to provide catchment-based estimates of weathering, however, they are fundamentally different to process-based estimates from PROFILE. L24. I do not completely agree with this. Many jurisdictions were faced with national scale modelling, and the application of simple approaches such as 'skokloster' provide a practical solution compared to the application of a process-based model that requires quantitative mineralogy on a high spatial resolution... more correctly, given the high loads at that time, the uncertainty in weathering was trivial. L29. Was it severity, or a shift in policy to support mitigation of climate change impacts? Page 4. L1. increase from 25L3. Substantial L5. Depletion methods needs more description... or just exclude such detail for the moment (estimates of base cation weathering...) L11. 'Akselsson et al. (2007) used a mass balance approach (with weathering estimated using PROFILE) ...' L17. I suggest 'Similarly, the influence of whole-tree harvesting ...' L20. Yes, but only in a Scandinavian context, this has not spilled over into the rest of Europe or north America (yet). L23. Was the conclusion valid? I would suggest the greatest uncertainty was derived from comparing approaches that should not have been compared? L30. It is okay to call out errors. Three approaches? In truth there are two. Mass balance approaches where you indirectly estimate weathering rate (there are also other indirect methods) OR mineralogy-based approaches, often if mineralogy

is not available you have surrogate-based approaches but 'at least three approaches' may verge on ridiculous? Weathering is the breakdown of minerals... so what does three independent approaches refer to? Page 5. L1. Provide background on the depletion method... total analysis regression... the reader need help. L3-4. reference to other methods are difficult to navigate... L7-8. Combine sentences... reduce words... L17. '... by revisiting older w...' L21. Flows or removals? L30. Replace flows with 'sources and sinks' Page 6. L2. Cite 'Warfvinge and Sverdrup, 1992' for PROFILE L2. The work of Susan Brantley should be cited here L7. The key point here, and what separates PROFILE from other approaches, it that weathering is derived from the breakdown of mineralogy (an essential input), the other inputs only estimate the amount of minerals that are being weathered. L13. Again, it might be worth citing Brantley here... L17. hydrological model... L21. The discussion / details on SAFE and ForSAFE can be removed. L30. Simplify to PROFILE Page 7 L4. There is an application of SAFE to Hubbard Brook which models the catchment (compared with MAGIC, VSD, etc.) L7. Should MAGIC be cited here or under 'mass balance' approaches? PnET-BGC is another example of a model that uses a mass-balance approach L17. Assumes that deeper soil is the parent material, so does not work for glaciofluvial soils, etc. L26. Could add that the approach has been widely used and cite a few examples? L27. Typically referred to as 'Catchment mass balance budgets' as they are widely estimated at the catchment scale, as such the estimates of weathering are an average of a larger landscape unit and can be highly influenced by localised geology. L28. MAGIC should really be mentioned in this section! Page 8 L10. Retitle to 'strontium isotope ratio' L17. This really should be included under the depletion, as it is a derivative of that approach L24. Remind the reader that you are focused on QWARTZ, i.e., 'Under QWARTZ, weathering ...' L30. 'profile 17-20cm deeper', this is a little unclear. I assume in simple terms the soil depth differs between estimates... Maybe present table on a 'weathering per cm'? Page 9 L2. This is an important point and should perhaps be stated much earlier, homogeneity of soil and bedrock are important considerations for agreement / lack of agreement between approaches L15. Why does the

depletion indicate a lower weathering rate? This could suggest that the un-weathered layer did have weathering? In many of the studies in Table 1, the depletion method is lower. Why? However, it may also be argued that the range between methods is smaller than the uncertainty? L21. Do you mean 'soil bulk density'? L23. Correct 'to to' Page 10 L19. Disqualified? Excluded! However, just exclude, and note in a footnote to the Table. No need to explain L30. Till is the most obvious / likely explanation... remove other excludes / reasons and only present this one L34. What does 'conceptual limitations' mean? clarify Page 11 L1. 'for Ca and Mg...' L3 to L10. This can be reduced to one sentence, state 'The majority of weathering rate estimates were classified as acidic or intermediate (cite Table, UNECE, etc.). L4. Figure 2 essentially repeats Table 2; the classification could be added to the Table 2 instead. L10 to L14. This text repeats detail already discussed. L20. This is a long section, and rather than go through each site (one by one), it would be more efficient to summarise and focus on the broad agreement, and disagree, but describe from the point-of-view of the factors that drive disagreement, e.g. ' ... there was slight disagreement between some estimates owing to difference in input data use by the different approaches, such as soil depth (give example) or soil moisture (give example). The table is provided, so the reader can evaluate the results, and there is no need to describe in detail. L22. are they scaled-up or just regional applications (more sites)? How are they scaled? L25 to L33. The relationship between ForSAFE and PROFILE has already been described at length. There is no need to repeat again. Page 12 L7. Is the analysis really only based on 11 sites? Did the study use 11 sites to predict at 400+? Page 13 L5. Is this the same approach as used with PROFILE? Did both use UPPSALA? Clarify L14. Again, is this similar to MATCH used in PROFILE. Perhaps have a consistent description (and only described in one place in the text) L15. Why compare PROFILE and ForSAFE. Are they dramatically different, or are you just comparing the effect of different hydrological data on estimates of weathering? The justification for this needs to be clearly stated under section 4. This is a very different comparison to total analysis (which is fundamentally a different estimate of weathering). L16 to L18. This detail should be

presented under the main part of section 4 L19. Why use climate regions? L24. Above you have noted that the differences in estimates of weathering is often driven by differences in inputs (under different applications) for the same model. Here you add further confusion to that issue... What is the goal of the comparison? L29. Above you state they are more-or-less the same, and since that statement we find that one is higher than the other, and so on... which is the truth? L33. Maximum weathering depth? This is confusing, why compare if they represent different depths / pools? This needs clarification, the text suggests that PROFILE covers a shallower depth compared with Total Analysis. They why compare? Normalise both to the same depth before comparing. Page 14 L6. Reword... what comparison between regions? L8-L9. This statement, and similarly many of the statements in the previous section, are very qualitative. There is no quantitative element to the comparison at the site or regional level. Statements such as broadly agree, similar / non-similar are fine IF they are also supported by a quantitative assessment. This is missing. L10+. This has been already stated, and is obvious, it should be noted in the main part of section 4, and not included as part of the comparison (it was known before starting the comparison). However, it would be useful to know the purpose for such a comparison? L15. I think (more-or-less) the results of this assessment are stated here, as such, perhaps the whole assessment could be collapsed to one paragraph? L27. If it is already described, then why repeat here? L30. wording 'dependent' Page 15. L1-L5. Citations? L11. The weathering process in safe is not directly affected by biological processes, it is only affected in as far as the recognition that some of the processes are likely influenced by biology. L12. PROFILE has some biological feedback? Really? L33. 'There is extensive literature...' Page 16. L1 to L34. The entire page (and some of the previous) presents a good review of the 'state of knowledge' but it can be much reduced... and the benefit / objective of such a review should be considered... why cover so much text if this if not part of the work under QWARTS described by the authors. Page 17. L24. 'We found'? Which 'we'? Page 18. L4. Was this stated already? L10. What does the section title mean? L17.Why 'state-of-the-art'. WHAM has been around for decades? L22. Replace 'former' with specific term. Page 19 L10. I wonder if this is some of the context for this manuscript that would be better to present at the start? L20. 'In regions where weathering rates...' L23. Clarify or reword 'data on site index could be found' Page 20 L7. Differences in weathering has already been discussed? L10. This sounds like it was stated already? L21. The preceding text could be summarised much more succinctly. L31. This is more-or-less the summary of the results (if a further quantitative description was added) then this would be sufficient. Page 21. L12 to L18. This has nothing to do with future, it is mostly repetitious text. L22. This section seems odd... we have just been presented with a 2+ page section that covered biological weathering. It is difficult to justify this additional text! I believe this section should be removed, and any 'fresh' text be included above. L23. Yet despite the previous lengthy discussion on biological weathering, we were not introduced to the term 'EPS'?? L23 to L4 Page 22. This text can be deleted. Page 22 L5to L11. Is this model development or uncertainties? L24. Again, it is difficult to justify such an extension section, shortly after we have already been presented with a discussion on the topic. L25 to L32. This is not model development...just repetitious text Page 23 L1. This is a trivial point, with the right implementation the sped can improve. Just because it takes an hour to cycle to work, does not mean that everyone must cycle! L8 to L11. I am sure this is repetitious text. L14. ' overestimated estimates of weathering rates' –> ' overestimated rates of weathering' L19.So it was not tested? Is this future research? It seems to be more of 'ongoing' research? L20. Is this catchment scale weathering? L21 to L27. This is not future research, this is improvements needed in the application of weathering models (and better linkages with hydrological models). As such, uncertainties or limitations is a more appropriate section. L28 to L4 Page 24. Not future... ongoing /current research? Page 24 L5. See comments above... much of the text presented so far under 'future research' speaks more to limitations in application or uncertainties. L6. This is PROFILE specific text... not the depletion method, or others... L8. Often? L9. 'Not only are...' L13 to L20. This is a very uncertain uncertainty... why so much space for something that is not 'very uncertain'? L24. Unless that span is used to estimate the

uncertainty... L26. This does not make sense. Is it possible to contain the solution space? L28. This is not true. However, many users make that assumption. However, others do not. Page 25. L1. BET may still be the best technology? L2. Are the 'current uncertainties (?)' quantified? Are there uncertainties? L5 to L9. Repetitious text. L10. Improved soil moisture should come with improved soil hydrological modelling... L12. All estimates are modelled? What are the other? L13 to L17 Page 26. This whole section can be deleted. Any useful should be moved to the section on comparisons. This is not 'future research' L23. Manual? This is a bit trivial... delete and move to personal 'to do' list. I suggest you write a paper on this. L28. They were not outliers. They represent measurements for different compartments. This is well understood. Page 26. L10. This is wrong. It is okay to state that. There independent methods? How many methods are there (truly independent)? More correctly, Futter et al. (2012) should have recommended that a method incorporating soil mineralogy be used (all other approaches are surrogates for weathering). L16. Good but you can more clearly call it out as an absurd suggestion. L22. Was some of this difference on single sites driven by differences in depths / inputs? L21 to L30. I agree that these are the primary conclusions from this work; I would urge the authors to reflect on this when revising the manuscript. Much of the text can be reduced and streamline to better present this issue (conclusion). Page 27. L10. Other approaches?

---

## Referee Comment (RC2) · Anonymous Referee #2 · 19 Mar 2019

General Comments

The estimation of weathering rates under field conditions is one of the least constrained exercises in biogeochemical research. Yet weathering is crucial to acid-base chemistry and the supply of mineral nutrients to terrestrial (and aquatic) ecosystems. The wide variety of methods available to estimate weathering rates is itself a testimony to our profound lack of clarity regarding the quantification of the key factors and processes. This paper, part of a series of related papers, attempts to bring together weathering flux estimates from a large number of sites in Sweden, made using several estimation methods. The strength of the paper lies in the large number of sites studied and the thoughtful review of the key uncertainties in the various methods of estimation. It is somewhat hard to evaluate this paper without reading the other papers in the proposed series, but it is clear that, together, these represent a comprehensive state-of-the-science assessment of weathering research, at least in the Scandinavian context.

[Figure]

It is very exciting to see the fruits of the QWARTS project coming to the publication stage.

Specific Comments

1a. The consistency of the weathering rates estimated by different methods at the same sites in this study was remarkable. Older studies frequently reported one or two order of magnitude differences in estimates of weathering by depletion versus budgets. The paper discusses the fact that those older studies were done when acid rain was at its peak, but it is true that precipitation is still highly acidic relative to pristine conditions. Have weathering rates changed substantially in the last 30 years, or were the differences always muted in Swedish soils?

1b. Following up on the previous comment, have the authors tried plotting the estimated weathering rates against one another? For example, PROFILE vs. Depletion, etc.? Perhaps this is in one of the other papers in the series?

2. Given the spatial coverage of the sites in this study, I was surprised by the narrow range of estimated weathering rates - for example, in Figures 2 and 3. Were carbonate sites avoided? Were there other selection criteria for the study sites?

3. When comparing weathering rates to harvesting removals, what rotation length was assumed to calculate a meq/m2/yr value? Perhaps I just missed it, but I could not find it.

4. An awful lot of confidence is placed in this paper in modeling approaches in general and in the PROFILE/ForSAFE family of models in particular. They are good models conceptually and they produce results that appear to track the results from other methods (but see comment 1b above). The problem is that there are no "measured" values of weathering flux to use to validate these (or any) weathering models. So, just as budget-based approaches may be contaminated by non-weathering fluxes like net losses from exchange sites, and depletion approaches may suffer from invalid as-

**BGD**

sumptions, model results almost certainly contain a host of errors. This is addressed somewhat in the paper. Depletion methods and budget methods are based on field observations and data. With all their flaws, they are, at least in my view, fundamentally stronger than model results. To compare them as equivalent approaches is problematic.

---

## Editor Comment (EC1) · Suzanne Anderson (Editor) · 20 Mar 2019

Overall, the manuscript is well written and clear, although several sections are longer and a bit more tedious than others. In particular section 5 on "Potential for biological weathering", and section 8 on "Future research" could be streamlined.

The challenge addressed in the manuscript is whether the losses of nutrients due to forest harvest can sustainably be balanced by release of nutrients by chemical weathering. This simple mass balance could be illustrated with a conceptual diagram showing the relevant fluxes at the beginning of the manuscript. Such a diagram might help focus on the important sources of uncertainty in long-term predictions of sustainability.

The most important question the manuscript, and indeed the QWARTS program addressed is if weathering rates, as computed by different models, are greater than or less than rates of harvest loss. The range of weathering values from different models

gave a sense of uncertainty in weathering rates, but harvest losses were presented as a single value at each site. There must be uncertainty in export losses due to harvesting, and it seems important to convey what these uncertainties are.

Two aspects of weathering that I would have expected to be important (and that might be highlighted in the conceptual diagram I suggest above) are rock type and depth of weathering. The first of these is perhaps simpler to consider. Rock type or soil substrate was not described anywhere or for any site. I could not tell if this was because all of Sweden has the same rock type, and so is dismissed as playing any role in variations in weathering rate or nutrient release, or if something else was going on. I would expect rock type to be the very first control on weathering rate, as nutrient availability on basalt vs limestone vs serpentinite vs . . .. is quite different. The depth of weathering is a more difficult problem to address, yet could also be important. Weathering often occurs at depths below the top 30-50 cm.

A few detailed comments: Acronyms should be defined at their first use. Several are not defined at all, or only after their first use (this list may not be exhaustive): UNECE, CLTRAP, EMF, A2M

Whole-tree harvesting is defined on p 3 as being "harvesting of branches". Does this imply that stems are not included in whole-tree harvesting? This seems contradictory.

Total-regression analysis: what is the temperature sum?

Figure 5: The boxes in the key are so small that they are indistinguishable. The gray patterns all look very similar.

p 8, line 5: Casetou-Gustafson's name is incomplete. p10, line 4: cation, not carion p 12, line 9: Ca, Mg and K at 640 sites, not Ca, Mg and K on 640 sites p. 17, line 2: to sparse grass, not via sparse grass p. 17, line 2: naturally lead-contaminated, not natural, lead-contaminated. p. 18, line 23: However, in 1996. . ., not However, already in 1996. . .

---

## Author Comment (AC1) · 3 May 2019

Referee 1 starts off with a paragraph with general comments, followed by a long paragraph with specific comments. Below we list all comments, followed by our answers. We also attach a pdf with this text, but formatted in a way so that it is easier to separate between comments and answers.

General comments The authors present an overview or synthesis of base cation weathering studies carried out under the Swedish QWARTS project. The paper is clearly part of a special issue, as much of the text refers to other papers in 'this issue'. Given the dependence on other papers, it is not a standalone paper but a very 'Swedish' view of soil weathering. Nonetheless, the paper has an important objective, to demonstrate that despite the variation in estimated soil base cation weathering rates at the site level, there is general agreement and these data can be used to support the assessment of

sustainable forestry. However, the paper falls down in several areas: 1. The text is overly long, at times there is extensive repetition within and between sections, We will shorten the paper by removing redundant and repetitive text and shorten parts that are described in other papers in the special issue. The main changes are listed below: We will change to a more conventional structure with a Methods section and a Result and discussion section. This will eliminate much of the repetitions. This is further described in the answer to general comment #2 below. We will remove descriptions of models not represented in the results, according to the answer to general comment #2 below. We will rework the chapter "Potential for biological weathering", so that the main conclusions from the paper about biological weathering are highlighted and details are removed, according to the description below. This will shorten the chapter by about 30%. We have also described that in the answer to the Editor, and therefore the following text overlaps with parts of the answer to the Editor. The main measures are: -Shortening of the introduction in the first paragraph by removal of the three first sentences on p14, line 19-22. -Removal of details about the capacity of EM fungi to allocate C selectively to different minerals, by removal of 2 sentences starting on p15 line 34, describing results in three studies (Smits et al., 2012; Leake et al., 2008; Schmalenberger et al., 2015). -Removal of redundant first sentences in paragraphs starting on p 16, line 17 and 25. -Removal of paragraph starting on p16, line 33 and ending on p17, line 6, about the study by Smits and Wallander showing the effect of vegetation on apatite weathering. It is enough to mention that the effect of vegetation is partly but not fully built in to the models in the following paragraph. -Removal of the last two paragraphs, starting on p17, line 26 and ending at p18, line 9, which are based on on-going work, and thus would have needed even more explanation for the reader to be able to evaluate it. We will shorten the chapter about Future research, according to the answer to general comment #3 below. 2. The text had a tendency to loose focus, the manuscript jumps between project summary, scientifigc review, and comparison of specific results, and while the authors forewarn of the contents in the abstract, the conclusion more succinctly speaks to the true contribution of the paper, if there are

other papers in this special issue, do the authors need to be so broad in their coverage? This comment, as well as general comment #3, questions the structure of the paper. Based on these comments, and recommendation 'a' (below), we will clarify the aims, restructure the text, remove parts that are not contributing to answering the aims, remove repetitious text and shorten parts that are described in other papers in the special issue. This restructuring and shortening of the text is described more thoroughly below, and in answers to general comment #1 and #3. -As the referees point out, an important aim of the paper was to investigate the variation in weathering rates from different approaches, review the results with the key uncertainties for each method in mind and assess the robustness of the results in relation to sustainable forestry. However, a second aim of the paper, based on the research progress within the QWARTS programme, was to highlight potential implications for these results of new in-sights regarding two mechanisms identified as potential sources of errors in modelled weathering rates: lack in the representation of biological weathering as well as overly simplified descriptions of base cation exchange and aluminium complexation. A third aim was to propose future research to further reduce uncertainties, also based on insight from the QWARTS programme. We will clarify the aims accordingly in the end of the introduction. -Based on the aims, we don't want to completely remove the parts about the results from the other papers in the special issue, since they contribute to aims #2 and #3. However, we agree that those parts could be substantially shortened, in order not to overlap too much with the other papers in the special issue. -We will change the structure, so that it follows a more conventional scheme, with a Methods section and a Results and Discussion section. We will start the Methods section by describing how we have compiled the data for the weathering rate comparison based on literature studies, we will continue with a brief description of each method (shortened versions of the present ones), including methods for regional applications. Finally we will describe how we made the comparison with harvest losses. We will remove the descriptions of the models not used for any of the sites in the paper, i.e. WHITCH and Crunchflow. By having a pure Methods chapter, we will avoid some of the repetition that now exists, e.g. regarding

model descriptions (in the chapter "Methods for estimating weathering rates" and the chapter "Weathering rate comparisons on a regional scale"). In the Results and discussion chapter we will start with site level results, continue with regional results and then the comparison with harvest losses. Thereafter, sections about potential for biological weathering, more detailed reactions and future research will follow. 3. Section names and section contents are confusing, the section on future research seems to focus on limitations, while repeating text from previous sections, and generally has the feeling that much of the text could have been integrated into previous sections- Regarding confusing section names and contents, see answer to general comment #2, where we describe how we will restructure the paper with new section names. We agree that the section about future research is too long and insufficiently focused. We will shorten it substantially, so that only the parts describing actual future research are kept. We list our major changes below: -The first paragraph will be replaced by one sentence which introduces the chapter. -The chapter "8.1 Model development: Biological weathering" will be more straight forward towards actual model development. Thus, the first paragraph will be removed, so that the chapter starts with the sentence "Existing models...". Minor adjustments are required in that paragraph, since it refers to the paragraph that will be removed. The last paragraph will be removed. -The chapter "8.2 Model development: Higher resolution chemical reaction" is repeating some of the results presented in chapter 6. The repetitive text will be removed (large parts of the first paragraph). -The chapter "8.3 Model development: Implementing weathering brakes" will be shortened, by removing the first part, going directly to the actual work that needs to be done to better model the saturated zone. -The chapter "8.4 Model development: Weathering below the root zone – for surface water quality assessments" will be shortened by removing most of the first paragraph, jumping directly into the further testing and improvement of the ForSAFE2D model. -In the chapter "8.5 Reducing uncertainties in model input data", a few parameters that are of key importance but where the uncertainties are large, are gone through. This chapter will be shortened by removing much of the text describing results from a number of studies, and instead

going directly in to what needs to be improved. -The first paragraph of the chapter "8.6 Comparison between modelled weathering and other estimates of weathering" will be removed. The last paragraph will also be removed, but parts of it will be incorporated in the chapter were the results from the site level approaches are presented. 4. Unfortunately, much of the comparison between weathering estimates is too qualitative, there is no quantitative assessment, statements such as 'they agree', 'do not agree' or 'estimates are similar' need quantitative support. We will add statistical measures, i.e. median and interquartile ranges (or average and range in the case of only two extimates), for each site to the table where the weathering estimates from all single sites, according to the different approaches, are presented (Table 1). We will refer to those measures in the text. Moreover, we will develop statements like 'they agree' or 'they do not agree' so that they reflect the difference quantitatively, by adding information about how much they differ (in %). We will also add a Table, in conjunction with Figure 5, statistical measures of : [harvest losses after wth – weathering] for each of the sites in Figure 5. By doing that we can differ between sites where the results are more or less robust. I suggest the authors (a) step back from their manuscript and try to pinpoint their exact (unique) contribution, (b) they should remove repetitious text, and remove text that is described (reviewed) elsewhere in the special issue, and (c) add a stronger quantitative element to their comparison / assessment of weathering / sustainable forestry. (a) See answer to question #2 above. (b) See answer to question #1 above. (c) See answer to question #4 above. Specific comments Page 2 L1. It was internationally recognised during the 1970s but regionally recognised long before that... 1 to 2 decades! We will change from "first recognised" to "internationally recognized". L2. one could argue that the peak was a little later... 1980s to 1990s? We will change to "1980s to 1990s". L4. Reword / clarify 'more harvest', more correctly you are referring to the use of forest residues for renewables! We will replace "harvest" with "forest residues" L7. lab –> laboratory We will change accordingly. L7. There was no intensive modelling? Perhaps 'extensive' is superfluous? Yes, we will remove "extensive". L8. Simplify (here and throughout): 'This paper presents the state...' We

will make clarify the aim based on the general comments of Referee 1, as described in the answers to those comments above. We will try to simplify them according to this comment, when we do these changes. L9. You jump too quickly into the specific of the results, give the reader a more guided introduction, 'Under the project, we found that... ' We will start the presentation of the results with the suggested phrase. L10. Variation from what? Data? Methods? Remember the international audience knows nothing of the project! We refer to the variation in estimated weathering rates from different approaches. We will clarify that. L12. Important but the manuscript would greatly benefit from the 'word smiting' of the native English-speaking co-authors, Finlay and Bishop? The manuscript was corrected by a professional language editor before submission. Bishop will be deeply involved in the review work, which will ensure the quality of the revised text. L13. I think this is an important result but the term 'clear imbalances' obscures the implications of the findings. The activities are unstainable. We will clarify the text: "...showed sites where whole-tree harvesting was clearly not sustainable...". L16. Step back and provide greater support... approaches based on the weathering of (observed) mineralogy, such as PROFILE..., provide the most important fundamental understanding of the contribution of weathering to long-term availability of base cations to support forest growth, nonetheless, these approaches should be continually assessed against...' We will change the text based on the suggestion. L19. this point needs further development / clarity The two last sentences will be modified: "Uncertainties in the model approaches can be further reduced, mainly by finding ways to reduce uncertainties in input data on soil texture and associated hydrological parameters, but also by developing the models to better represent the delivery of weathering products to runoff waters and biological feedbacks under the influence of climate change. Another future research activity will be to develop the models to better represent the delivery of weathering products to runoff waters," Page 3. L1. change acid to acidic throughout We will go through the document and change accordingly. L3. remove one 'processes' We will reformulate to: "...extensive research examined processes that acidifies and counteracts acidification...". L5. refer to SWAP first, it

started before NAPAP (and is more important in a European context) We will change the order of SWAP and NAPAP. L7. You need to provide more context for critical loads; it is an effects-based approach for emissions reductions, essentially a direct response to the recognition that emissions of sulphur dioxide were causing significant impacts. Notably it has nothing to do with SWAP or NAPAP! We will develop the part about critical loads accordingly. L9. 'A critical load...' We will change from "The.." to "A..". L11. '... critical loads of acidity...' We will add "..of acidity". L15. Yes, very true but those of us interested in water barely consider weathering directly...? We don't understand what Referee 1 suggests here. We have not done any changes. L16. '... and as such a sink of acidity'. We will revise as suggested. L19. You need to differentiate between plot and catchment scale estimates, models such as MAGIC are process-based, and can be used to provide catchment-based estimates of weathering, however, they are fundamentally different to process-based estimates from PROFILE. We will clarify that the different approaches are addressing different scales. L24. I do not completely agree with this. Many jurisdictions were faced with national scale modelling, and the application of simple approaches such as 'skokloster' provide a practical solution compared to the application of a process-based model that requires quantitative mineralogy on a high spatial resolution... more correctly, given the high loads at that time, the uncertainty in weathering was trivial. We agree that the uncertainties were less important in the times of high loads, which affected the weathering research. We will add a sentence about that. L29. Was it severity, or a shift in policy to support mitigation of climate change impacts? The policies changed when the severity of climate change became fully recongnized, and it was the policy shift which led to higher demand of renewable fuel. We will change the text accordingly. Page 4. L1. increase from 25 Yes it increased again after 2016, it was a temporal dip. We will clarify that. L3. Substantial Will be corrected. L5. Depletion methods needs more description... or just exclude such detail for the moment (estimates of base cation weathering...) The description of the depletion method comes later. We did not do any changes. L11. 'Akselsson et al. (2007) used a mass balance approach (with weathering estimated using PROFILE) ...' We will

change accordingly. L17. I suggest 'Similarly, the influence of whole-tree harvesting ...' Yes, we agree and will change accordingly. L20. Yes, but only in a Scandinavian context, this has not spilled over into the rest of Europe or north America (yet). OK, it is interesting to think about why. We will add "in Scandinavia". L23. Was the conclusion valid? I would suggest the greatest uncertainty was derived from comparing approaches that should not have been compared? We agree that it makes no sense to compare weathering rates estimated for different soil depths. That is also what Futter et al concluded, which we write about in the next paragraph. We have not done any changes here. L30. It is okay to call out errors. Three approaches? In truth there are two. Mass balance approaches where you indirectly estimate weathering rate (there are also other indirect methods) OR mineralogy-based approaches, often if mineralogy is not available you have surrogate-based approaches but 'at least three approaches' may verge on ridiculous? Weathering is the breakdown of minerals... so what does three independent approaches refer to? We are also skeptical about the conclusion about three approaches. However, we have written about that in the discussion, and base it on the results from this study. We think that the discussion is a better place to "call out errors" than the introduction.

Page 5. L1. Provide background on the depletion method... total analysis regression... the reader need help. We write more about this later, we don't want to make the introduction too long. However, we will with a few words describe each method also here. L3-4. reference to other methods are difficult to navigate... We will put each reference directly after the method that they refer to. L7-8. Combine sentences... reduce words... We will shorten the part about the two sources of uncertainties. L17. '... by revisiting older w...' Will be corrected. L21. Flows or removals? Removals is better. We will change accordingly. L30. Replace flows with 'sources and sinks' We will change to 'sources and sinks'. Page 6. L2. Cite 'Warfvinge and Sverdrup, 1992' for PROFILE OK, we will do change to that reference. L2. The work of Susan Brantley should be cited here We will add a reference to Brantley. L7. The key point here, and what separates PROFILE from other approaches, it that weathering is derived from

the breakdown of mineralogy (an essential input), the other inputs only estimate the amount of minerals that are being weathered. We agree and will clarify that. L13. Again, it might be worth citing Brantley here... We will do that. L17. hydrological model... Will be corrected. L21. The discussion / details on SAFE and ForSAFE can be removed. We agree and will remove this. L30. Simplify to PROFILE This section will be removed, as part of the shortening of the paper. See answer to general comment #2.

Page 7 L4. There is an application of SAFE to Hubbard Brook which models the catchment (compared with MAGIC, VSD, etc.) Also this section will be removed or shortened, we will focus on the methods used in this study, in order to get a shorter and more focused paper, according to the reviewer comments. L7. Should MAGIC be cited here or under 'mass balance' approaches? PnET-BGC is another example of a model that uses a mass-balance approach We actually thought a lot about that. It fits in both chapters. We have thought about it again and will put it under mass balance approaches. L17. Assumes that deeper soil is the parent material, so does not work for glaciofluvial soils, etc. We will clarify that in assumption nr 2 on row 23. L26. Could add that the approach has been widely used and cite a few examples? We will do that. L27. Typically referred to as 'Catchment mass balance budgets' as they are widely estimated at the catchment scale, as such the estimates of weathering are an average of a larger landscape unit and can be highly influenced by localised geology. We have examples of both catchment mass balance budgets and site mass balance studies in this paper. The weathering assessments in Asa and Flakaliden are site based. Therefore, we will not change to 'Catchment mass balance budgets'. L28. MAGIC should really be mentioned in this section! See answer to comment on L7 about. Page 8 L10. Retitle to 'strontium isotope ratio' We will. L17. This really should be included under the depletion, as it is a derivative of that approach We agree and will merge those chapters. We separated them because they have been presented as different methods before, but we agree that it is confusing. L24. Remind the reader that you are focused on QWARTZ, i.e., 'Under QWARTZ, weathering ...' We will change

accordingly. L30. 'profile 17-20cm deeper', this is a little unclear. I assume in simple terms the soil depth differs between estimates... Maybe present table on a 'weathering per cm'? We will simplify to "...the highest rates in the interval are associated with the deepest soil profiles." We don't want to add a new table, since we are trying to shorten the paper, and since the depths are the same for the same site except for in Gårdsjön and Svartberget. Page 9 L2. This is an important point and should perhaps be stated much earlier, homogeneity of soil and bedrock are important considerations for agreement / lack of agreement between approaches We think it fits good here, in the very first part of the results. L15. Why does the depletion indicate a lower weathering rate? This could suggest that the un-weathered layer did have weathering? In many of the studies in Table 1, the depletion method is lower. Why? However, it may also be argued that the range between methods is smaller than the uncertainty? We have some potential explanations to this in rows 27-31. One of the is according to the suggestion above about that the "original till" is actually weathered till from earlier glaciations. L21. Do you mean 'soil bulk density'? Yes, we will clarify that. L23. Correct 'to to' We will remove one 'to'. Page 10 L19. Disqualified? Excluded! However, just exclude, and note in a footnote to the Table. No need to explain We will change to 'excluded' and refer to Stendahl et al. instead of explaining. However, we keep it here, not in a footnote in the table, since the sites are not part of the table, and a footnote about three other sites feels a bit out of place. L30. Till is the most obvious / likely explanation... remove other excludes / reasons and only present this one. We will change the order, so that the till explanation comes first. However, we keep the others as they also might be contributory explanations. L34. What does 'conceptual limitations' mean? clarify We agree that this is not a good expression. We will clarify, ased on the discussion in Stendahl et al. (2013). Page 11 L1. 'for Ca and Mg...' We will add 'for'. L3 to L10. This can be reduced to one sentence, state 'The majority of weathering rate estimates were classified as acidic or intermediate (cite Table, UNECE, etc.). We will reduce the text accordingly. L4. Figure 2 essentially repeats Table 2; the classification could be added to the Table 2 instead. Is the suggestion to

remove Table 2? We think that it gives a good overview, much better than the tables, so we would like to keep it. L10 to L14. This text repeats detail already discussed. We will remove repetitious text here.

L20. This is a long section, and rather than go through each site (one by one), it would be more efficient to summarise and focus on the broad agreement, and disagree, but describe from the point-of-view of the factors that drive disagreement, e.g. ' ... there was slight disagreement between some estimates owing to difference in input data use by the different approaches, such as soil depth (give example) or soil moisture (give example). The table is provided, so the reader can evaluate the results, and there is no need to describe in detail. We will rework and shorten this section, according to the suggestion. L22. are they scaled-up or just regional applications (more sites)? How are they scaled? It is more regional applications (more sites). We will clarify that. L25 to L33. The relationship between ForSAFE and PROFILE has already been described at length. There is no need to repeat again. This will be solved by restructuring the paper to a more conventional structure with methods, results and discussion, see more detailed description in the answer to general comment #2. Page 12 L7. Is the analysis really only based on 11 sites? Did the study use 11 sites to predict at 400+? Yes, this is described in Olsson et al. (1993). Page 13 L5. Is this the same approach as used with PROFILE? Did both use UPPSALA? Clarify No, for PROFILE another normative method was used. We will add information about that. L14. Again, is this similar to MATCH used in PROFILE. Perhaps have a consistent description (and only described in one place in the text) Yes, MATCH was used for both. We will restructure so that all method descriptions are in one place, see answer to general comment #2. L15. Why compare PROFILE and ForSAFE. Are they dramatically different, or are you just comparing the effect of different hydrological data on estimates of weathering? The justification for this needs to be clearly stated under section 4. This is a very different comparison to total analysis (which is fundamentally a different estimate of weathering). We agree that the comparison is very different than a comparison with to completely different approaches. PROFILE and ForSAFE are built on the same

weathering reactions, but the dynamics in ForSAFE affects weathering rates. Also, the fact that e.g. hydrology is modelled in ForSAFE, whereas it is input data in PROFILE, may affect weathering rates. This is described thoroughly in Kronnäs et al. in the same issue. We will clarify this, and also explain why we want to include both PROFILE and ForSAFE in the Methods part of the paper. L16 to L18. This detail should be presented under the main part of section 4 This will be solved with the new structure with a methods section (see answer to general comment #2).

L19. Why use climate regions? We want to show regional differences, and those regions are logical in the way that they represent different climate and atmospheric deposition regimes. They have been used in one paper, just accepted for publication, and are used also in Belyazid et al in this issue (under review). L24. Above you have noted that the differences in estimates of weathering is often driven by differences in inputs (under different applications) for the same model. Here you add further confusion to that issue... What is the goal of the comparison? The goal of the comparison was not to run different approaches on many sites with exactly the same input data. The goal was to collect existing weathering estimates from different approaches, in many cases run independently of each other, and compare, to get a span covering both differences in approaches and differences in input data used. This represents the reality for e.g stakeholders. We show that, even though we do like this, we can draw conclusions about sustainability. We will, when we rework the aim according to the answer to general comment # 2, clarify this. L29. Above you state they are more-or-less the same, and since that statement we find that one is higher than the other, and so on... which is the truth? In the beginning of the chapter we write: "The weathering rates varied widely within the regions for all methods, but there were no large systematic differences between the medians or ranges for the different methods. However, ForSAFE gave somewhat higher medians than PROFILE for all regions, especially in the northern regions." On L29 we write: "The total analysis regression method gave somewhat higher weathering rates than PROFILE for several of the regions." We don't think that this is contradictory, but to clarify we will move the

sentence about that total analysis regression gave somewhat higher weathering than PROFILE to the beginning of the chapter, right after the sentence about ForSAFE. L33. Maximum weathering depth? This is confusing, why compare if they represent different depths / pools? This needs clarification, the text suggests that PROFILE covers a shallower depth compared with Total Analysis. They why compare? Normalise both to the same depth before comparing. In this paper we have used results already published. We have decided not to try to normalize them, then we would add extra assumptions (e.g that weathering per cm is constant in the soil profile, which we know is not true). We know e.g. that the weathering rates according to the depletion method are much slower further down in the soil profile. We have thus decided to discuss the differences in depths instead of normalizing. We will try to clarify that when we rework the aim (see also answer to comment to L24 above). Page14L6. Reword... what comparison between regions? Here we mean the comparison of weathering rate intervals between different regions. We will clarify this. L8-L9. This statement, and similarly many of the statements in the previous section, are very qualitative. There is no quantitative element to the comparison at the site or regional level. Statements such as broadly agree, similar / non-similar are fine IF they are also supported by a quantitative assessment. This is missing. See answer to general comment #4. L10+. This has been already stated, and is obvious, it should be noted in the main part of section 4, and not included as part of the comparison (it was known before starting the comparison). However, it would be useful to know the purpose for such a comparison? This will be solved by restructuring according to the answer to general comment #2. About the reason: see answer to comment L15 and L24. L15. I think (more-or-less) the results of this assessment are stated here, as such, perhaps the whole assessment could be collapsed to one paragraph? The results from site-level and regional applications will be lumped together in one results chapter, and yes, it will be substantially shortened. L27. If it is already described, then why repeat here? Repetitious text will be removed. L30. wording 'dependent' We will reformulate. Page 15. L1-L5. Citations? References will be added. L11. The weathering process

in safe is not directly affected by biological processes, it is only affected in as far as the recognition that some of the processes are likely influenced by biology. Biological processes do affect weathering in PROFILE, SAFE and ForSAFE through the following pathways: 1- uptake reduces cation concentrations, directly alleviating the brakes on weathering, 2- transpiration reduces water availability, thus limiting one of the four weathering kinetics (dissolution in water), and increasing concentrations, which can have positive (in case of acids) or negative (in case of cations) effects on weathering, 3- cation uptake produces more protons in the soil solution, lowering pH and promoting weathering, and 4- the production of dissolved organic radicals through litterfall decomposition and in the case of ForSAFE exudation contribute directly to one of the four weathering kinetics. Through these four pathways, the models do directly modify weathering rates based on the outcome of biological activity as described here. L12. PROFILE has some biological feedback? Really? See answer to L11. L33. 'There is extensive literature...' We will remove "an". Page 16. L1 to L34. The entire page (and some of the previous) presents a good review of the 'state of knowledge' but it can be much reduced... and the benefit / objective of such a review should be considered... why cover so much text if this if not part of the work under QWARTS described by the authors. We will shorten this chapter, see answer to general comment Âď1. Page 17. L24. 'We found'? Which 'we'? We will reformulate. Page 18. L4. Was this stated already? The chapter will be substantially shortened, we will make sure to get rid of repetitious text. L10. What does the section title mean? We will try to simplify the title. L17.Why 'state-of-the-art'. WHAM has been around for decades? We will remove 'state of the art'. L22. Replace 'former' with specific term. OK, we will do that. Page 19 L10. I wonder if this is some of the context for this manuscript that would be better to present at the start? We don't understand what the suggestion is here. L20. 'In regions where weathering rates...' We will change accordingly. L23. Clarify or reword 'data on site index could be found' We will change to "site quality". Page 20 L7. Differences in weathering has already been discussed? We will shorten the text about this, but we need to mention the differences also here,

since the sites with large differences are the ones where it is more difficult to draw conclusions about sustainability. L10. This sounds like it was stated already? No, this is the first time we compare weathering and WTH in the north. However, we will restructure and shorten the text to make it easier to follow. L21. The preceding text could be summarised much more succinctly. Se answer to comment L10 above. L31. This is more-or-less the summary of the results (if a further quantitative description was added) then this would be sufficient. Se answer to comment L10 above. Page 21. L12 to L18. This has nothing to do with future, It is mostly repetitious text. See answer to general comment# 3. L22. This section seems odd... we have just been presented with a 2+ page section that covered biological weathering. It is difficult to justify this additional text! I believe this section should be removed, and any 'fresh' text be included above. We will shorten the chapter substantially, according to answer to general comment #3. L23. Yet despite the previous lengthy discussion on biological weathering, we were not introduced to the term 'EPS'?? This will be looked over when we do the comprehensive revision of the biological weathering part. L23 to L4 Page 22. This text can be deleted. We don't understand which text Referee 1 refers to, it seems to be something wrong with the line numbers (L23-L4?) Page 22 L5 to L11. Is this model development or uncertainties? It is a lot of uncertainties. We will shorten the chapter substantially, according to answer to general comment #3. L24. Again, it is difficult to justify such an extension section, shortly after we have already been presented with a discussion on the topic. This will be shortened, to only contain the main model improvements required. L25 to L32. This is not model development...just repetitious text See answer to L24 above. Page 23 L1. This is a trivial point, with the right implementation the sped can improve. Just because it takes an hour to cycle to work, does not mean that everyone must cycle! The organic complexation does not have an arithmetic solution, the only way to do it without compromising its purpose is by optimization (of multiple complexation parameters) and a whole set of assumptions (including that the Al pool is constant for example), i.e. iterations until near-equilibrium. In HD-Minteq this is feasible because most state variables and fluxes

(soil organic matter, decomposition, uptake. . .) are given as input and therefore not dependent on changes in soil solution chemistry. While in ForSAFE these processes are internally simulated, meaning that each iteration of Al complexation will require an entire simulation of all fluxes and equilibria, exponentially increasing computation time or even requiring a new optimization of exchange parameters, which in the word of the original author "is impossible" (see Gustafsson 2001, Journal of Colloid and Interface Science 244, 102-112. doi:10.1006/jcis.2001.7871). L8 to L11. I am sure this is repetitious text. We will remove this. L14. ' overestimated estimates of weathering rates' –> ' overestimated rates of weathering' We will change accordingly L19.So it was not tested? Is this future research? It seems to be more of 'ongoing' research? New brakes have not been implemented and tested in PROFILE/ForSAFE. This is future work. In Erlandsson et al., we implemented and tested silicate release through weathering and the feedback brakes from Si concentrations on the four weathering kinetic equations, while dictating other inputs such as uptake, water flow and so on. This has produce the expected results (constraining weathering in the saturated zone). Implementation in PROFILE/ForSAFE has not yet been carried out. L20. Is this catchment scale weathering? Yes, we will call it that instead. L21 to L27. This is not future research, this is improvements needed in the application of weathering models (and better linkages with hydrological models). As such, uncertainties or limitations is a more appropriate section. We will, in the restructuring, divide the chapter called "Future research" in a chapter called "Model Limitation/Uncertainties" or similar and one chapter called "On-going and future research" or similar. Both wil be part of the "Discussion" chapter. L28 to L4 Page 24. Not future... ongoing /current research? See answer to comment L21-L27 above. Page 24 L5. See comments above... much of the text presented so far under 'future research' speaks more to limitations in application or uncertainties. See answer to comment L21-L27 above. L6. This is PROFILE specific text... not the depletion method, or others... Yes. We will clarify that in the title. L8. Often? Often will be removed. L9. 'Not only are...' We will correct this. L13 to L20. This is a very uncertain uncertainty... why so much space for something that

is not 'very uncertain'? It is part of the QWARZ work and refers to one of the papers in the special issue. So we want to refer to it, but we will shorten the paragraph. L24. Unless that span is used to estimate the uncertainty... It is in one of the papers of the special issue. But still, often one value is used. L26. This does not make sense. Is it possible to contain the solution space? The reviewer is right that this is not a problem if we have information on which minerals are present, their respective stoichimetries and in what proportions (See Posch and Kurz, 2007). This however is most often missing, and that is the purpose of using A2M. It would, as indicated, be possible to narrow the space in more information is available, on for example the fraction of light primary minerals in the soil. So yes, it is possible to constrain the space if a considerably more information is available. L28. This is not true. However, many users make that assumption. However, others do not. In the cases we have seen, where one of the mineralogies is chosen from A2M, it has been the centre point, although everyone who uses A2M should know that all solutions are as likely. We think that we have been clear about that all have the same probability, but that many chose the center point, since they want one value. Page 25. L1. BET may still be the best technology? We will remove "using modern technology". Even if the BET technology hasn't developed since the 80/90:ies (which we don't know), the regressions could be revised based on a larger soil material. L2. Are the 'current uncertainties (?)' quantified? Are there uncertainties? The regression graphs in Warfvinge and Sverdrup (1995) indicate large uncertainties. We will clarify that. L5 to L9. Repetitious text. When restructuring, we will make sure to only write about this once. L10. Improved soil moisture should come with improved soil hydrological modelling... Yes, and since this chapter is about improved input data we mention soil data which is very important for the hydrology modelling in ForSAFE. We don't understand what Referee 1 suggests here. L12. All estimates are modelled? What are the other? We will change to "Comparison of weathering estimates from different methods" or similar. L13 to L17 Page 26. This whole section can be deleted. Any useful should be moved to the section on comparisons. This is not 'future research' OK, this will be

removed from here. L23. Manual? This is a bit trivial... delete and move to personal 'to do' list. I suggest you write a paper on this. OK, we remove this. However, it is very difficult to evaluate results from different depletion method studies, since it is so much up to the user to set e.g. the reference depth. It would have been helpful if a routine was developed, maybe even built in to a model. L28. They were not outliers. They represent measurements for different compartments. This is well understood. Mass balance methods are still used for weathering rate calculations and advocated by many. We included it here and came to the conclusion that it was problematic in the cases we had. We write "In the compilation of weathering rates in this paper, the most extreme outliers came from the budget method, which can be explained by the fact that other sources than weathering are included (Rosenstock et al., in review (this issue)). For a fair comparison between weathering rates from the budget method and from other methods, ways to distinguish between different sources need to be further developed." We think that this is a rather strong recommendation, which we would like to keep as it is. Page 26. L10. This is wrong. It is okay to state that. There independent methods? How many methods are there (truly independent)? More correctly, Futter et al. (2012) should have recommended that a method incorporating soil mineralogy be used (all other approaches are surrogates for weathering). We think that we are rather clear when we conclude that it is unrealistic. We think that it is stronger to say that, than to say that is absurd or wrong, which are more "value" words. L16. Good but you can more clearly call it out as an absurd suggestion. We think it is enough to say that it is unrealistic in practice. L22. Was some of this difference on single sites driven by differences in depths / inputs? Yes. We will change to "Although the variation in weathering estimates was large on single sites". By doing that we include methods as well as input data and to some extent depths. L21 to L30. I agree that these are the primary conclusions from this work; I would urge the authors to reflect on this when revising the manuscript. Much of the text can be reduced and streamline to better present this issue (conclusion). We will do this, see further answer to general comment # 1-3. Page 27. L10. Other approaches? Yes,

the depletion method, mass balance calculations and other models. We will clarify that.

Please also note the supplement to this comment:
https://www.biogeosciences-discuss.net/bg-2019-1/bg-2019-1-AC1-supplement.pdf

---

## Author Comment (AC2) · 3 May 2019

Referee 2 starts off with a paragraph with general comments, describing the paper and its strengths. In this paragraph there are no questions or suggestions of changes. After the general comments, a few specific comments are listed. Below we list the specific comments, followed by our answers. We also attach a pdf with this text, but formatted in a way so that it is easier to separate between comments and answers.

Specific comments 1a. The consistency of the weathering rates estimated by different methods at the same sites in this study was remarkable. Older studies frequently reported one or two order of magnitude differences in estimates of weathering by depletion versus budgets. . . . . . . . . . . . Have weathering rates changed substantially in the last 30 years, or were the differences always muted in Swedish soils? In our site level comparisons we have included both old and new estimates of weathering rates, for all

sites in Sweden that we have found where estimations have been done to the same depth with at least two methods. We think that the consistency is remarkable on some sites (e.g Gårdsjön C taken from a publication from 1998) whereas there is less consistency on other sites, e.g. Asa, Flakaliden and many of the sites from Stendahl et al. (2013). The differences are however not one or two orders of magnitudes. If the studies that Referee 2 refers to are from Nordic sites, we are not aware of which they are. We know that such large differences have been seen in other settings though. One example is when mass balance estimates of weathering are included. We have commented on this. Both the uncertainty in the components of the mass balance, and the possibility of changing pools render this an unsatisfactory method for weathering estimates. In our case several mass balance estimates are included, and they do expand the range of estimates, even though not to order of magnitude levels. Another way to create large uncertainties is to compare studies that have some fundamentally inconsistent assumptions, such as depth of soil considered. This is the situation at Svartberget, where the differences were orders of magnitudes as reported by Klaminder et al. (2011). In that study weathering rates for different root depths and from catchment level calculations were compared, and site level estimates were compared with regional estimates. All of the differences between those estimates cannot be attributed to uncertainties. We only included the estimates from that site, here called Svartberget B, that referred to the same exact site and the same root depth. There, the budget calculations gave weathering rates more than twice as high as the depletion method, which is the case for three of the four sites where both depletion method and the budget method has been applied (Table 1): Svartberget B, Flakaliden and Asa. Only in Gårdsjön the results are of similar size. We have not done any changes based on this comment, we would need additional information about which studies Referee 2 refers to.

1b. Following upon the previous comment, have the authors tried plotting the estimated weathering rates against one another? For example, PROFILE vs. Depletion, etc.? Perhaps this is in one of the other papers in the series In Stendahl et al.

(2013) weathering rates from the depletion method was plotted against PROFILE weathering rates on 16 sites. The number of sites in that study is enough to be able to draw general conclusions about differences in weathering rates from the two different methods. In the present study, those sites are included, but also other sites, based on different combinations of methods. Based on the material in this paper, it is not as straight forward to do complementary pairwise comparisons, since most of the approaches (all except PROFILE and the depletion method that are already compared) are performed only on a few sites. Instead we have focused on presenting the weathering rate intervals for each site, as a way of framing the weathering rates. 3. "When comparing weathering rates to harvesting removals, what rotation length was assumed to calculate a meq/m2/yr value?" We use site quality (biomass per area unit and year) to calculate harvesting removals, in the same way as in Akselsson et al. (2007: 2016) and Stendahl et al. (2013), which we refer to. Site quality (in Swedish "bonitet") is the biomass growth on a specific place if the forest is managed optimally. In our calculations we assume that the actual growth is 80% of the optimal growth. The average growth in the site quality concept "assumes " that the forest is harvested at the optimal point in time, which varies between north and south and between stands, from about 70 years in the south to 120 years in the north. We will explain more thoroughly how the harvesting losses were estimated. 4. "An awful lot of confidence is placed in this paper in modeling approaches in general and in the PROFILE/ForSAFE family of models in particular They are good models conceptually and they produce results that appear to track the results from other methods (but see comment 1b above). The problem is that there are no "measured" values of weathering flux to use to validate these (or any) weathering models. So, just as budget-based approaches may be contaminated by non-weathering fluxes like net losses from exchange sites, and depletion approaches may suffer from invalid assumptions, model results almost certainly contain a host of errors. This is addressed somewhat in the paper. Depletion methods and budget methods are based on field observations and data. With all their flaws, they are, at least in my view, fundamentally stronger than model

results. To compare them as equivalent approaches is problematic." We agree that the depletion method and the budget method, that are based on measurements, are very important in framing weathering rates, which we also point out in the paper. However, we don't think that the results from those methods are fundamentally stronger than the model results (which actually also are based on measurements to a large extent, e.g. laboratory measurements). On the contrary, we think that further studies are required if more robust results from those methods are desired – something we point to in the conclusions. There are not short-cuts to certainty, but in this paper we have tried to show how far we can go with the available publications. Regarding budget calculations our results show that the most extreme outliers came from the budget method. We explain that by the fact that other sources than weathering are included as discussed in Rosenstock et al., in review (this issue), as well as the documented uncertainty in terms used in the mass balances. We conclude that "for a fair comparison between weathering rates from the budget method and from other methods, ways to distinguish between different sources need to be further developed." The depletion method gave unrealistically low weathering rates in some cases. On one of the sites, Asa, that was studied in detail within another paper in this special issue (Casetou Gustafson et al., in review), the analysis of the soil profile indicated that the soil profile has been disturbed, introducing errors in the weathering estimates. We highlight the importance of excluding sites with disturbed profiles as well as to perform studies where the average weathering rate since the last glaciation is related to the present weathering rates are required, to be able to make necessary adjustments of the historical rates.

Please also note the supplement to this comment:
https://www.biogeosciences-discuss.net/bg-2019-1/bg-2019-1-AC2-supplement.pdf

---

## Author Comment (AC3) · 3 May 2019

Below we list the comments from the Editor, followed by our answers. We also attach a pdf with this text, but formatted in a way so that it is easier to separate between comments and answers.

General comments Overall, the manuscript is well written and clear, although several sections are longer and a bit more tedious than others. In particular section 5 on "Potential for biological weathering", and section 8 on "Future research" could be streamlined. We will rework the chapter "Potential for biological weathering", so that the main conclusions from the paper about biological weathering are highlighted and details are removed. This will shorten the chapter by about 30%. The main measures are: -Shorten the introduction in the first paragraph by removal of the three first sentences on p14, line 19-22. -Removal of details about the capacity of EM fungi

to allocate C selectively to different minerals, by removal of 2 sentences starting on p15 line 34, describing results in three studies (Smits et al., 2012; Leake et al., 2008; Schmalenberger et al., 2015). -Removal of redundant first sentences in paragraphs starting on p 16, line 17 and 25. -Removal of paragraph starting on p16, line 33 and ending on p17, line 6, about the study by Smits and Wallander showing the effect of vegetation on apatite weathering. It is enough to mention that the effect of vegetation is partly but not fully built in to the models in the following paragraph. -Removal of the last two paragraphs, starting on p17, line 26 and ending at p18, line 9, which are based on on-going work, and thus would have needed even more explanation for the reader to be able to evaluate it. We agree that the section about future research is too long and insufficiently focused. This is in line with comments from Referee 1, and our answer here therefore overlaps to a large extent with our response to Referee 1. We will shorten it substantially, so that only the parts describing actual future research are kept. We list our major changes below: -The first paragraph will be replaced by one sentence which introduces the chapter. -The chapter "8.1 Model development: Biological weathering" will be more straight forward towards actual model development. Thus, the first paragraph will be removed, so that the chapter starts with the sentence "Existing models. . .". Minor adjustments are required in that paragraph, since it refers to the paragraph that will be removed. The last paragraph will be removed. -The chapter "8.2 Model development: Higher resolution chemical reaction" is repeating some of the results presented in chapter 6. The repetitive text will be removed (large parts of the first paragraph). -The chapter "8.3 Model development: Implementing weathering brakes" will be shortened, by removing the first part, going directly to the actual work that needs to be done to better model the saturated zone. -The chapter "8.4 Model development: Weathering below the root zone – for surface water quality assessments" will be shortened by removing most of the first paragraph, jumping directly into the further testing and improvement of the ForSAFE-2D model. -In the chapter "8.5 Reducing uncertainties in model input data", a few parameters that are of key importance but where the uncertainties are large, are gone through. This chapter will be shortened

by removing much of the text describing results from a number of studies, and instead going directly in to what needs to be improved. -The first paragraph of the chapter "8.6 Comparison between modelled weathering and other estimates of weathering" will be removed. The last paragraph will also be removed, but parts of it will be incorporated in the chapter were the results from the site level approaches are presented. The challenge addressed in the manuscript is whether the losses of nutrients due to forest harvest can sustainably be balanced by release of nutrients by chemical weathering. This simple mass balance could be illustrated with a conceptual diagram showing the relevant fluxes at the beginning of the manuscript. Such a diagram might help focus on the important sources of uncertainty in long-term predictions of sustainability. We will add a figure in the Introduction, showing the relevant fluxes in the simple mass balance of base cations in the forest (weathering, deposition, harvest losses and leaching), to put our weathering results and the sustainability assessments in a context. The most important question the manuscript, and indeed the QWARTS program addressed is if weathering rates, as computed by different models, are greater than or less than rates of harvest loss. The range of weathering values from different models gave a sense of uncertainty in weathering rates, but harvest losses were presented as a single value at each site. There must be uncertainty in export losses due to harvesting, and it seems important to convey what these uncertainties are. We agree that the sustainability of harvesting is an important part of this paper, and that the uncertainties in the harvesting estimates should be mentioned. Research related to those uncertainties were not included in QWARTS, which focused on the actual weathering rates, but we will add a paragraph about the uncertainties, based on results from other studies, e.g. Zetterberg et al. (2014) in Science of the Total Environment. Two aspects of weathering that I would have expected to be important (and that might be highlighted in the conceptual diagram I suggest above) are rock type and depth of weathering. The first of these is perhaps simpler to consider. Rock type or soil substrate was not described anywhere or for any site. I could not tell if this was because all of Sweden has the same rock type, and so is dismissed as playing any role in variations in weathering rate

or nutrient release, or if something else was going on. I would expect rock type to be the very first control on weathering rate, as nutrient availability on basalt vs limestone vs serpentinite vs : : :. is quite different. The depth of weathering is a more difficult problem to address, yet could also be important. Weathering often occurs at depths below the top 30-50 cm. Yes parent material is of key importance for weathering rates. We will include a table describing parent material, possibly as a supplementary table. For the sites in which some of the authors have been involved in the weathering estimations, we have thorough databases from which we can get the information. For the other sites, we will extract information from published papers. We have focused on the root zone in this study, since we are interested in the sustainability of forestry. Therefore, for this study we see the depth of the root zone as more interesting that the weathering depth, therefore we haven't discussed the weathering depth. In Swedish applications the depth 50 cm is often used, based on studies in Sweden. In this paper, where one important aim is to compare different approaches, we have included sites where weathering rates have been calculated to deeper depths, up to 1 m, but we have only accepted sites where the different approaches have been run to the same depths. In the sustainability calculations only sites with depths<0.7 m are included. The uncertainties related to how much of the soil profile the roots can reach are large. We address this in the section about sustainability: "Currently there is even less research on weathering rates in relation to root depth. . ..than on weathering."

A few detailed comments: Acronyms should be defined at their first use. Several are not defined at all, or only after their first use (this list may not be exhaustive): UNECE, CLTRAP, EMF, A2M Will be done! Whole-tree harvesting is defined on p3 as being "harvesting of branches". Does this imply that stems are not included in whole-tree harvesting? This seems contradictory. Stems are included in whole-tree harvesting, we will clarify that! Total-regression analysis: what is the temperature sum? The temperature sum is a measure often used in forestry. It is the daily mean temperature above a threshold value, in Sweden often +5°C, summed during the growing season. We will develop this a bit in the text. Now we only refer to a

paper (Morén and Perttu). Figure 5: The boxes in the key are so small that they are indistinguishable. The gray patterns all look very similar. We will improve Figure 5 based on the comments. Figure 5 is now written twice in the figure caption, we will correct that. p 8, line 5: Casetou-Gustafson's name is incomplete. Will be corrected. p10, line 4: cation, not carion Will be corrected. p 12, line 9: Ca, Mg and K at 640 sites, not Ca, Mg and K on 640 sites Will be corrected. p. 17, line 2: to sparse grass, not via sparse grass Will be corrected. p. 17, line 2: naturally lead-contaminated, not natural, lead-contaminated. Will be corrected. p. 18, line 23: However, in 1996..., not However, already in 1996.. Will be changed.

Please also note the supplement to this comment:
https://www.biogeosciences-discuss.net/bg-2019-1/bg-2019-1-AC3-supplement.pdf

---

## Author Response (AR1)

**Answer to Referees and Editor, and description of how the comments were handled**

*Cecilia Akselsson, August 2019*

Associate Editor Suzanne Anderson asked for major revisions (10 May 2019), based on the comments
from the two reviewers and on our answer to the comments. The manuscript has been thoroughly
reworked, based on the comments, and in our opinion substantially improved. In this document, we
describe how we answer the comments and describe how we have handled them (pages 1-33), and we
also attach the manuscript with track changes (pages 34-98). We first answer the comments and
describe the changes related to the comments from Referee 1 (pages 1-24), then Referee 2 (pages 25-
27), the Editor (pages 28-31) and finally answers to the comments by the Editor that came along with
the decision (pages 32-33). All answers are written in italics, in order to be able to separate between
comments and answers. In the comments, the page and line numbers in the original manuscript are
referred to, whereas in the answers, the page and line numbers in the revised version (without track
changes) are referred to (akselsson_etal_aug2019.pdf)).

**Answers to comments by Referee 1 (R1)**

*Comments from 19 March 2019*

R1 starts off with a paragraph with general comments, followed by a long paragraph with specific
comments.

**General comments**

"The authors present an overview or synthesis of base cation weathering studies carried out under the
Swedish QWARTS project. The paper is clearly part of a special issue, as much of the text refers to
other papers in 'this issue'. Given the dependence on other papers, it is not a standalone paper but a very
'Swedish' view of soil weathering. Nonetheless, the paper has an important objective, to demonstrate
that despite the variation in estimated soil base cation weathering rates at the site level, there is general

agreement and these data can be used to support the assessment of sustainable forestry. However, the paper falls down in several areas:"

"1. The text is overly long, at times there is extensive repetition within and between sections,"

*We have removed redundant and repetitive text and shortened parts that are described in other papers in the special issue. The paper (including first page and references but without Tables and Figures) is now 29 pages, compared to more than 37 pages before. The main changes are listed below:*

*We have changed to a more conventional structure with a Methods section and a Result and Discussion section. This eliminated much of the repetitions. This is further described in the answer to general comment #2 below.*

*We have removed descriptions of models not represented in the results, according to the answer to general comment #2 below.*

*We have reworked the chapter "Potential for biological weathering", which has reduced the length from 3.5 pages to 1 2/3 pages. We have removed detailed descriptions and discussions that can be found in the paper Finlay et al (in review, this issue), and instead focused on the implications of the new in-sights from a modelling perspective.*

*We have shortened the chapter about "Future research", according to the answer to general comment #3 below, and renamed it to "Prospects for method development".*

"2. The text had a tendency to loose focus, the manuscript jumps between project summary, scientific review, and comparison of specific results, and while the authors forewarn of the contents in the abstract, the conclusion more succinctly speaks to the true contribution of the paper, if there are other papers in this special issue, do the authors need to be so broad in their coverage?"

*This comment, as well as general comment #3, questions the structure of the paper. Based on these comments, and recommendation 'a' (below), we have clarified the aims in the end of the introduction, restructured the text, removed parts that are not contributing to answering the aims, removed repetitious text and shortened parts that are described in other papers in the special issue. The aims now read: "The aims of this study were to (1) investigate the variation in weathering rates from*

*different approaches in Sweden, with consideration of the key uncertainties for each method, (2) assess the robustness of the results in relation to sustainable forestry and (3) discuss the results in relation to new insights from the QWARTS programme, and propose future research to further reduce uncertainties." The restructuring and shortening of the text is described more thoroughly below, and in answers to general comment #1 and #3.*

*-We still want to be a bit broad in the coverage, e.g. including the part of biological weathering and the implications of higher resolution of chemical reactions (to answer aim 3), but we have reduced the length of those parts substantially.*

*-We have changed the structure, so that it follows a more conventional scheme, with a Methods section and a Results and Discussion section. We start the Methods section by describing how we have compiled the data for the weathering rate comparison based on literature studies, we continue with a brief description of each method (shortened versions of the old ones), including methods for regional applications. Finally we describe how we made the comparison with harvest losses. We removed the descriptions of the models not used for any of the sites in the paper, i.e. WHITCH and Crunchflow. By having a pure Methods chapter, we avoid some of the repetition that now exists, e.g. regarding model descriptions (in the chapter "Methods for estimating weathering rates" and the chapter "Weathering rate comparisons on a regional scale"). In the Results and discussion chapter we start with site level results, continue with regional results and then the comparison with harvest losses. Thereafter, sections about potential for biological weathering, more detailed chemical reactions and prospects for method development follow.*

3. Section names and section contents are confusing, the section on future research seems to focus on limitations, while repeating text from previous sections, and generally has the feeling that much of the text could have been integrated into previous sections-

*Regarding confusing section names and contents, see answer to general comment #2, where we describe how we have restructured the paper with new section names. We agree that the section about future research is too long and insufficiently focused. We have shortened it substantially, renamed it to "Prospects for method development", and removed or moved parts that do not fit in.*

4. Unfortunately, much of the comparison between weathering estimates is too qualitative, there is no quantitative assessment, statements such as 'they agree', 'do not agree' or 'estimates are similar' need quantitative support.

*We have added medians, maximum, minimum and "maximal deviation from median (%)" in Table 3,*

5 *with site-level weathering rates, and we refer to those numbers in the text (sectin 3.1). For the regional-level comparison and the sustainability assessments, we have added numbers to quantify the difference. In the regional-level comparison we compare medians and the width of the intervals in the text (section 3.2). In the sustainability assessments, we give the difference between weathering rates and harvest losses in %, in the text (section 3).*

10 I suggest the authors (a) step back from their manuscript and try to pinpoint their exact (unique) contribution, (b) they should remove repetitious text, and remove text that is described (reviewed) elsewhere in the special issue, and (c) add a stronger quantitative element to their comparison / assessment of weathering / sustainable forestry.

*(a) See answer to question #2 above.*

15 *(b) See answer to question #1 above.*

*(c) See answer to question #4 above.*

**Specific comments**

Page 2 L1. It was internationally recognised during the 1970s but regionally recognised long before

20 that... 1 to 2 decades!

*We changed from "first recognised" to "internationally recognized". Moreover, we changed "during the 1970's" to "the late 1960's", in accordance with what is written in the introduction. (Page 2, line 1)*

L2. one could argue that the peak was a little later... 1980s to 1990s?

25 *We have changed to "1980s and 1990s". (Page 2, line 2)*

L4. Reword / clarify 'more harvest', more correctly you are referring to the use of forest residues for renewables!

*We replaced "more harvest" with "forest residues".  (Page 2, line 4)*

L7. lab –> laboratory

*We have change accordingly. (Page 2, line 8)*

L7. There was no intensive modelling? Perhaps 'extensive' is superfluous?

*We have removed "extensive" (Page 2, line 8)*

L8. Simplify (here and throughout): 'This paper presents the state...'

*We have clarified the aims (Page 2, lines 8-11) based on the general comments, as described in the answers to the comments above. By doing this, the text has been simplified.*

L9. You jump too quickly into the specific of the results, give the reader a more guided introduction, 'Under the project, we found that... '

*We have now started the presentation of the results with "In this study we…". (Page 2, line 11)*

L10. Variation from what? Data? Methods? Remember the international audience knows nothing of the project!

*We refer to the variation in estimated weathering rates from different approaches. We have now clarified that. (Page 2, line 12)*

L12. Important but the manuscript would greatly benefit from the 'word smiting' of the native English-speaking co-authors, Finlay and Bishop?

*The manuscript was corrected by a professional language editor before submission! Finlay and Bishop have participated in the revision of the manuscript.*

L13. I think this is an important result but the term 'clear imbalances' obscures the implications of the findings. The activities are unstainable.

*We have clarified the text accordingly: "...showed sites where whole-tree harvesting was clearly not sustainable…". (Page 2, lines 14-15)*

L16. Step back and provide greater support... approaches based on the weathering of (observed) mineralogy, such as PROFILE..., provide the most important fundamental understanding of the contribution of weathering to long-term availability of base cations to support forest growth, nonetheless, these approaches should be continually assessed against...'

*We changed to "Based on the research findings in the QWARTS programme, it was concluded that the PROFILE/ForSAFE family of models provides the most important fundamental understanding of the*

*contribution of weathering to long-term availability of base cations to support forest growth. However, these approaches should be continually assessed against other approaches." (Page 2, lines 17-20).*

L19. this point needs further development / clarity

*The two last sentences were replaced with one: "Uncertainties in the model approaches can be further*

5  *reduced, mainly by finding ways to reduce uncertainties in input data on soil texture and associated hydrological parameters, but also by developing the models, e.g. to better represent biological feedbacks under the influence of climate change." (Page 2, lines 20-22).*

Page 3. L1. change acid to acidic throughout

*We have gone through the document and changed accordingly.*

10  L3. remove one 'processes'

*We have reformulated to: "...extensive research examined processes that acidifies and counteracts acidification....". (Page 3, lines 2-3)*

L5. refer to SWAP first, it started before NAPAP (and is more important in a European context)

*We have changed the order of SWAP and NAPAP. (Page 3, lines 3-6)*

15  L7. You need to provide more context for critical loads; it is an effect-based approach for emission reductions, essentially a direct response to the recognition that emissions of sulphur dioxide were causing significant impacts. Notably it has nothing to do with SWAP or NAPAP!

*We changed the first sentence about critical loads to: "In the end of the 1980s, the critical load concept was developed as an effects-based approach for emissions reductions (Nilsson and Grennfelt,*

20  *1988),....". By doing that, we add info about CL being an effect-based approach for emission reductions. By starting with "In the end of the 1980s..." we discouple this sentence from the sentence about SWAP and NAPAP. (Page 3, lines 6-7)*

L9. 'A critical load...'

*We have changed from "The.." to "A..". (Page 3, line 9)*

25  L11. '... critical loads of acidity...'

*We have added "..of acidity". (Page 3, line 11)*

L15. Yes, very true but those of us interested in water barely consider weathering directly...?

*We don´t understand what R1 suggests here. We have not done any changes.*

L16. '... and as such a sink of acidity'.

*We have revised as suggested. (Page 3, line 17)*

L19. You need to differentiate between plot and catchment scale estimates, models such as MAGIC are process-based, and can be used to provide catchment-based estimates of weathering, however, they are fundamentally different to process-based estimates from PROFILE.

*In the changes of the introduction, to meet other review comments, this paragraph has been changed. Now the main content can be found on page 3, line 30-: "Therefore, a number of indirect methods to quantify weathering rates have been developed, e.g. process-based modelling (Sverdrup and Warfvinge, 1993), soil measurements where the depletion of weathering products in different soil layers is determined in order to assess average weathering rates since soil formation (Olsson et al., 1993), and budget calculations where all other parameters except weathering are measured (Lundström, 1990; Jacks and Åberg, 1987; Wickman and Jacks, 1991; Sverdrup et al., 1998)." In the way it is written now, we don´t think it is necessary to mention scales here (we don´t want to make the Introduction longer). The differences between methods are described in the Methods sections.*

L24. I do not completely agree with this. Many jurisdictions were faced with national scale modelling, and the application of simple approaches such as 'skokloster' provide a practical solution compared to the application of a process-based model that requires quantitative mineralogy on a high spatial resolution... more correctly, given the high loads at that time, the uncertainty in weathering was trivial.

*We changed to "The uncertainties in weathering rates were, during the times of high deposition, less important. Therefore, the interest waned in further weathering research that might revise these weathering estimates." (page 4, lines 6-8).*

L29. Was it severity, or a shift in policy to support mitigation of climate change impacts?

*The policies changed when the severity of climate change became fully recongnized, and it was the policy shift which led to higher demand of renewable fuel. The sentence now reads: "As the severity of climate change became fully recognised, policies for mitigation of climate change led to increased demand for renewable fuels, thereby increasing the pressure on forests." (Page 4, lines 9-10).*

Page 4. L1. increase from 25

*Yes it increased again after 2016, it was a temporal dip. The sentence now reads: "Since 2000 in*

*Sweden, the proportion of clearcuts involving whole-tree harvesting has increased from around 15% to 25-35%, according to statistics from the Swedish Forest Agency, except for 2014-2016, when the proportion was temporarily 15-25% due to lower energy prices during that period." (Page 4, lines 11-13).*

5 L3. Substantial

*It now says: "a substantially increased removal". We have not changed since we think that it is correct as it is.*

L5. Depletion methods needs more description... or just exclude such detail for the moment (estimates of base cation weathering...)

10 *The description of the Depletion method comes later. We did not do any changes.*

L11. 'Akselsson et al. (2007) used a mass balance approach (with weathering estimated using PROFILE) ...'

*This part of the introduction has been shortened during the revision. However, we still mention this study: "Accordingly, mass balance calculations, with weathering and deposition as inputs and harvest*
15 *losses and losses through leaching as outputs (Akselsson et al., 2007) (Fig. 1), as well as simplified calculations, where weathering rates are compared with base cation losses through harvesting (Olsson et al., 1993), have been used during the last decades, for forest sustainability assessments." (Page 4, Line 19-22).*

L17. I suggest 'Similarly, the influence of whole-tree harvesting ...'

20 *This part has been removed during the revisions.*

L20. Yes, but only in a Scandinavian context, this has not spilled over into the rest of Europe or north America (yet).

*OK, it is interesting to think about why. We have added "in Scandinavia". (Page 4, line 25).*

L23. Was the conclusion valid? I would suggest the greatest uncertainty was derived from comparing
25 approaches that should not have been compared?

*We agree that it makes no sense to compare weathering rates estimated for different soil depths. That is also what Futter et al concluded, which we write about in the next paragraph. We have not done any changes here.*

L30. It is okay to call out errors. Three approaches? In truth there are two. Mass balance approaches where you indirectly estimate weathering rate (there are also other indirect methods) OR mineralogy-based approaches, often if mineralogy is not available you have surrogate-based approaches but 'at least three approaches' may verge on ridiculous? Weathering is the breakdown of minerals... so what does three independent approaches refer to?

*We are also skeptical about the conclusion about three approaches. However, we have written about that in the discussion, and base it on the results from this study. We think that the discussion is a better place to "call out errors" than the introduction.*

Page 5. L1. Provide background on the depletion method... total analysis regression... the reader need help.

*To not make the introduction too long, we removed the names of the methods. They are presented in the Methods section.*

L3-4. reference to other methods are difficult to navigate...

*Methods names removed, see above.*

L7-8. Combine sentences... reduce words...

*We changed to: "Two other potential sources of uncertainties that have been explored, but that are still widely discussed, were revisited: (1) the role of biological weathering that might generate weathering not included in the current generation of biogeochemical models (Banfield et al., 1999; Finlay et al., 2009; Finlay et al., in review (this issue)) and (2) simplifications relating to base cation exchange and aluminium complexation (Tipping 2002; Gustafsson et al., 2018 (this issue); van der Heijden et al., 2018)." (Page 5, lines 10-14).*

L17. '... by revisiting older w...'

*This part has been removed as part of the simplification suggested by the referees.*

L21. Flows or removals?

*This part has been removed as part of the simplification suggested by the referees.*

*L30. Replace flows with 'sources and sinks'*

*This part is now in the Introduction (Page 3, line 29). We changed according to the suggestion.*

Page 6. L2. Cite 'Warfvinge and Sverdrup, 1992' for PROFILE

*We have changed accordingly. (Page 6, line 22)*

L2. The work of Susan Brantley should be cited here

*We added a sentence and included a Brantley reference. We also included references to other*
*weathering models: "Due to the difficulties in measuring field weathering rates, weathering kinetic*
*haves been frequently studied in laboratory environments (Brantley et al., 2008). Mechanistic modelling*
*of weathering rates, based on laboratory-determined weathering kinetics, is one of the most widely*
*used approaches for estimating field weathering rate (Warfvinge and Sverdrup, 1992; Godderis et al.,*
*2006; Maher et al., 2009)." (Page 6, lines 19-22).*

L7. The key point here, and what separates PROFILE from other approaches, it that weathering is
derived from the breakdown of mineralogy (an essential input), the other inputs only estimate the
amount of minerals that are being weathered.

*We included that in the first sentence: "The PROFILE model (Warfvinge and Sverdrup, 1992) is a*
*steady state soil chemistry model where weathering is derived from the breakdown of minerals…"*
*(Page 6, lines 22-23).*

L13. Again, it might be worth citing Brantley here...

*We didn´t add a reference to Brantley also here, since this part is specifically about PROFILE.*

L17. hydrological model...

*We have changed "hydrology" to "hydrological" (Page 7, line 6).*

L21. The discussion / details on SAFE and ForSAFE can be removed.

*We want to include a description of ForSAFE, as for the other approaches. However, we have reduced*
*the description about the models, in the restructuring of the paper. SAFE is mentioned since it is still*
*more well-known in some contexts, in order to explain how SAFE and ForSAFE relate to each other.*
*(Page 7, line 3-)*

L30. Simplify to PROFILE

*This section has been removed, as part of the shortening of the paper. See answer to general comment*
*#2.*

Page 7 L4. There is an application of SAFE to Hubbard Brook which models the catchment (compared with MAGIC, VSD, etc.)

*Also this section has been removed we now focus on the methods used in this study, in order to get a shorter and more focused paper, according to the referee comments.*

5 L7. Should MAGIC be cited here or under 'mass balance' approaches? PnET-BGC is another example of a model that uses a mass-balance approach

*We actually thought a lot about that. It fits in both chapters. We have thought about it again have now decided to put it under "Budget calculations" (page 9, lines 11-18). We have´t mentioned PnET BGC, since we now only include models that have contributed to weathering rates in this paper (see above).*

10 L17. Assumes that deeper soil is the parent material, so does not work for glaciofluvial soils, etc.

*We will clarify that in assumption nr 2 (Page 8, line 2): "(2) the soil pedon consists of homogeneous, where the deep soil constitutes the parent material,…"*

L26. Could add that the approach has been widely used and cite a few examples?

*We have done that. (Page 7, line 26-27)*

15 L27. Typically referred to as 'Catchment mass balance budgets' as they are widely estimated at the catchment scale, as such the estimates of weathering are an average of a larger landscape unit and can be highly influenced by localised geology.

*The examples in this paper is site mass balance studies. Therefore, we will not change to 'Catchment mass balance budgets'.*

20 L28. MAGIC should really be mentioned in this section!

*See answer to comment on Page 7, L7 above.*

Page 8 L10. Retitle to 'strontium isotope ratio'

*This chapter is now part of the budget calculation chapter, so the title is removed.*

L17. This really should be included under the depletion, as it is a derivative of that approach

25 *Yes, the total analysis regression method is a derivative of the depletion method. However, we decided to keep them separated, but clarify in the beginning that this is the case. The reason is that, in the single site comparison, the total analysis regression methods are in many cases referred to, and in one case (Gårdsjön), weathering rates have been calculated both using the depletion method, and using the total*

*analysis correlations. We now start the chapter about the total analysis regression with "The total analysis regression method is a derivative from the depletion method, which requires much less soil data than the depletion method." (Page 8, lines 9-11).*

L24. Remind the reader that you are focused on QWARTZ, i.e., 'Under QWARTZ, weathering ...'

5   *The restructuring of the paper, to the more conventional division into methods and results, makes this information superfluous. Moreover, we have used data from the QWARTS programme, but even more from earlier studies. Thus, we did not do any changes.*

L30. 'profile 17-20cm deeper', this is a little unclear. I assume in simple terms the soil depth differs between estimates... Maybe present table on a 'weathering per cm'?

10  *We have made major changes in this chapter, and this part is removed. We don´t want to add a new table, since we are trying to shorten the paper, and since the depths are the same for the same site, except for in Gårdsjön and Svartberget, where we have different rows for the different subsites (with different depths).*

Page 9 L2. This is an important point and should perhaps be stated much earlier, homogeneity of soil

15  and bedrock are important considerations for agreement / lack of agreement between approaches

*We think it fits good here, in the very first part of the results.*

L15. Why does the depletion indicate a lower weathering rate? This could suggest that the un-weathered layer did have weathering? In many of the studies in Table 1, the depletion method is lower. Why? However, it may also be argued that the range between methods is smaller than the uncertainty?

20  *We have some potential explanations to this in page 11, lines 24-28. One of them is in accordance with the suggestion above about that the "original till" is actually weathered till from earlier glaciations.*

L21. Do you mean 'soil bulk density'?

*Yes, we have clarified that (page 11, line 11).*

L23. Correct 'to to'

25  *This sentence has been removed in the restructuring and shortening of the paper.*

Page 10 L19. Disqualified? Excluded! However, just exclude, and note in a footnote to the Table. No need to explain

*We changed to 'excluded' and refer to Stendahl et al. instead of explaining in detail. However, we keep*

*it here, not in a footnote in the Table, since the sites are not part of the Table, and a footnote about three other sites feels a bit out of place. (Page 5, line 30).*

L30. Till is the most obvious / likely explanation... remove other excludes / reasons and only present this one.

*We have changed the order, so that the till explanation comes first. However, we keep the others as they also might be contributory explanations (Page 11, lines 24-28).*

L34. What does 'conceptual limitations' mean? clarify

*We agree that this is not a good expression. We have shortened this part, and removed the "explanation", and refer to the discussion in Stendahl et al.(Page 11, lines 30-33).*

Page 11 L1. 'for Ca and Mg...'

*We have added 'for' (page 11, line 32).*

L3 to L10. This can be reduced to one sentence, state 'The majority of weathering rate estimates were classified as acidic or intermediate (cite Table, UNECE, etc.).

*We shortened the paragraph. It now reads: "The intervals of all sites were compared with four reference weathering intervals, based on weathering rate approximations frequently used in the critical load work (Fig. 3; de Vries, 1994; Umweltsbundesamt, 1996). The majority of weathering rate estimates were within or close to the interval outlined for acidic/intermediate parent material with coarse texture." (Page 12, lines 7-9).*

L4. Figure 2 essentially repeats Table 2; the classification could be added to the Table 2 instead.

*We think that the figure referred to, which is now Figure 3, gives a good overview, much better than the table (which is now Table 3), but we also want to share the actual numbers in the paper, so we would like to keep both.*

L10 to L14. This text repeats detail already discussed.

*This part has been removed in the restructuring and shortening of the paper.*

L20. This is a long section, and rather than go through each site (one by one), it would be more efficient to summarise and focus on the broad agreement, and disagree, but describe from the point-of-view of the factors that drive disagreement, e.g. ' ... there was slight disagreement between some estimates owing to difference in input data use by the different approaches, such as soil depth (give example) or

soil moisture (give example). The table is provided, so the reader can evaluate the results, and there is no need to describe in detail.

*We have shortened  section 3.1 abut site-level comparisons substantially (from three pages to just over two). We have added quantitative measures and have removed many f the details.*

5  L22. are they scaled-up or just regional applications (more sites)? How are they scaled?

*It is more regional applications (more sites). However, this formulation was removed during the restructuring of the paper.*

L25 to L33. The relationship between ForSAFE and PROFILE has already been described at length. There is no need to repeat again.

10  *This has been solved by restructuring the paper to a more conventional structure with methods, results and discussion, see more detailed description in the answer to general comment #2. The difference is now described in Methods. (Page 7, lines 15-18)*

Page 12 L7. Is the analysis really only based on 11 sites? Did the study use 11 sites to predict at 400+?

*Yes, this is described in Olsson et al. (1993) which we refer to.*

15  Page 13 L5. Is this the same approach as used with PROFILE? Did both use UPPSALA? Clarify

*No, another normative model was used for PROFILE. However, we have shortened the method description according to earlier comments, and do not include this information in the paper anymore, but instead refer to the papers where it has been described thoroughly, so we don´t add anything based on this comment.*

20  L14. Again, is this similar to MATCH used in PROFILE. Perhaps have a consistent description (and only described in one place in the text)

*Yes, MATCH was used for both. However, due to the shortening of the method description (see above), we don´t include this info in the paper, instead we refer to the papers where it has been described thoroughly, so we don´t add anything based on this comment.*

25  L15. Why compare PROFILE and ForSAFE. Are they dramatically different, or are you just comparing the effect of different hydrological data on estimates of weathering? The justification for this needs to be clearly stated under section 4. This is a very different comparison to total analysis (which is fundamentally a different estimate of weathering).

*We agree that the comparison is very different than a comparison with to completely different*
*approaches. PROFILE and ForSAFE are built on the same weathering reactions, but the dynamics in*
*ForSAFE affects weathering rates. Also, the fact that e.g. hydrology is modelled in ForSAFE, whereas*
*it is input data in PROFILE, may affect weathering rates. This is described thoroughly in Kronnäs et al.*
5 *in the same issue. We have now clarified this in the Methods chapter, page 7, line 15-19: "Although the*
*weathering module is the same in PROFILE and ForSAFE, some differences can be expected since*
*ForSAFE includes dynamics, which means that weathering is affected by other processes over time, and*
*that soil moisture, which is an input in PROFILE, is dynamically modelled in ForSAFE."*
L16 to L18. This detail should be presented under the main part of section 4
10 *This part has been completely changed and shortened in the restructuring of the paper. The information*
*about methodology for regional runs can be found in Methods, on Page 7, lines 20-24.*
L19. Why use climate regions?
*We want to show regional differences, and those regions are logical in the way that they represent*
*different climate and atmospheric deposition regimes. They have been used in one paper, just accepted*
15 *for publication (Belyazid & Zanchi, 2019), and are used also in Belyazid et al (this issue).*
L24. Above you have noted that the differences in estimates of weathering is often driven by differences
in inputs (under different applications) for the same model. Here you add further confusion to that
issue... What is the goal of the comparison?
*The goal of the comparison was not to run different approaches on many sites with exactly the same*
20 *input data. The goal was to collect published weathering estimates from different approaches, in many*
*cases run independently of each other, and compare, to get a span covering both differences in*
*approaches and differences in input data used. This represents the reality for e.g stakeholders. We show*
*that, even though we do like this, we can draw conclusions about sustainability. This is clarified in our*
*new first aim: "(1) investigate the variation in published weathering rates from different approaches in*
25 *Sweden, under consideration of the key uncertainties for each method,"*
L29. Above you state they are more-or-less the same, and since that statement we find that one is higher
than the other, and so on... which is the truth?
*We wrote that there were "no large systematic differences between the medians or ranges for the*

*different methods". And that "ForSAFE gave somewhat higher medians than PROFILE for all regions, especially in the northern regions." We don´t think that it is contradictory, it is still no large systematic differences. However, we agree that it could be better formulated Now we start the chapter about regional level comparisons with: "The weathering rates for the nutrient base cations Ca, Mg and K*

5 *varied widely within the regions for all methods, but there were no large systematic differences between the medians or ranges for the different methods (Fig. 4-5). However, PROFILE gave generally somewhat lower weathering rates than ForSAFE and the Depletion method/Total analysis regression approach, with overall medians of 14.3 mekv $m^{-2}y^{-1}$ (PROFILE) and 17.8 mekv $m^{-2}y^{-1}$ (ForSAFE and the Depletion method/Total analysis regression approach). " (Page 12, lines 15-19).*

10 L33. Maximum weathering depth? This is confusing, why compare if they represent different depths / pools? This needs clarification, the text suggests that PROFILE covers a shallower depth compared with Total Analysis. They why compare? Normalise both to the same depth before comparing.

*In this paper we have used results already published. We have decided not to try to normalize them, then we would add extra assumptions (e.g that weathering per cm is constant in the soil profile, which*

15 *we know is not true). We know e.g. that the weathering rates according to the depletion method are much slower further down in the soil profile. We have thus decided to discuss the differences in depths instead of normalizing.*

Page14 L6. Reword... what comparison between regions?

*Here we mean the comparison of weathering rate intervals between regions. The sentence now reads:*

20 *"In most cases, the width of the weathering rate intervals showed no major differences between the regions." (Page 13, line 5)*

L8-L9. This statement, and similarly many of the statements in the previous section, are very qualitative. There is no quantitative element to the comparison at the site or regional level. Statements such as broadly agree, similar / non-similar are fine IF they are also supported by a quantitative

25 assessment. This is missing.

*We have now quantified differences in the chapter 3.2 Weathering rate comparisons on a regional level. For example, we compare medians between the approaches and regions. See also answer to general comment #4, regarding similar changes in other parts of the paper.*

L10+. This has been already stated, and is obvious, it should be noted in the main part of section 4, and not included as part of the comparison (it was known before starting the comparison). However, it would be useful to know the purpose for such a comparison?

*See answer to comment regarding page 13, L15.*

L15. I think (more-or-less) the results of this assessment are stated here, as such, perhaps the whole assessment could be collapsed to one paragraph?

*We have substantially shortened chapter 3.2, as part of the restructuring and shortening of the paper.*

L27. If it is already described, then why repeat here?

*We have substantially shortened the chapter about biological weathering, focusing on implications of insufficient process descriptions for modelling results, in line with the comments from Referees and Editor.*

L30. wording 'dependent'

*Chapter 3.4 about Biological weathering has been very much shortened and this formulation has been removed.*

Page 15. L1-L5. Citations?

*We added the reference Sverdrup and Warfvinge, 1993, where this is gone through more thoroughly (Page 15, line 11).*

L11. The weathering process in safe is not directly affected by biological processes, it is only affected in as far as the recognition that some of the processes are likely influenced by biology.

*Biological processes do affect weathering in PROFILE, SAFE and ForSAFE through the following pathways: 1- uptake reduces cation concentrations, directly alleviating the brakes on weathering, 2- transpiration reduces water availability, thus limiting one of the four weathering kinetics (dissolution in water), and increasing concentrations, which can have positive (in case of acids) or negative (in case of cations) effects on weathering, 3- cation uptake produces more protons in the soil solution, lowering pH and promoting weathering, and 4- the production of dissolved organic radicals through litterfall decomposition and in the case of ForSAFE exudation contribute directly to one of the four weathering kinetics. Through these four pathways, the models do directly modify weathering rates based on the outcome of biological activity as described here.*

L12. PROFILE has some biological feedback? Really?

*See answer to L11.*

L33. 'There is extensive literature...'

*This part has been removed when the chapter about biological weathering was shortened.*

5   Page 16. L1 to L34. The entire page (and some of the previous) presents a good review of the 'state of knowledge' but it can be much reduced... and the benefit / objective of such a review should be considered... why cover so much text if this if not part of the work under QWARTS described by the authors.

*We have shortened this chapter substantially, see answer to general comment ¤1.*

10  Page 17. L24. 'We found'? Which 'we'?

*This part has been removed when the chapter about biological weathering was shortened.*

Page 18. L4. Was this stated already?

*We have shortened the chapter substantially and got rid of repetitious text.*

L10. What does the section title mean?

15  *We changed the title to "Implications of improved model descriptions of base cation exchange and aluminium complexation", which is similar to how it was phrased in the Introduction (Page 16, line 19).*

L17.Why 'state-of-the-art'. WHAM has been around for decades?

*We removed 'state of the art'. (Page 16, lines 26-27).*

20  L22. Replace 'former' with specific term.

*We did that: "...but for a long time, the simpler ion-exchange equations have been more widely used..." (Page 16, lines 31-32).*

Page 19 L10. I wonder if this is some of the context for this manuscript that would be better to present at the start?

25  *In the restructuring, the chapter about Weathering in a sustainability perspective is focused on the results from the sustainability assessments. The first part where it was introduced has now been moved to the Introduction. (See e.g Page 4, line 16-).*

L20. 'In regions where weathering rates...'

*This part has been removed in the restructuring and shortening of the paper.*

L23. Clarify or reword 'data on site index could be found'

*We changed the word "site index" to "site quality" This part has been reformulated in the restructuring and shortening of the text. (Page 9, lines 30-34).*

Page 20 L7. Differences in weathering has already been discussed?

*We have shortened the chapter about Weathering in a sustainability perspective, but we need to mention the differences also here, since the sites with large differences are the ones where it is more difficult to draw conclusions about sustainability.*

L10. This sounds like it was stated already?

*No, this is the first time we compare weathering and WTH in the north. However, after the restructuring and shortening of the text, it is hopefully more easy to follow.*

L21. The preceding text could be summarised much more succinctly.

*The text has been restructured, so this chapter now only includes results, whereas the methods have been moved to the "Methods" chapter, and the first part to the introduction.*

L31. This is more-or-less the summary of the results (if a further quantitative description was added) then this would be sufficient.

*We have shortened this chapter and added quantitative measures.*

Page 21. L12 to L18. This has nothing to do with future, It is mostly repetitious text.

*See answer to general comment# 3.*

L22. This section seems odd... we have just been presented with a 2+ page section that covered biological weathering. It is difficult to justify this additional text! I believe this section should be removed, and any 'fresh' text be included above.

*We have shortened the chapter about biological weathering substantially and we have replaced the chapter "future research", with "Prospects for method development". The first subchapter in "Prospects for method development" is about "PROFILE/ForSAFE – Development of model descriptions" There we included one sentence about improvements related to biological weathering (Page 18, line 2-7).*

L23. Yet despite the previous lengthy discussion on biological weathering, we were not introduced to the term 'EPS'??

*We have now introduced the term extracellular polysaccharides (EPS) (Page 15, lines 23-24)*

L23 to L4 Page 22. This text can be deleted.

5 *This chapter has been merged with the chapter about Potential for biological weathering, and the text in those chapters has been very much shortened.*

Page 22 L5 to L11. Is this model development or uncertainties?

*In the restructuring about the Biological weathering part, this chapter has been removed, and we now only include one model development sentence about biological weathering in the chapter "Prospects*

10 *for method development", that has replaced "Future research".*

L24. Again, it is difficult to justify such an extension section, shortly after we have already been presented with a discussion on the topic.

*In the restructuring of the paper, this chapter was removed, and we now only include one short paragraph about this in the chapter "Prospects for method development".*

15 L25 to L32. This is not model development...just repetitious text

*See answer to Page 22, L24 above.*

Page 23 L1. This is a trivial point, with the right implementation the sped can improve. Just because it takes an hour to cycle to work, does not mean that everyone must cycle!

*The organic complexation does not have an arithmetic solution, the only way to do it without*

20 *compromising its purpose is by optimization (of multiple complexation parameters) and a whole set of assumptions (including that the Al pool is constant for example), i.e. iterations until near-equilibrium. In HD-Minteq this is feasible because most state variables and fluxes (soil organic matter, decomposition, uptake…) are given as input and therefore not dependent on changes in soil solution chemistry. While in ForSAFE these processes are internally simulated, meaning that each iteration of*

25 *Al complexation will require an entire simulation of all fluxes and equilibria, exponentially increasing computation time or even requiring a new optimization of exchange parameters, which in the word of the original author "is impossible" (see Gustafsson 2001, Journal of Colloid and Interface Science 244, 102-112. doi:10.1006/jcis.2001.7871).*

L8 to L11. I am sure this is repetitious text.

*This text was removed.*

L14. ' overestimated estimates of weathering rates' –> ' overestimated rates of weathering'

*This was changed accordingly (page 18, line 15)*

5   L19.So it was not tested? Is this future research? It seems to be more of 'ongoing' research?

*New brakes have not been implemented and tested in PROFILE/ForSAFE. This is future work. In Erlandsson et al., we implemented and tested silicate release through weathering and the feedback brakes from Si concentrations on the four weathering kinetic equations, while dictating other inputs such as uptake, water flow and so on. This has produce the expected results (constraining weathering in*

10   *the saturated zone). Implementation in PROFILE/ForSAFE has not yet been carried out. The chapter "Future research" is now replaced with the chapter "Prospects for method development".*

L20. Is this catchment scale weathering?

*Yes, however this heading has been removed.*

L21 to L27. This is not future research, this is improvements needed in the application of weathering

15   models (and better linkages with hydrological models). As such, uncertainties or limitations is a more appropriate section.

*We thought about, but didn´t think that "Uncertainties" was suitable. We want this chapter to include advices based on the QWARTS program, about how to continue to reduce uncertainties. We have renamed the chapter: "Prospects for method development", which we think fits with what we want to*

20   *include. We have also removed or moved parts that do not fit in.*

L28 to L4 Page 24. Not future... ongoing /current research?

*See answer to comment L21-L27 above.*

Page 24 L5. See comments above... much of the text presented so far under 'future research' speaks more to limitations in application or uncertainties.

25   *See answer to comment L21-L27 above.*

L6. This is PROFILE specific text... not the depletion method, or others...

*The chapter is now called: "PROFILE/ForSAFE – Improving input data quality"*

L8. Often?

*This sentence has been removed in the shortening and restructuring of the paper.*

L9. 'Not only are...'

*We corrected this (Page 19, line 5).*

L13 to L20. This is a very uncertain uncertainty... why so much space for something that is not 'very uncertain'?

*It is part of the QWARTS work and refers to one of the papers in the special issue. So we want to refer to it, but we have shortened the paragraph (Page 19, lines 6-15).*

L24. Unless that span is used to estimate the uncertainty...

*It is used for that in one of the papers of the special issue (Casetou Gustafson et al.). But still, often one value is used.*

L26. This does not make sense. Is it possible to contain the solution space?

*The reviewer is right that this is not a problem if we have information on which minerals are present, their respective stoichimetries and in what proportions (See Posch and Kurz, 2007). This however is most often missing, and that is the purpose of using A2M. It would, as indicated, be possible to narrow the space in more information is available, on for example the fraction of light primary minerals in the soil. So yes, it is possible to constrain the space if a considerably more information is available.*

L28. This is not true. However, many users make that assumption. However, others do not.

*In the cases we have seen, where one of the mineralogies is chosen from A2M, it has been the centre point, although everyone who uses A2M should know that all solutions are as likely. We think that we have been clear about that all have the same probability, but that many choose the center point, since they want one value.*

Page 25. L1. BET may still be the best technology?

*We removed "using modern technology". Even if the BET technology hasn´t developed since the 80/90:ies (which we don´t know), the regressions could be revised based on a larger soil material (Page 19, lines 26-28).*

L2. Are the 'current uncertainties (?)' quantified? Are there uncertainties?

*The regression graphs in Warfvinge and Sverdrup (1995) indicate large uncertainties. On page 19,*

*lines 27-28, we now write: "The regressions reveal that the uncertainties are large. Revisions of the*
*regressions, based on a larger data material, could reduce the uncertainties."*

L5 to L9. Repetitious text.

*This is now removed from here, instead we focus on the need for better soil moisture quantification in*
5  *the models: "The soil moisture is one of the most important factors that introduces large uncertainties*
*in the results, both in PROFILE where it is an input (Rapp and Bishop, 2003), and in ForSAFE where it*
*is modelled based on hydrological parameters (Kronnäs et al., 2019). Improved input data quality for*
*soil moisture would substantially reduce uncertainties in PROFILE and, even more importantly, soil*
*moisture modelled by ForSAFE needs to be evaluated, and the sensitivity to soil input data needs to be*
10  *examined." (Page 19, lines 29-33).*

L10. Improved soil moisture should come with improved soil hydrological modelling...

*Yes, and since this chapter is about improved input data we mention soil data which is very important*
*for the hydrology modelling in ForSAFE. We don´t understand what R1 suggests here.*

L12. All estimates are modelled? What are the other?

15  *This chapter was removed in the restricting and shortening of the text.*

L13 to L17 Page 26. This whole section can be deleted. Any useful should be moved to the section on
comparisons. This is not 'future research'

*This chapter was removed in the restricting and shortening of the text.*

L23. Manual? This is a bit trivial... delete and move to personal 'to do' list. I suggest you write a paper
20  on this.

*We don´t mention to propose a manual any more.*

L28. They were not outliers. They represent measurements for different compartments. This is well
understood.

*Mass balance methods are still used for weathering rate calculations and advocated by many. We*
25  *included it here and came to the conclusion that it was problematic in the cases we had. We write "In*
*the compilation of weathering rates in this paper, the most extreme outliers came from the budget*
*method, which can be explained by the fact that the sources and sinks are not described in a completely*
*accurate way (Rosenstock et al., in review (this issue)). For a fair comparison between weathering rates*

*from the budget method and from other methods, ways to distinguish between different sources and sinks need to be further developed. We think that this is a rather strong recommendation, which we would like to keep as it is.*

Page 26. L10. This is wrong. It is okay to state that. There independent methods? How many methods are there (truly independent)? More correctly, Futter et al. (2012) should have recommended that a method incorporating soil mineralogy be used (all other approaches are surrogates for weathering).

*We think that we are rather clear when we conclude that it is unrealistic. We think that it is stronger to say that, than to say that is absurd or wrong, which are more "value" words.*

L16. Good but you can more clearly call it out as an absurd suggestion.

*We think it is enough to say that it is unrealistic in practice.*

L22. Was some of this difference on single sites driven by differences in depths / inputs?

*Yes. We have changed to "Although the variation in weathering estimates was large on single sites…". By doing that we include methods as well as input data and to some extent depths. (Page 20, lines 30-31).*

L21 to L30. I agree that these are the primary conclusions from this work; I would urge the authors to reflect on this when revising the manuscript. Much of the text can be reduced and streamline to better present this issue (conclusion).

*We agree and our revisions have been done with this in mind, see further answer to general comment # 1-3.*

Page 27. L10. Other approaches?

*Yes, the last sentence in the Conclusions now reads: "However, it is also important to continue to compare with results from the Depletion method and the Budget approach."*

**Answers to comments by Referee 2 (R2)**

*Comments from 19 March 2019*

R2 starts off with a paragraph with general comments, describing the paper and its strengths. In this
paragraph there are no questions or suggestions of changes. After the general comments, a few specific
comments are listed.

**Specific comments**

1a. "The consistency of the weathering rates estimated by different methods at the same sites in this
study was remarkable. Older studies frequently reported one or two order of magnitude differences in
estimates of weathering by depletion versus budgets………… Have weathering rates changed
substantially in the last 30 years, or were the differences always muted in Swedish soils?"

*The main aim of this study was to investigate the variation in weathering rates from different
approaches in Sweden, and assess the results from a sustainability perspective. To do that, we searched
for Swedish sites where weathering rates had been estimated with more than one method, to the same
depth, and where the method and data was well described. We didn´t exclude any sites where those
criteria was met, but included all old and new studies that we could find in literature. We did not find
any sites with differences of one or two order of magnitudes between weathering rates estimated with
the depletion method and the budget method, which according to R2 was frequently reported in older
studies. We didn´t find those differences when comparing other methods either. We pointed that out in
our first answer to the review comments, and then got another interpretation of the question than our
own from the Associate editor. We answer her comments under the heading "Answer to Associate
Editor Decision". Regarding the question from R2 about whether weathering rates have changed
substantially in the last 30 year, we don´t think that, and the dynamic modelling with the ForSAFE
model by Kronnäs et al. (this issue), does not indicate any large changes.*

1b. "Following upon the previous comment, have the authors tried plotting the estimated weathering

rates against one another? For example, PROFILE vs. Depletion, etc.? Perhaps this is in one of the other papers in the series?"

*In Stendahl et al. (2013) weathering rates from the depletion method was plotted against PROFILE weathering rates on 16 sites. The number of sites in that study is enough to be able to draw general conclusions about differences in weathering rates from the two different methods. In the present study, those sites are included, but also other sites, based on different combinations of methods. Based on the material in this paper, it is not as straight forward to do complementary pairwise comparisons, since most of the approaches (all except PROFILE and the depletion method that are already compared) are performed only on a few sites. Instead we have focused on presenting the weathering rate intervals for each site, to illustrate how much they differ.*

3. "When comparing weathering rates to harvesting removals, what rotation length was assumed to calculate a meq/m2/yr value?"

*We use site quality to calculate harvesting removals, in the same way as in Akselsson et al. (2007: 2016) and Stendahl et al. (2013), which we refer to. Site quality (in Swedish "bonitet") is the average yearly biomass growth on a specific place during a forest rotation, if the forest is managed optimally. In our calculations we assume, like in earlier studies, that the actual growth is 80% of the optimal growth. The average growth in the site quality concept "assumes " that the forest is harvested at the optimal point in time, which varies between north and south and between stands, from about 70 years in the south to 120 years in the north. We have now explained more thoroughly how the harvesting losses were estimated (Page 9, lines 32-34).*

4. "An awful lot of confidence is placed in this paper in modeling approaches in general and in the PROFILE/ForSAFE family of models in particular They are good models conceptually and they produce results that appear to track the results from other methods (but see comment 1b above). The problem is that there are no "measured" values of weathering flux to use to validate these (or any) weathering models. So, just as budget-based approaches may be contaminated by non-weathering fluxes like net losses from exchange sites, and depletion approaches may suffer from invalid assumptions,

model results almost certainly contain a host of errors. This is addressed somewhat in the paper. Depletion methods and budget methods are based on field observations and data. With all their flaws, they are, at least in my view, fundamentally stronger than model results. To compare them as equivalent approaches is problematic."

5 *We agree that the depletion method and the budget method, that are based on measurements, are very important in framing weathering rates, which we also point out in the paper. However, we don´t think that the results from those methods are fundamentally stronger than the model results (which actually also are based on measurements to a large extent, e.g. laboratory measurements). On the contrary, we think that further studies are required to get more robust results from those methods – something that*
10 *we point to in the "Prospects for method development" chapter. In this paper we have tried to show how far we can go with the available publications. In the revised version, we have a chapter called "Prospects for method development" on page 17-20 (modified from the chapter "Future research" in the old version). There we discuss uncertainties for all methods, and how to reduce those uncertainties.*

*Regarding budget calculations our results show that the most extreme outliers came from the budget*
15 *method. We explain that by the fact that other sources than weathering are included as discussed in Rosenstock et al. (in review, this issue), as well as the documented uncertainty in terms used in the mass balances. We conclude that "for a fair comparison between weathering rates from the budget method and from other methods, ways to distinguish between different sources and sinks need to be further developed."*
20 *The depletion method gave unrealistically low weathering rates in some cases. On one of the sites, Asa, that was studied in detail within another paper in this special issue (Casetou Gustafson et al., in review, this issue), the analysis of the soil profile indicated that the soil profile has been disturbed, introducing errors in the weathering estimates. We highlight the importance of excluding sites with disturbed profiles as well as to perform studies where the average weathering rate since the last glaciation is*
25 *related to the present weathering rates are required, to be able to make necessary adjustments of the historical rates.*

Answers to comments by the Editor, Professor Suzanne Anderson

*Comments from 19 March 2019*

**General comments**

5   "Overall, the manuscript is well written and clear, although several sections are longer and a bit more tedious than others. In particular section 5 on "Potential for biological weathering", and section 8 on "Future research" could be streamlined."

*We have reworked the chapter "Potential for biological weathering", according to the descriptions in the "Answer to comments by Referee 1". The chapter "Future research" overlapped to some extent*

10  *with the chapters "Potential for biological weathering" and "Implication of higher resolution of chemical reactions in weathering modelling" and has now been restructured, renamed ("Prospects for method development"), and shortened from more than 5 to just above 3 pages. The paper has also been restructured to follow a more traditional outline (with methods, results and discussion), see further Answers to comments by Referee 1.*

15  "The challenge addressed in the manuscript is whether the losses of nutrients due to forest harvest can sustainably be balanced by release of nutrients by chemical weathering. This simple mass balance could be illustrated with a conceptual diagram showing the relevant fluxes at the beginning of the manuscript. Such a diagram might help focus on the important sources of uncertainty in long-term predictions of sustainability."

20  *We have added a figure (Fig. 1) in the Introduction, showing the relevant fluxes in the simple mass balance of base cations in the forest ecosystem (weathering, deposition, harvest losses and leaching), to put our weathering results and the sustainability assessments in a context. We introduce the Figure in the Introduction.*

"The most important question the manuscript, and indeed the QWARTS program addressed is if

25  weathering rates, as computed by different models, are greater than or less than rates of harvest loss. The range of weathering values from different models gave a sense of uncertainty in weathering rates, but harvest losses were presented as a single value at each site. There must be uncertainty in export losses due to harvesting, and it seems important to convey what these uncertainties are."

*We agree that the sustainability of harvesting is an important part of this paper, and that the uncertainties in the harvesting estimates should be mentioned. Research related to those uncertainties were not included in QWARTS, which focused on the actual weathering rates, but we have added a paragraph about the uncertainties in the chapter "Weathering in a sustainability perspective", based on results from other studies, e.g. Zetterberg et al. (2014) in Science of the Total Environment. (Page 14, lines 14-28).*

"Two aspects of weathering that I would have expected to be important (and that might be highlighted in the conceptual diagram I suggest above) are rock type and depth of weathering. The first of these is perhaps simpler to consider. Rock type or soil substrate was not described anywhere or for any site. I could not tell if this was because all of Sweden has the same rock type, and so is dismissed as playing any role in variations in weathering rate or nutrient release, or if something else was going on. I would expect rock type to be the very first control on weathering rate, as nutrient availability on basalt vs limestone vs serpentinite vs : : :. is quite different. The depth of weathering is a more difficult problem to address, yet could also be important. Weathering often occurs at depths below the top 30-50 cm."

*Yes parent material is of key importance for weathering rates. We have included a table (Table 2) describing the mineralogy used as input in the modelling for each sites, as a measure of the parent material. For the sites in which some of the authors have been involved in the weathering estimations, we have thorough databases from which we can get the information. For the other sites, we have extracted information from published papers. We refer to Table 2 at Page 5, line 29.*

*We have focused on the root zone in this study, since we are interested in the sustainability of forestry (clarified on Page 3, lines 23-24). Therefore, for this study we see the depth of the rooting zone as more interesting that the weathering depth, therefore we haven´t discussed the weathering depth.*

*In Swedish applications the depth 50 cm is often used for the rooting zone, based on studies in Sweden. In this paper, where one important aim is to compare different approaches, we have included sites where weathering rates have been calculated to deeper depths, up to 1 m, but we have only accepted sites where the different approaches have been run to the same depths. In the sustainability calculations only sites with depths<0.7 m are included. This is described in the Methods part (Page 9, lines 28-30).*

*We address the uncertainties related to a fixed root depth, with reference e.g. to Hodge, as suggested in the Associate editor decision (Page 14, lines 17-19).*

**A few detailed comments:**

Acronyms should be defined at their first use. Several are not defined at all, or only after their first use (this list may not be exhaustive): UNECE, CLTRAP, EMF, A2M

*We defined UNECE and CLRTAP the first time it was mentioned. We skipped the acronym EMF, and instead wrote Ectomycorrhizal fungi on the places were the acronym was used before. For A2M we did not do any changes. It is the model name, and when we introduce it we explain what it does and give a reference (just like for PROFILE and ForSAFE).*

Whole-tree harvesting is defined on p3 as being "harvesting of branches". Does this imply that stems are not included in whole-tree harvesting? This seems contradictory.

*Stems are included in whole-tree harvesting. We have gone through the paper and clarified that, where required.*

Total-regression analysis: what is the temperature sum?

*The temperature sum is a measure often used in forestry. It is the daily mean temperature above a threshold value, in Sweden often +5ºC, summed during the growing season. Before we only referred to Morén and Perttu, now we added an explanation (Page 8, lines 13-14).*

Figure 5: The boxes in the key are so small that they are indistinguishable. The gray patterns all look very similar.

*This figure (which is now Fig. 6) has been improved, by increasing the size of the legend and changing the patterns so that they can be more easily distinguished from each others. Fig. 7 has been changed in the same way.*

p 8, line 5: Casetou-Gustafson's name is incomplete.

*Corrected, on all places in the document.*

p10, line 4: cation, not carion

*Corrected, in the revised document it is on page 10, line 23.*

p 12, line 9: Ca, Mg and K at 640 sites, not Ca, Mg and K on 640 sites

5  *Corrected (although the exact formulation has changes in the revised version), page 8, line 22.*

p. 17, line 2: to sparse grass, not via sparse grass

*The whole paragraph has been removed, to shorten the chapter about biological weathering.*

p. 17, line 2: naturally lead-contaminated, not natural, lead-contaminated.

*The whole paragraph has been removed, to shorten the chapter about biological weathering.*

10  p. 18, line 23: However, in 1996..., not However, already in 1996..

*This has been changed according to the suggestion, see p. 16, line 32.*

Answer to Decision by Associate Editor (Professor Suzanne Anderson)

*Decision from 10 May 2019*

Along with her decision about major revisions, added a few comments, which we reply to here.

1. "The proposed strategy of reorganizing the manuscript along more traditional "methods" and "results" lines seems useful, especially as it will reduce redundancies in the text. I look forward to seeing a new conceptual diagram in the introduction to further aid in this streamlining."

*We reorganized the paper accordingly, which reduced redundancies substantially shortened the text*

10 *substantially, from 37 pages to 29 pages (including first page and references, but not figures and tables). We also added a new conceptual picture, Figure 1, based on the suggestion, and referred to it in the Introduction, to create a better basis for the sustainability assessments.*

"Referee 2's comment 1, expressing surprise at the remarkable consistency in weathering rates between

15 methods reported in this manuscript is I believe a comment aimed at what is commonly referred to as the "field-lab discrepancy" in weathering rates. For a review of methods of measuring weathering rates, and a short discussion of this controversy, I recommend reading **Riebe et al. (2017)** Earth Surface Processes and Landforms (doi: 10.1002/esp.4052), in particular p. 133-135. I do suggest that this comment be addressed, as it is relevant. This topic also applies to Referee 2's comment 4."

20 *We interpreted Referee 2´s comment 1 about "one or two order of magnitude differences in estimates of weathering by depletion versus budgets" in "older studies" as a comment directed towards the depletion method and the budget method. Here it is instead suggested that it addresses the "field-lab discrepancy", discussed e.g. in Riebe et al. (2017). We find it difficult to go in to a discussion about that in this paper, since the aim with the paper was to investigate the variation in weathering rates from*

25 *different approaches in Sweden, and assess the results from a sustainability perspective, not to describe the development of each method and discuss them in detail. We included all Swedish sites that we could find in literature, old and new, that met our criteria (that at least two approaches for estimating*

*weathering rates have been applied, that the estimates have been done to the same depths, and that data and methods are well described). We don't think that discussions about things not related to the results from the analysis of weathering rates from those sites fit in the paper.*

*Regarding the inclusion of lab data in PROFILE: It is described in books and papers, e.g. "Sverdrup*
5 *(1990): The Kinetics of Chemical Weathering. Lund University Press, Lund, Sweden and Chartwell-Bratt Ltd, London, ISBN 0-86238-247-5" and "Sverdrup, H., Warfvinge, P. 1993. Calculating field weathering rates using a mechanistic geochemical model–PROFILE. Journal of Applied Geochemistry, 8:273–283." In the first, the experiments for different minerals are reviewed, evaluated and discussed, and the use of them in PROFILE is described. In the latter, the field-lab discrepancy is highlighted,*
10 *starting with the sentence: "The rate coefficients published here are sometimes different from rate coefficient values found in the literature due to the division of the rate between different reactions, and the consistent determination of the exposed surface area." We think that a discussion about that is outside the scope of the paper.*

15 "Finally, on the focus on a narrow rooting zone in this study (one of my comments), I think a comment on the assumption inherent in this focus is worthwhile. Root systems respond to nutrient availability, so if the 30-50 cm depth is depleted, they may grow deeper. The question is to what extent root systems are static versus plastic. You might want to at least acknowledge that thinking of the depth of nutrient access by roots as a static parameter is an assumption. See Hodge in doi:10.1016/B978-0-12-409548-
20 9.05232-5

*We clarified that we work with the rooting zone in the introduction (Page 3, lines 23-24), and we added a discussion about uncertainties related to the assumption of a specific rooting depth, with references, e.g. to Hodge, in the chapter "Weathering in a sustainability perspective" (Page 14, lines 16-19).*

[revised manuscript text omitted]

---

## Author Response (AR2)

**Answer to Associate Editor,**
**and description of how the comments were handled**
*Cecilia Akselsson, 27 September 2019*

5   Associate Editor Suzanne Anderson asked for minor revisions (13 september 2019), after the thorough revision of the manuscript in response to the reviews. We have gone through all her comments in the manuscript, and describe how we have handled them below (page 2-4).

We have also checked the manuscript for "typos, missing co-authors and their affiliations, terminology, updates of data in tables, or updates of variables in equations" as requested. All changes are listed after the answers to the comments by the
10   Associate editor in this document (page 4-6).

This paper refers to published as well as not yet accepted papers in the special issue. We were not sure how to refer to them. We have treated the published papers as any other paper (e.g. not indicating that they are in the same special issue). The accepted papers are:

-Casetou-Gustafson et al., 2019
15   -McGivney et al., 2019
-Kronnäs et al., 2019
-Gustafsson et al., 2018

We have referred to the not accepted papers, as "XXX et al., in review". In the reference list, we have written "in review (this issue)", instead of a year. The not yet accepted papers that we refer to are:

20   -Rosenstock et al., in review
-Belyazid et al., in review
-Sverdrup et al., in review
-Finlay et al., in review
-Erlandsson Lampa et al., in review
25   -Casetou et al., in review

The way we refer to the published and not yet accepted papers in the special issue should be adjusted to the journal standards, and we would need help on that.

**Answers to comments by Associate Editor**

The page and line numbers for the comments refer to the version in which the comments were made by the Associate Editor. The page and line numbers we refer to when we describe how we have addressed the comments apply to the revised manuscript with "track changes" in this document (page 7-49). All answers are written in italics, in order to be able to

5 separate between comments and answers.

Page 3, line 22-23: runon – confusing
*We don´t fully understand the comment, but we agree that the sentences are confusing. We have now changed to "Finally, weathering is the long-term source of base cations, depending largely on soil mineral content and soil texture. In areas that have been covered by ice, like Scandinavia, the soil properties at a specific site depend on the parent material from which*

10 *the glacial till originates." See page 9, line 22-24.*

Page 3, line 30: Use of e.g. is a little heavy. There are other ways to make the point.
*We simply removed "e.g." See page 9, line 31.*

Page 5, line 28: quartz
*Typo corrected accordingly. See page 11, line 29.*

15 Page 5, line 28: mafic minerals
*We changed from "dark minerals" to "mafic minerals". See page 11, line 29.*

Page 6, line 24: chose one verb!
*We corrected this typo by removing "modelled" and keeping "calculated". See page 12, line 24.*

Page 7, line 30: I would call it "recalcitrant", rather than inert.
20 *We replaced "inert" with "recalcitrant". See page 13, line 30.*

Page 15, line 10-11: Did Sverdrup and Warfvinge discover this? Seems like an odd citation for this truism.
*We agree. The reference was meant to apply to the whole paragraph, but we now moved it, to after the first sentence in the paragraph, where we think it fits better: "Although the four weathering pathways, upon which PROFILE is built (the reaction with $H^+$, $CO_2$ and DOC) are chemical (the dismantling of mineral matrices by charged or dissolving particles to*
25 *produce free elements), their drivers are strongly dependent on biological activity in the soil (Sverdrup and Warfvinge, 1993)." See page 21, line 10-15.*

Page 15, line 22: Citation? You seem to be pointing to a particular paper.
*We have added references: McMaster, 2012; Gazzè et al., 2013, 2014; Saccone et al., 2012. See page 21 line 27-28.*

Page 16, line 17-18: word missing?

*Yes, we have added "mycorrhizal fungi". See page 22, line 23.*

Page 18, line 8: Not in references

*Here we refer to "Gustafsson et al., 2018", and we have changed accordingly. See page 24, line 12.*

5 Page 19, line 1: Is this an acronym? If it is, please spell it out. If it is not, then you'll need to introduce it, for instance: "...
estimated from total chemistry with a model called A2M (Posch and Kurz, 2007)."

*Yes, there is a longer name, although it is not used a lot. We have now changed to "Mineralogy inputs to
PROFILE/ForSAFE are often estimated from total chemistry with the A2M model ("Analysis to Mineralogy", Posch and
Kurz, 2007), since direct mineralogy measurements are not available on most sites." See page 25, line 8-9.*

10 Page 21, line 3: should be "conceptual"

*We have changed accordingly. See page 27, line 10.*

Page 30, Table 1: Might be useful to show which methods were applied (or available) for each site.

*We added a column containing that information. See page 37.*

Page 31, Table 2: What are the units?  wt%?

15 *Yes, we have changed to "Mineral content in soil at 50 cm depth (weight %)…" See page 38.*

Page 31, Table 2: quartz

*Typo corrected accordingly. See page 38.*

Page 32, Table 3: Spell out-- I don't know this abbreviation

*We have now changed to "Total analysis regression". See page 39.*

20 Page 34, Table 5: What is CH?  What is WTH? What is PROFILE?  (thought it was weathering model, not "Harvest losses")

*We added explanations in the Table caption "Weathering rates estimated with different approaches and harvest losses at
conventional harvesting (CH, only stems) and whole-tree harvesting (WTH, stems and branches) (meq $m^{-2}$ $y^{-1}$)."
Furthermore, we changed the column name "Weathering" to "Weathering rates estimated with different methods". After
those changes, it should be clear that PROFILE is a method for calculating weathering rates. See page 41.*

Page 38, Figure 4: Seems to have many more sites than shown in Fig 2.

*Yes, it is, this is the regional approach. To make this more clear we made a few changes:*

*-In the caption of Figure 4, we changed "Weathering rates calculated with…" to "Weathering rates on a regional scale on 346 National Forest Inventory (NFI) sites, calculated with…". See page 45-46.*

5 *-In the caption of Figure 2 we changed "Sites where weathering rates have been calculated…" to "Well-investigated sites where weathering rates have been calculated...". See page 43.*

*-We changed the first sentence in Methods from "Weathering rates from different approaches were compared on a site-level and on a regional level." to "Weathering rates from different approaches were compared on a site-level on well-investigated sites, and on a regional level on a larger number of sites but with more generalized input data." See page 11, lines 24-25.*

10  ## Other minor changes

The page and line numbers we refer to apply to the revised manuscript with "track changes" in this document (page 7-49).

-On Page 9, line 26-27 we changed from "Due to its central role in mass balance and acidity calculations, weathering was studied extensively…" to "Due to its central role in mass balance and acidity calculations, weathering was studied extensively during the 1980s and 1990s…", to link to the earlier text about critical loads etc.

15  -Page 10, line 15, space removed.

-Page 11, line 8 : "Gustafson et al. (in review)"and "Belyazid et al., (in review)" was changed to "Gustafson et al., in review" and "Belyazid et al., in review"

-Page 11, line 13: "(Banfield et al., 1999; Finlay et al., 2009; Finlay et al. (in review)" was changed to "(Banfield et al., 1999; Finlay et al., 2009; Finlay et al., in review)".

20  -Page 11, line 27: Change "Table 1 & 3") to "Table 1", since Table 3 is a results table, referred to later.

-Page 12, line 11 : A missing word "described" was added.

-Page 12, line 21: Space removed.

-Page 15, line 10: Changed from "Rosenstock et al. ( in review)" to "Rosenstock et al., in review)"

-Page 15, line 33: "and" changed to "at".

25  -Page 16, line 14: The interval should be 31-85 (see Table 3), and was changed accordingly.

-Page 17, line 2: Interval corrected to 35-43, according to Table 3.

Page 17, line 14-15: "in most cases PROFILE and the depletion method" was removed.

-Page 17, line 24-25: Addition of "for this discrepancy": "The reasons for this discrepancy are not fully known…" for clarification.

-Page 18, line 13: "Sect." changed to "section", in accordance with the rest of the manuscript.

-Page 19, line 6-7: Correction of Region 4 to region 5 on two places.

5 -Page 19, line 9: Correction from 20 to 24 (we found a mistake in the calculations)

-Page 19, line 14-17: Sentence reformulated to better fit to Figure 6-7, and to adapt to the structure of sentence below and above (comparing havest losses with weathering rates, not weathering rates with harvest losses). Old sentence: "The Depletion method gave generally higher weathering rates than harvest losses at stem-only harvesting in the northern sites, but lower in the southern sites." New sentence: "The harvest losses at stem-only harvesting was lower or at the same level as

10 weathering rates according to the depletion method in the northern sites, but in most cases higher in the southern sites."

-Page 19, line 19: We changed the interval to 5-740%, since we by mistake didn´t include the large difference between the depletion method and WTH in Skånes Värsjö before (see Figure 6).

-Page 20, line 2-3: We removed "s" from "weathering rates", and accordingly changed from "were" to "was".

-Page 20, line 3: "i.e." added before "that".

15 -Page 20, line 11: McGivney et al (in review) has been changed to McGivney et al., 2019.

-Page 20, line 16: Removal of one "to".

-Page 20, line 21: "coditions" corrected to "conditions".

-Page 21, line 30: "Gazze" changed to "Gazzè".

-Page 22, line 31: A "3" is converted to superscript.

20 -Page 23, line 5: Typo corrected, PROfFILE has been changed to PROFILE.

-Page 28, line 10: "Erlandsson" was changed to "Erlandsson Lampa".

-Page 28-36: The reference list has been looked over, and minor changes have been done, to match the style template. For the special issue papers, author lists, title and status (in review or accepted) were updated. All changed can be seen in the "track changes" version of the manuscript below. The references added based on the Associate editor comments, according

25 to above, have been added to the reference list.

Page 31, line 21: "Gazze" changed to "Gazzè"

-Page 38, Table 2 : Incorrect characters (30333 removed from column Risfallet B, row Quartz. It was hided in the document, but could have become unhided at the layout.)

-Page 39, Table 3: The column name "Sr" has been changed to "Budget: Sr" and the column name "MAGIC" to Budget: MAGIC, since those methods are budget approaches, described under that chapter in the Methods section.

-Page 39, Table 3: A sentence has been added to the caption: "Values in parentheses after the intervals represent the middle of the intervals." The middle of the interval with MAGIC in Stubbetorp (35) has been added.

5   -Page 39, under Table 3: Clarification of the explanation of footnoot a.

-Page 45-46, Figure 4: "meq m$^{-2}$ y$^{-2}$" was corrected to "meq m$^{-2}$ y$^{-1}$".

-On multiple places in the manuscript (Page 11, line 17; Page 24, line 20; Page 24, line 23; Page 30, line 26; Page 35, line 17: "Erlandsson" was changed to "Erlandsson Lampa".

[revised manuscript text omitted]